# Endoglycan plays a role in axon guidance by modulating cell adhesion

**Thomas Baeriswyl[†], Alexandre Dumoulin[†], Martina Schaettin[†], Georgia Tsapara[†], Vera Niederkofler, Denise Helbling, Evelyn Avilés, Jeannine A Frei, Nicole H Wilson, Matthias Gesemann, Beat Kunz, Esther T Stoeckli***

Department of Molecular Life Sciences and Neuroscience Center Zurich, University of Zurich, Zurich, Switzerland

**Abstract** Axon navigation depends on the interactions between guidance molecules along the trajectory and specific receptors on the growth cone. However, our in vitro and in vivo studies on the role of Endoglycan demonstrate that in addition to specific guidance cue – receptor interactions, axon guidance depends on fine-tuning of cell-cell adhesion. Endoglycan, a sialomucin, plays a role in axon guidance in the central nervous system of chicken embryos, but it is neither an axon guidance cue nor a receptor. Rather, Endoglycan acts as a negative regulator of molecular interactions based on evidence from in vitro experiments demonstrating reduced adhesion of growth cones. In the absence of Endoglycan, commissural axons fail to properly navigate the midline of the spinal cord. Taken together, our in vivo and in vitro results support the hypothesis that Endoglycan acts as a negative regulator of cell-cell adhesion in commissural axon guidance.

**\*For correspondence:**
esther.stoeckli@mls.uzh.ch

[†]These authors contributed equally to this work

**Competing interests:** The authors declare that no competing interests exist.

## Introduction

Cell migration and axonal pathfinding are crucial aspects of neural development. Neurons are born in proliferative zones from where they migrate to their final destination. After arrival, they send out axons that have to navigate through the tissue to find the target cells with which they establish synaptic contacts. Intuitively, it is clear that the same cues provided by the environment can guide both cells and axons to their target. Although we know relatively little about guidance cues for cells compared to guidance cues for axons, both processes are dependent on proper cell-cell contacts (*Gomez and Letourneau, 2014*; *Short et al., 2016*).

One of the best-studied model systems for axon guidance are the commissural neurons located in the dorsolateral spinal cord (*de Ramon Francàs et al., 2017*; *Stoeckli, 2017* and *Stoeckli, 2018*). These neurons send out their axons toward the ventral midline under the influence of the roof plate-derived repellents BMP7 (*Augsburger et al., 1999*) and Draxin (*Islam et al., 2009*). At the same time, axons are attracted to the floor plate, their intermediate target, by Netrin (for a review on Netrin function, see *Boyer and Gupton, 2018*), VEGF (*Ruiz de Almodovar et al., 2011*), and Shh (*Yam et al., 2009* and *Yam et al., 2012*). At the floor-plate border, commissural axons require the short-range guidance cues Contactin2 (aka Axonin-1) and NrCAM to enter the midline area (*Stoeckli and Landmesser, 1995*; *Stoeckli et al., 1997*; *Fitzli et al., 2000*; *Pekarik et al., 2003*). Slits and their receptors, the Robos, were shown to be required as negative signals involved in pushing axons out of the midline area (*Long et al., 2004*; *Blockus and Chédotal, 2016*; *Pignata et al., 2019*). Members of the Semaphorin family are also involved in midline crossing, either as negative signals mediated by Neuropilin-2 (*Zou et al., 2000*; *Parra and Zou, 2010*; *Nawabi et al., 2010*; *Charoy et al., 2012*), or as receptors for floor-plate derived PlexinA2 (*Andermatt et al., 2014a*). Once commissural axons exit the floor-plate area, they turn rostrally along the longitudinal axis of the spinal cord. Morphogens of the Wnt family (*Lyuksyutova et al., 2003*; *Domanitskaya et al., 2010*; *Avilés and Stoeckli, 2016*) and Shh (*Bourikas et al., 2005*; *Wilson and Stoeckli, 2013*) were

identified as guidance cues directing post-crossing commissural axons rostrally. In the same screen that resulted in the discovery of Shh as a repellent for post-crossing commissural axons (*Bourikas et al., 2005*), we found another candidate that interfered with the rostral turn of post-crossing commissural axons. This candidate gene was identified as *Endoglycan*.

Endoglycan is a member of the CD34 family of sialomucins (*Nielsen and McNagny, 2008*; *Sassetti et al., 2000*; *Furness and McNagny, 2006*). The family includes CD34, Podocalyxin (also known as Thrombomucin, PCLP-1, MEP21, or gp135), and Endoglycan (also known as Podocalyxin-like 2). They are single-pass transmembrane proteins with highly conserved transmembrane and cytoplasmic domains. A C-terminal PDZ recognition site is found in all three family members (*Furness and McNagny, 2006*; *Nielsen and McNagny, 2008*). The hallmark of sialomucins is their bulky extracellular domain that is negatively charged due to extensive N- and O-glycosylation. Despite the fact that CD34 was identified more than 20 years ago, very little is known about its function. It has been widely used as a marker for hematopoietic stem cells and precursors. Similarly, Podocalyxin is expressed on hematopoietic stem and precursor cells. In contrast to CD34, Podocalyxin was found in podocytes of the developing kidney (*Kerjaschki et al., 1984*; *Doyonnas et al., 2005*; *Furness and McNagny, 2006*). In the absence of Podocalyxin, podocytes do not differentiate, resulting in kidney failure and thus perinatal lethality in mice (*Doyonnas et al., 2001*). *Podocalyxin*, but not *CD34*, is expressed widely in the developing and mature mouse brain (*Vitureira et al., 2005*; *Vitureira et al., 2010*). Podocalyxin was shown to induce microvilli and regulate cell-cell adhesion via its binding to the NHERF ($Na^+/H^+$ exchanger regulatory factor) family of adaptor proteins that link Podocalyxin to the actin cytoskeleton (*Nielsen et al., 2007*; *Nielsen and McNagny, 2008*; *Nielsen and McNagny, 2009*). Like *Podocalyxin*, *Endoglycan* is expressed in the brain and in the kidney. Only low levels were found in hematopoietic tissues (*Sassetti et al., 2000*). To date, nothing is known about the function of Endoglycan.

Based on its temporal and spatial expression pattern, we first analyzed the function of Endoglycan in the embryonic chicken spinal cord. In the absence of Endoglycan, commissural axons failed to turn rostrally upon floor-plate exit. Occasionally, they were observed to turn already inside the floor-plate area. Furthermore, the trajectory of commissural axons in the midline area was tortuous in embryos lacking Endoglycan, but straight in control embryos. Live imaging data of dI1 axons crossing the floor plate confirmed changes in axon – floor plate interaction. In addition, axons were growing more slowly after silencing *Endoglycan* and faster after overexpression of *Endoglycan*. In vitro assays confirmed a lower adhesive strength of growth cones to a layer of HEK cells expressing Endoglycan. Similarly, adhesion between growth cones overexpressing Endoglycan and control HEK cells was reduced, in agreement with a model suggesting that Endoglycan acts as a negative regulator of cell-cell adhesion during axon guidance.

## Results

### Endoglycan was identified as a candidate guidance cue for commissural axons

In a subtractive hybridization screen, we identified differentially expressed floor-plate genes as candidate guidance cues directing axons from dorsolateral commissural neurons (dI1 neurons) along the longitudinal axis after midline crossing (*Bourikas et al., 2005*; see Materials and methods). Candidates with an expression pattern that was compatible with a role in commissural axon navigation at the midline were selected for functional analysis using in ovo RNAi (*Pekarik et al., 2003*; *Wilson and Stoeckli, 2011*). One of these candidates that interfered with the correct rostral turning of commissural axons after midline crossing turned out to be *Endoglycan*, a member of the CD34 family of sialomucins.

CD34 family members share a common domain organization that consists of a mucin-like domain followed by a cysteine-containing globular domain, a membrane associated stalk region, a transmembrane spanning domain and the cytoplasmic domain (*Figure 1*; *Sassetti et al., 2000*; *Furness and McNagny, 2006*; *Nielsen and McNagny, 2008*). With the exception of the mucin-like domain at the N-terminus, the conservation between species orthologues of CD34, Endoglycan and Podocalyxin is in the range of 80%, but drops below 40% within the mucin domain. Homologies of these paralogous proteins within the same species are generally only in the range of 40% (*Figure 1*),

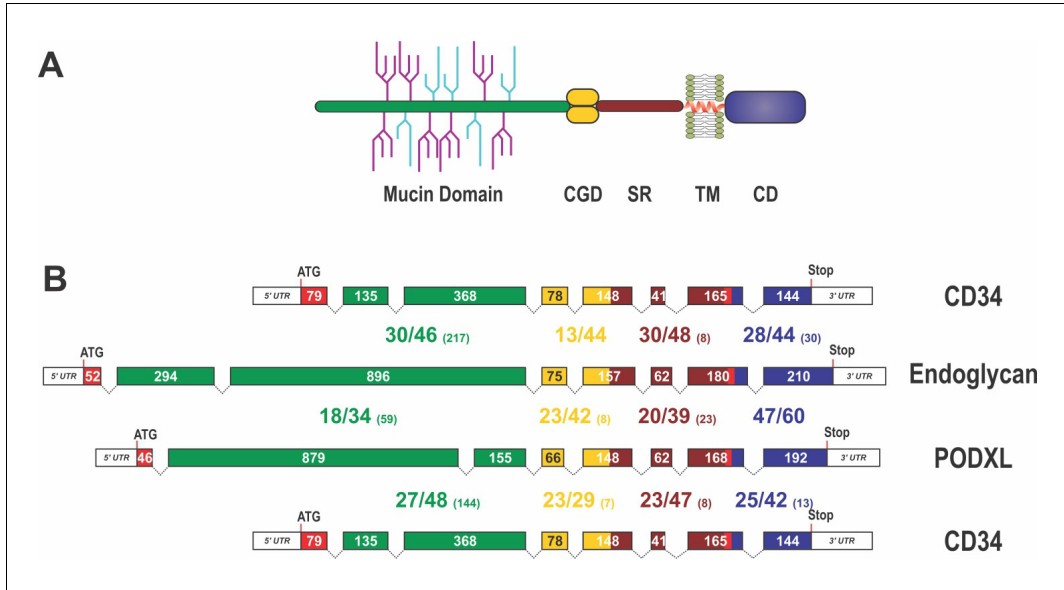

**Figure 1.** Domain organization, exon alignment, and conservation of the CD34 family of sialomucins. (**A**) Schematic drawing of the domain organization of CD34 family members. They all contain an N-terminal, highly glycosylated, mucin domain (green), a cysteine-containing globular domain (CGD, yellow), a juxtamembrane stalk region (SR, brown), as well as a transmembrane alpha-helix (TM, red) and a cytoplasmic domain (CD, blue). O-linked glycosylation sites within the mucin domain are depicted in light blue, whereas further sialylated residues are symbolized in purple. N-linked glycosylation sites are not shown. Note, that the indicated glycosylation sites in this scheme are only symbolizing the extensive amount of glycosylation in sialomucins and are not representing the actual position of glycosylation. (**B**) Exon organization and domain conservation of sialomucins. Transcripts of CD34 family members are encoded by eight separate exons (colored boxes). While the length of exons coding for the cysteine containing globular domain (yellow), the juxtamembrane stalk region (brown), and the cytoplasmic domain (blue) are more or less conserved (exon sizes are given within the boxes), exons coding for the mucin domain (green) vary markedly in their length and organization. The translational start sites are highlighted by the ATG and the end of the coding sequences are indicated by the given Stop codon. Protein homology between the different chicken sialomucins is depicted by the large numbers between the exon pictograms. All domains were compared separately and the colors used indicate the corresponding domains. The first number indicates identical amino acids between the compared proteins, the second number represents conserved residues and the number in brackets designates the amount of gap positions within the alignment of the domains (e.g. 28/44 (30)). The alignment was done using MUSCLE version 3.7 configured to the highest accuracy (*Edgar, 2004*). Single gap positions were scored with high penalties, whereas extensions of calculated gaps were less stringent. Using such parameters homologous regions of only distantly related sequences can be identified. Note, that within the mucin domain only some blocks, interspaced by sometimes large gap regions, are conserved between the different proteins.

demonstrating that, while they might share a similar overall structure, the structure can be built by quite diverse primary amino acid sequences.

*Endoglycan* was expressed mainly in the nervous system during development, as levels in non-neuronal tissues were much lower (*Figure 2* and *Figure 2—figure supplement 1*). In the neural tube, *Endoglycan* was expressed ubiquitously, including floor-plate cells at HH21 (Hamburger and Hamilton stage 21; *Hamburger and Hamilton, 1951*; *Figure 2*). By HH25, expression was still found throughout the neural tube with higher levels detected in dorsal neurons (including dI1 neurons) and motoneurons. *Endoglycan* expression was also maintained in the floor plate (*Figure 2B*). For functional analysis, dsRNA was produced from the *Endoglycan* cDNA fragment obtained from a screen and used for in ovo electroporation of the spinal cord at HH18 (*Figure 2D*). The analysis of commissural axons' trajectories at HH26 by DiI tracing in 'open-book' preparations (*Figure 2E*; quantified in *Figure 2L*) revealed either failure to turn or erroneous caudal turns along the contralateral floor-plate border in embryos lacking Endoglycan in the floor plate (*Figure 2G*), in only the dorsal spinal cord (*Figure 2J*), or in one half of the spinal cord, including the floor plate (*Figure 2H,I*).

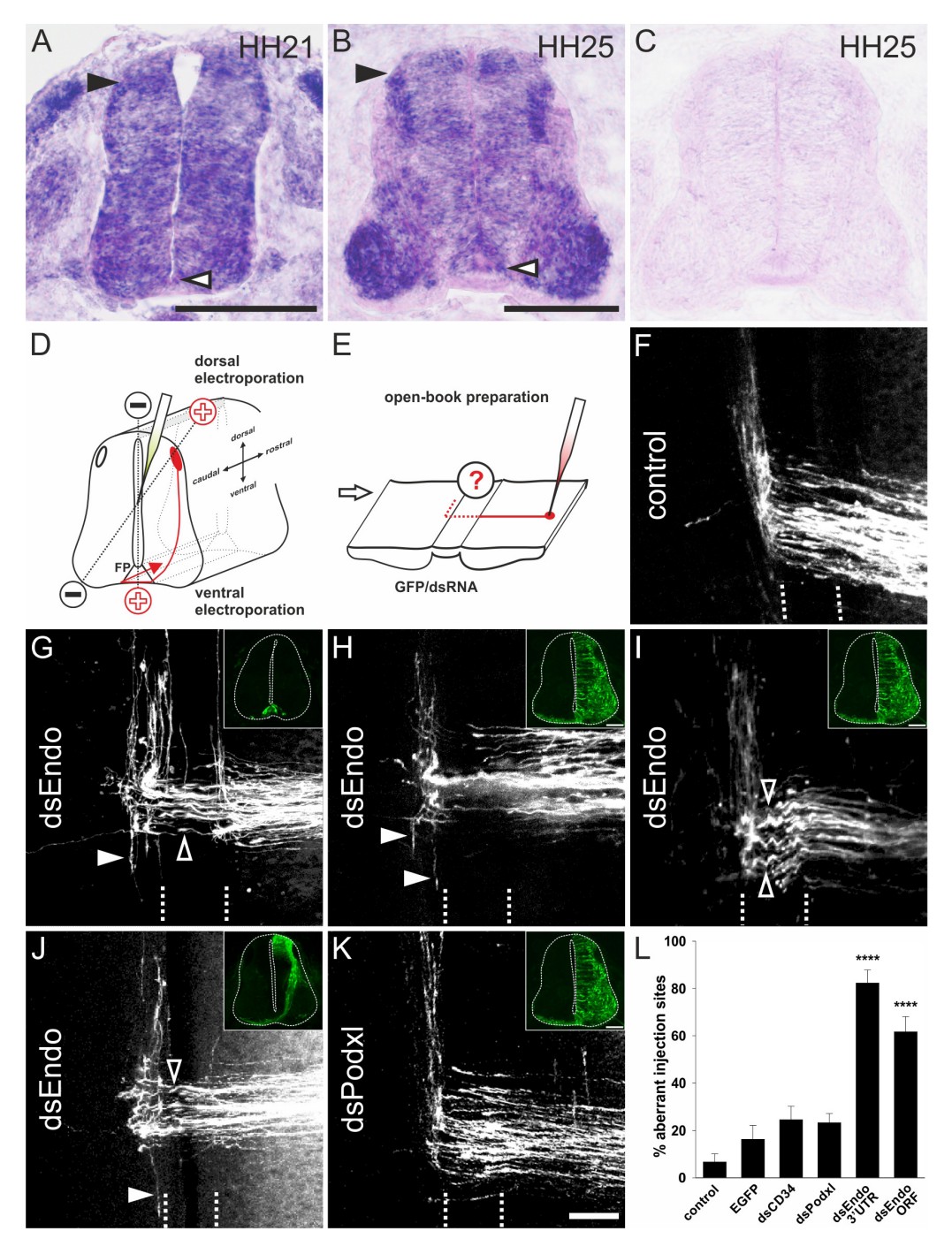

**Figure 2.** Endoglycan is required for correct turning of post-crossing commissural axons. (A,B) Endoglycan is expressed in the developing neural tube during commissural axon guidance. Endoglycan is expressed throughout the neural tube at HH21, including the floor plate (white arrowhead) (A). (B) At HH25, Endoglycan is still found in most cells of the spinal cord. High levels are found in motoneurons and interneurons, including the dorsal dI1 neurons (black arrowhead), and in the floor plate (white arrowhead). No staining was found when hybridization was carried out with a sense probe (C). Commissural axon pathfinding was analyzed in 'open-book' preparations (D,E; see Materials and methods for details). The positions of the electrodes for dorsal and ventral electroporation are indicated (D). In control embryos at HH26, commissural axons have crossed the floor plate and turned rostrally along the contralateral floor-plate border (F). In contrast, after downregulation of Endoglycan (G–J) commissural axons failed to turn along the contralateral floor-plate border or they turned randomly either rostrally or caudally (arrowheads in G-J). Occasionally, axons were turning already inside the floor plate (open arrowhead in G). A closer look at the morphology of the axons in the floor plate revealed their tortuous, 'corkscrew'-like trajectory across the midline at many DiI injection sites (open arrowheads in I). To knockdown Endoglycan either in the floor plate or in commissural neurons, the

*Figure 2 continued on next page*

*Figure 2 continued*

ventral or dorsal spinal cord was targeted as indicated in (D) (see inserts in G and J, respectively). Phenotypes were the same as those observed after targeting one half of the spinal cord including the floor plate (H,I). Pathfinding was normal in embryos electroporated with dsRNA derived from Podocalyxin (K). The quantification of injection sites with pathfinding errors after targeting the floor plate or one half of the spinal cord is shown in (L). Pathfinding errors were seen only at 6.7±3.4% of the injection sites in untreated control embryos (n=10 embryos, 45 injection sites). In control embryos injected and electroporated with the EGFP plasmid alone, pathfinding errors were found at 16.2±6% of the injection sites (n=17 embryos, 92 injection sites). Injection and electroporation of dsRNA derived from CD34 (24.6±5.8%, n=8 embryos, 80 injection sites) and Podocalyxin (23.3±3.9%, n=17 embryos, 147 injection sites) did not affect midline crossing and turning behavior of commissural axons. By contrast, 82.3±5.6% (n=11 embryos, 65 sites) and 61.7±6.4% (n=18, 161 sites) of the injection sites in embryos injected with dsRNA derived from the 3′-UTR or the ORF of Endoglycan, respectively, showed aberrant pathfinding of commissural axons. One-way ANOVA with Tukey's multiple comparisons test. P values ****<0.0001, compared to EGFP-injected control groups. The two groups electroporated with dsRNA derived from Endoglycan were not different from each other. Values represent average percentage of DiI injection sites per embryo with aberrant axonal navigation ± standard error of the mean. Source data and statistics are available in the *Figure 2—source data 1* spreadsheet. Bar: 50 µm.

The online version of this article includes the following source data and figure supplement(s) for figure 2:

**Source data 1.** Raw data and statistics for *Figure 2L*.
**Figure supplement 1.** Endoglycan is mainly expressed in the developing nervous system.
**Figure supplement 2.** Podocalyxin and CD34 are expressed in the developing spinal cord during midline crossing of dI1 commissural axons.
**Figure supplement 3.** Electroporation of dsRNA derived from Endoglycan effectively downregulates Endoglycan mRNA.
**Figure supplement 3—source data 1.** Raw data demonstrating efficient downregulation of Endoglycan.

Furthermore, axons were occasionally found to turn prematurely either before midline crossing or within the floor-plate area. Detailed analysis of the axonal morphology in the floor-plate area revealed a tortuous, 'corkscrew'-like trajectory in embryos lacking Endoglycan in dI1 neurons and the floor plate (*Figure 2I*), whereas axons crossed the midline in a straight trajectory in untreated control embryos (*Figure 2F*) and in embryos injected and electroporated with dsRNA derived from either CD34 (not shown) or Podocalyxin (*Figure 2K*).

To demonstrate specificity of Endoglycan downregulation and to verify that the phenotype was not due to an off-target effect, we used three non-overlapping cDNA fragments to produce dsRNA. All fragments resulted in the same phenotypes. Downregulation of Endoglycan with dsRNA derived from the ORF resulted in 61.7 ± 6.4% injection sites with aberrant axon guidance. The effect on axon guidance was also seen with dsRNA derived from the 3'UTR, with 82.3 ± 5.6% of the injection sites with aberrant axon guidance (*Figure 2L*). In contrast, aberrant axonal pathfinding was seen only at 6.7 ± 3.4% of the injection sites in untreated control embryos. Values were 16.2 ± 6.0 for EGFP-expressing control embryos, 24.6 ± 5.8% for embryos transfected with dsRNA derived from *CD34*, and 23.3 ± 3.9% for embryos transfected with dsRNA derived from *Podocalyxin*. Thus, silencing either *CD34* or *Podocalyxin* did not interfere with correct navigation of axons at the midline. Because both of them were expressed in the developing spinal cord (*Figure 2—figure supplement 2*), these results further support the specificity of the observed effects of *Endoglycan* silencing.

## Lack of Endoglycan affects the morphology of the floor plate only after dI1 axons have crossed the midline

Because the hallmark of sialomucins is their bulky, negatively charged extracellular domain with extensive glycosylation, a role as regulators of cell-cell adhesion has been postulated (*Vitureira et al., 2010*; *Takeda et al., 2000*; *Nielsen and McNagny, 2008* and *Nielsen and McNagny, 2009*). This together with our observation that commissural axons have a 'corkscrew'-like phenotype in the midline area in Endoglycan-deficient embryos prompted us to analyze the morphology of the floor plate. Sections were taken from the lumbar level of the spinal cord at HH25 from control-treated and experimental embryos and stained for HNF3β/FoxA2 to label floor-plate cells, and for Contactin2 (aka Axonin-1) to label commissural axons (*Figure 3*). In untreated (*Figure 3A–C*) and control-treated embryos (*Figure 3D–F*), HNF3β/FoxA2-positive cells were aligned to form the characteristic triangular shape of the floor plate. In particular, the ventral border of the floor plate, where commissural axons traverse the midline was smooth, because all floor-plate cells were precisely aligned (*Figure 3A,D*). In contrast, floor-plate cells were no longer aligned to form a smooth ventral border in embryos lacking Endoglycan after electroporation of dsEndo into one half of the spinal cord (*Figure 3G,J*). On the one hand, floor-plate cells were found dislocated

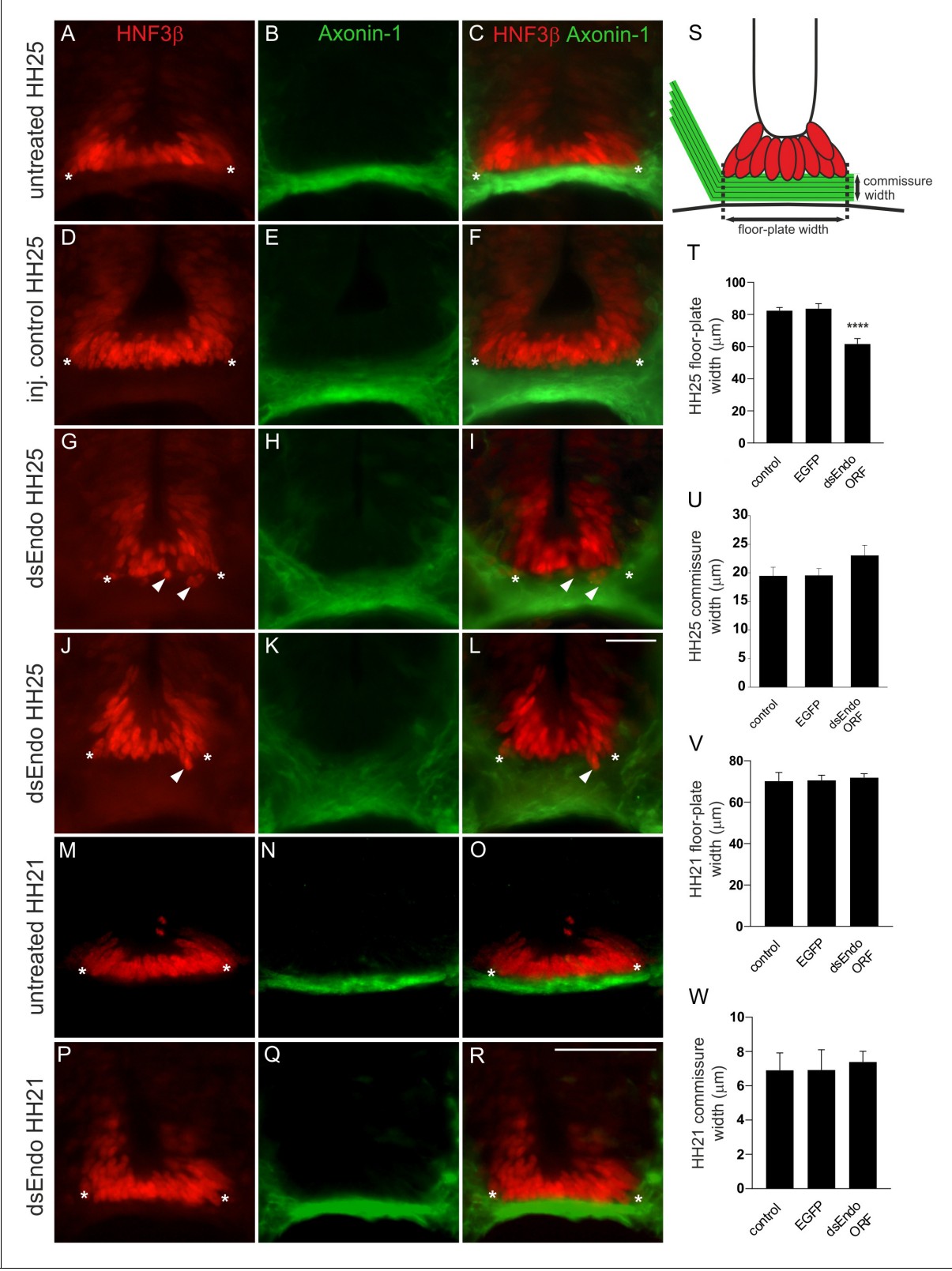

**Figure 3.** After downregulation of Endoglycan in one half of the spinal cord, the floor-plate morphology is compromised only after axonal midline crossing. In untreated (A-C) and control-treated embryos (D-F), the floor plate is of triangular shape with floor-plate cells precisely aligned at the ventral border. There is no overlap between the floor plate (visualized by HNF3β staining; red) and the commissure (visualized by anti-Axonin1 staining; green). The shape of the floor plate is no longer triangular in embryos lacking Endoglycan (G-L). The floor-plate cells are not aligned ventrally (arrowheads in G,

*Figure 3 continued on next page*

*Figure 3 continued*

I, J, and L) and the floor plate appears to have gaps. This change in morphology is only seen at HH25, when midline crossing is completed. When the floor-plate morphology was analyzed at HH21, there was no difference between control (M-O) and experimental embryos electroporated with dsRNA derived from Endoglycan (P-R). Note that some more ventral commissural axon populations have crossed the floor plate at this stage. But overall, the number of axons that form the commissure at HH21 is still very small. The width of the floor plate (indicated by asterisks) was measured (S,T). There was no significant difference in spinal cord width at HH25 (400.2 ± 54.5 µm in untreated controls, 438.2 ± 30.3 µm in EGFP-expressing controls, and 394 ± 12.0 µm in dsEndo embryos), but floor plates were significantly narrower in embryos lacking Endoglycan (T; 61.6 ± 2.9 µm; n = 7 embryos; p<0.001) compared to untreated (82.4 ± 1.8 µm; n = 6 embryos) and EGFP-injected control embryos (83.6 ± 2.6 µm; n = 6 embryos). One-way ANOVA with Tukey's multiple comparisons test. The commissure had a tendency to be wider in experimental compared to control embryos but the effect was not statistically significant (U). The width of the floor plate was not different between groups when measured at HH21 (V) with 70.1 ± 4.2 µm (n = 3) for untreated and 70.6 ± 2.5 µm for EGFP controls (n = 4), compared to 71.8 ± 2.0 µm for experimental embryos (n = 3). No difference was seen in the width of the commissure. Two-tailed T-test. ****p<0.0001. Mean ± SEM are given. Electroporation was targeted to one side of the spinal cord as shown in the insert in *Figure 2H*. Bar: 50 µm. Source data and statistics are available in *Figure 3—source data 1* spreadsheet.

The online version of this article includes the following source data and figure supplement(s) for figure 3:

**Source data 1.** Raw data and statistics for *Figure 3*.
**Figure supplement 1.** Downregulation or overexpression of Endoglycan does not affect spinal cord patterning.
**Figure supplement 2.** Experimental manipulation of Endoglycan levels does not induce cell death in the floor plate.
**Figure supplement 3.** The errors in commissural axon pathfinding seen after perturbation of Endoglycan levels are not due to changes in the expression of known guidance cues for dI1 axons.
**Figure supplement 4.** Perturbation of Endoglycan expression does not affect guidance of post-crossing commissural axons indirectly by changing Shh or Wnt5a expression.

into the commissure formed by the Contactin2-positive axons (arrowheads in *Figure 3I,L*). On the other hand, the floor plate appeared to have gaps in embryos lacking Endoglycan. In addition, the floor-plate width was significantly narrower in embryos lacking Endoglycan in comparison to age-matched controls (*Figure 3S,T*). These changes in floor-plate morphology were not due to differences in cell differentiation or patterning (*Figure 3—figure supplement 1*). Furthermore, we can exclude cell death as a contributor to the changes in the floor plate, as we found no Cleaved Caspase-3-positive floor-plate cells in any of the conditions (*Figure 3—figure supplement 2*).

When embryos lacking Endoglycan were analyzed at HH21 (*Figure 3P–R*), that is at a time point when dI1 axons have reached but not yet crossed the floor plate, the morphology and the width of the floor plate were not different from controls (*Figure 3M–O*). In contrast to measurements of floor-plate width at HH25 (*Figure 3T*), the values for experimental and control embryos were not different at HH21 (*Figure 3V*). Taken together, these results suggested that the absence of Endoglycan did not affect primarily cell-cell adhesion between floor-plate cells. Rather the altered floor-plate morphology appeared to be an indirect effect of changes in axon to floor-plate adhesion.

To provide more evidence for this idea, we first ruled out an effect of the perturbation of Endoglycan levels on the expression of known guidance cues of dI1 axons, such as Contactin2 (Axonin-1) or NrCAM (*Figure 3—figure supplement 3*). Similarly, we did not find changes in the expression of Shh and Wnt5a, morphogens that are known to direct post-crossing dI1 axons rostrally (*Figure 3—figure supplement 4*; *Bourikas et al., 2005*; *Lyuksyutova et al., 2003*).

An alternative way of demonstrating the requirement for Endoglycan in both floor plate and commissural axons were rescue experiments (*Figure 4*). We used dsRNA derived from the 3'-UTR and expressed the ORF of *Endoglycan* either under control of the Math1 enhancer (expression only in dI1 neurons) or the Hoxa1 enhancer for floor-plate-specific expression. Because expression of these plasmids in control embryos (overexpression) resulted in aberrant behavior of axons at the floor plate, we used three different concentrations of plasmid for our rescue experiments and obtained a dose-dependent effect on axon guidance. Expression of high doses of *Endoglycan* was never able to rescue the axon guidance phenotype. However, axon guidance was not different from control embryos after transfection of dI1 neurons with a low concentration, or after transfection of floor-plate cells with a medium concentration of the *Endoglycan* ORF (*Figure 4B*; *Table 1*). Interestingly, the source of Endoglycan did not matter, but the amount of Endoglycan did. These findings were consistent with the idea that Endoglycan could be regulating adhesion, as both too much, but also too little adhesion would be a problem for axonal navigation. Furthermore, these results suggested that Endoglycan did not act as a receptor.

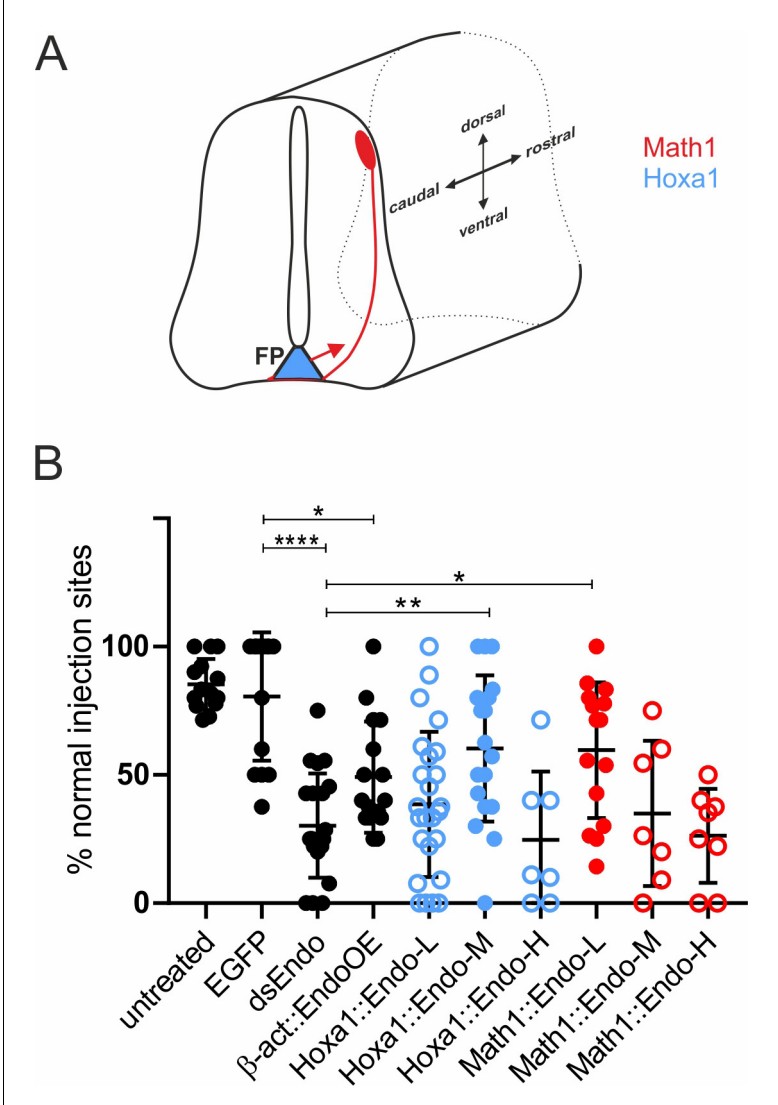

**Figure 4.** Too much or too little Endoglycan causes aberrant axon guidance. Because silencing Endoglycan either in commissural neurons or in the floor plate caused the same type of axon guidance defects, we wanted to test the idea that the presence of an adequate amount, but not the source of Endoglycan was important. We therefore downregulated Endoglycan by transfection of dsRNA derived from the 3'UTR of Endoglycan into one half of the spinal cord. We then tried to rescue the aberrant axon guidance phenotype by co-electroporation of the Endoglycan ORF specifically in dI1 neurons (using the Math1 enhancer, red) or in the floor plate (using the Hoxa1 enhancer; blue, **A** and **B**). The rescue constructs were used at a concentration of 150 (L = low), 300 (M = medium), and 750 (H = high) ng/μl, respectively. In both cases, rescue was only possible with one concentration: the medium concentration of the Endoglycan plasmid driven by the Hoxa1 promoter and the low concentration of the plasmid driven by the Math1 promoter. The lowest concentration of the Hoxa1-driven construct and the two higher concentrations of the Math1-driven constructs were not able to rescue the aberrant phenotype. Note that the amounts of Endoglycan cannot be compared between the Math1- and the Hoxa1 enhancers, as they differ in their potency to drive expression. However, we can conclude a response in a dose-dependent manner in both cases. Statistical analysis by one-way ANOVA: *p<0.05, **p<0.01, ***p<0.001. Values are shown ± standard deviation. See *Table 1* for values. Source data and statistics are available in *Figure 4—source data 1* spreadsheet.

The online version of this article includes the following source data for figure 4:

**Source data 1.** Raw data and statistics for rescue experiments.

**Table 1.** The amount, but not the source of Endoglycan matters.

| Treatment | No of embryos | No of inj. sites | % inj sites normal PT | p-Value | % inj sites stalling | % inj sites no turn |
|---|---|---|---|---|---|---|
| Untreated | 14 | 111 | 85.3 ± 2.6 | 0.999 | 0.8 ± 0.8 | 14.6 ± 2.7 |
| EGFP | 14 | 85 | 80.5 ± 6.7 | 1 | 5.1 ± 2.4 | 18.6 ± 7.4 |
| dsEndo | 21 | 161 | 30.2 ± 4.4 | <0.0001 | 31.6 ± 5 | 50.1 ± 3.9 |
| β-actin::EndoOE | 16 | 91 | 49.1 ± 5.4 | 0.0177 | 20 ± 5.1 | 30 ± 5 |
| Hoxa1::Endo-L | 25 | 234 | 38.5 ± 5.7 | <0.0001 | 24.1 ± 4.8 | 44.5 ± 5 |
| Hoxa1::Endo-M | 18 | 136 | 60.3 ± 6.7 | 0.3686 | 10.9 ± 4.2 | 32.1 ± 5 |
| Hoxa1::Endo-H | 7 | 72 | 24.6 ± 10.1 | <0.0001 | 40.0 ± 11.3 | 44.1 ± 7 |
| Math1::Endo-L | 15 | 157 | 59.6 ± 6.8 | 0.3738 | 12.3 ± 3.5 | 30.9 ± 5.4 |
| Math1::Endo-M | 7 | 86 | 35 ± 10.7 | 0.0031 | 31.7 ± 7.2 | 42.4 ± 9.3 |
| Math1::Endo-H | 8 | 86 | 26.3 ± 6.5 | <0.0001 | 50.8 ± 6.4 | 40.5 ± 6.1 |

Concomitant expression of Endoglycan could rescue the aberrant axon guidance phenotype induced by the downregulation of Endoglycan throughout the spinal cord. It did not matter whether Endoglycan was expressed under the Hoxa1 enhancer for specific expression in floor-plate cells, or under the Math1 enhancer for specific expression in dI1 neurons. However, the rescue effect was dose-dependent. Too little, or too much Endoglycan was inducing axon guidance defects. For rescue, *Endoglycan* cDNA under the control of the Hoxa1 enhancer (Hoxa1::Endo) or the Math1 enhancer (Math1::Endo) were injected at 150 ng/μl (low, L), 300 ng/μl (medium, M), or 750 ng/μl (high, H). The same criteria for quantification were applied as for the results shown in **Figure 2**. There was no fundamental difference between too much or too little stickiness. In both conditions, we found stalling in the floor plate and failure to turn into the longitudinal axis. The only difference was that we did not find any 'corkscrew' phenotypes after overexpression of Endoglycan. The number of embryos and the number of DiI injection sites analyzed per group are indicated. The average % of injection sites with normal axon guidance phenotypes (± standard error of the mean) and the p-value for the comparison between the respective group and the control-treated (EGFP-expressing) group are given. The last two columns list the average values for injection sites with the majority of axons stalling in the floor plate and the majority of axons not turning into the longitudinal axis. Because some injection sites can have both phenotypes, the values do not add up to 100%.

## Endoglycan is a negative regulator of cell adhesion

The observation that downregulation of Endoglycan seemed to increase the adhesion between commissural axons and floor-plate cells, together with the knowledge about its molecular features, led us to hypothesize that Endoglycan might act as a negative regulator of cell-cell adhesion. As a first test, we counted the number of commissural neurons that adhered to a layer of HEK cells stably expressing Endoglycan compared to control HEK cells (*Figure 5*). Expression of Endoglycan reduced the number of commissural neurons on Endoglycan-expressing HEK cells compared to control HEK cells. These suggested 'anti-adhesive' properties of Endoglycan depended on its post-translational modification (*Figure 5B,D*). Enzymatic removal of sialic acid by Neuraminidase or of O-linked glycans by O-glycosidase abolished the 'anti-adhesive' properties of Endoglycan expressed in HEK cells. Endoglycan had a similar effect on the adhesion of motoneurons to a layer of HEK cells expressing Endoglycan that also depended on the post-translational modification of Endoglycan with O-glycosylation and sialylation (*Figure 5—figure supplements 1* and *2*).

As an additional experiment to assess differences in adhesive strength modulated by Endoglycan, we adapted the growth cone blasting assay developed by Lemmon and colleagues (*Lemmon et al., 1992*). mRFP-expressing commissural neurons were cultured on a layer of HEK cells and growth cones were blasted off with a constant flow of buffer delivered from a glass micropipette (*Figure 6A*, *Video 1*). Thus, the time it took to detach the growth cone from the HEK cell layer was taken as an indirect measurement of adhesive strength. On average, it took only about half the time to detach commissural growth cones from HEK cells expressing Endoglycan compared to control HEK cells (*Figure 6B*). In line with this difference, overexpression of Endoglycan in neurons growing on control HEK cells significantly decreased the time it took to blast off growth cones compared to control neurons (*Figure 6C*). The difference in adhesive strength was not explained or markedly affected by differences in adhesive area between growth cones and HEK cells, as the growth cone areas did not differ between control and experimental conditions (*Figure 7*). Taken together, our in vitro experiments demonstrated that Endoglycan negatively regulated the adhesive strength between commissural growth cones and the HEK cells no matter whether Endoglycan was expressed in the growth cone or in the HEK cells.

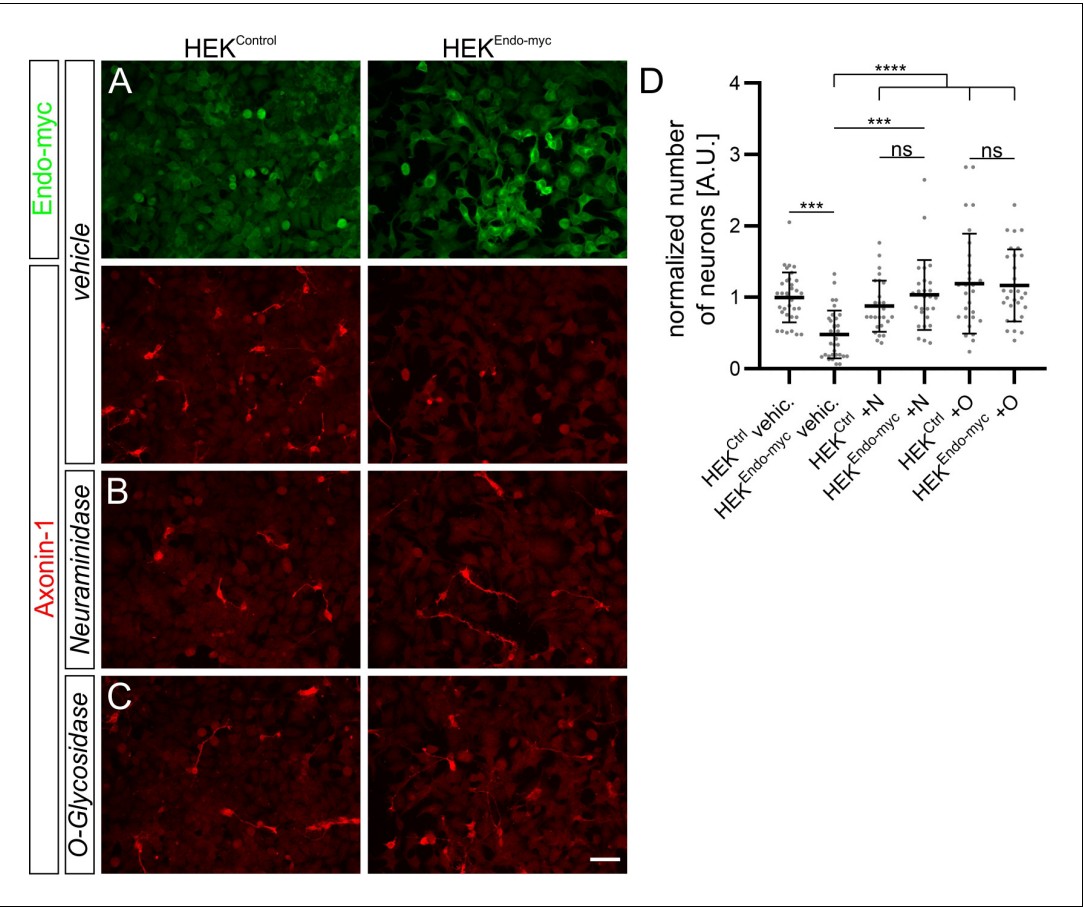

**Figure 5.** Endoglycan expression reduces adhesion of commissural neurons in vitro. (A-C) Commissural neurons dissected from HH25/26 chicken embryos were cultured on a layer of control HEK cells or HEK cells stably expressing human Endoglycan. Neurons were allowed to attach for 16 hr. Staining for Axonin-1 revealed a pronounced decrease in the number of commissural neurons on HEK cells expressing Endoglycan compared to control HEK cells (A and D). For each replicate, the number of neurons attached to HEK cells expressing Endoglycan was normalized to the number of cells attached to control HEK cells (D). The number of commissural neurons attached to control HEK cells was more than twice the number on Endoglycan-expressing HEK cells (only 0.47 ± 0.34 compared to 1 ± 0.35). Treatment with Neuraminidase (+N) or O-glycosidase (+O) abolished the difference. See *Table 2* for values. N(replicates)=4. ns (not significant), ***p<0.001, ****p<0.0001, one-way ANOVA with Tukey's multiple comparisons test. Error bars represent standard deviations. O, O-glycosidase; N, Neuraminidase; vehic, vehicle; ctrl, control. Scale bar: 50 µm. Source data and statistics are available in *Figure 5—source data 1* spreadsheet.

The online version of this article includes the following source data and figure supplement(s) for figure 5:

**Source data 1.** Commissural neuron counts on HEK cells expressing Endoglycan.

**Figure supplement 1.** Endoglycan expression reduces adhesion of motor neurons in vitro.

**Figure supplement 1—source data 1.** Motoneuron counts on HEK cells expressing Endoglycan.

**Figure supplement 2.** Post-translational modification of Endoglycan is required for its anti-adhesive effects on motoneurons.

**Figure supplement 2—source data 1.** Motoneuron counts on HEK cells expressing Endoglycan treated with Neuraminidase and O-Glycosidase.

---

Next, we tested our hypothesis that Endoglycan was a negative regulator of adhesion by manipulating the balance of adhesion between commissural axons and the floor plate in vivo. We had previously used a similar approach to demonstrate a role of RabGDI in Robo trafficking (*Philipp et al., 2012*). Commissural axons cross the midline because of the positive signals provided by the interaction of floor-plate NrCAM with growth cone Contactin2 (*Stoeckli and Landmesser, 1995*; *Stoeckli et al., 1997*; *Fitzli et al., 2000*). In the absence of NrCAM or Contactin2, commissural

**Table 2.** Quantification of the number of commissural neurons adhering to a layer of HEK cells and midline crossing of dI1 axons using live imaging.

*Figure 5*

| Part | Name | Value | Stdev | n(pictures) | N(replicates) |
|------|------|-------|-------|-------------|---------------|
| D | HEK$^{ctrl}$ vehic. | 1.00 | 0.35 | 34 | 4 |
| D | HEK$^{Endo-myc}$ vehic. | 0.47 | 0.34 | 34 | 4 |
| D | HEK$^{ctrl}$ vehic.+N | 0.88 | 0.36 | 28 | 4 |
| D | HEK$^{Endo-myc}$ vehic.+N | 1.03 | 0.49 | 28 | 4 |
| D | HEK$^{ctrl}$ vehic.+O | 1.19 | 0.70 | 29 | 4 |
| D | HEK$^{Endo-myc}$ vehic.+O | 1.17 | 0.50 | 29 | 4 |

*Figure 10*

| Part | Name | Value | Stdev | n(axons) | N(embryos) |
|------|------|-------|-------|----------|------------|
| B | Endo Ctrl | 5.38 | 1.25 | 161 | 3 |
| B | dsEndo | 5.50 | 1.20 | 168 | 3 |
| B | Endo OE | 4.35 | 1.38 | 234 | 3 |
| C | Endo Ctrl Ent-Mid | 2.74 | 0.94 | 161 | 3 |
| C | Endo Ctrl Mid-Ex | 2.64 | 1.01 | 161 | 3 |
| C | dsEndo Ent-Mid | 3.06 | 0.88 | 168 | 3 |
| C | dsEndo Mid-Ex | 2.45 | 1.02 | 168 | 3 |
| C | Endo OE Ent-Mid | 2.10 | 0.71 | 234 | 3 |
| C | Endo OE Mid-Ex | 2.26 | 0.83 | 234 | 3 |
| D | Endo Ctrl | 0.51 | 0.13 | 161 | 3 |
| D | dsEndo | 0.56 | 0.12 | 168 | 3 |
| D | Endo OE | 0.48 | 0.11 | 234 | 3 |
| E | Endo Ctrl | 0.49 | 0.13 | 161 | 3 |
| E | dsEndo | 0.44 | 0.12 | 168 | 3 |
| E | Endo OE | 0.52 | 0.11 | 234 | 3 |
| F | Endo Ctrl | 49.49 | 12.84 | 70 | 3 |
| F | dsEndo | 48.67 | 13.47 | 68 | 3 |
| F | Endo OE | 43.50 | 14.84 | 83 | 3 |
| G | Endo Ctrl | 47.45 | 11.58 | 70 | 3 |
| G | dsEndo | 40.87 | 10.44 | 68 | 3 |
| G | Endo OE | 41.09 | 11.18 | 83 | 3 |
| H | Endo Ctrl | 106.05 | 28.72 | 70 | 3 |
| H | dsEndo | 108.08 | 30.59 | 68 | 3 |
| H | Endo OE | 87.52 | 28.01 | 83 | 3 |

*Figure 11*

| Part | Name | Value | Stdev | n(axons) | N(embryos) |
|------|------|-------|-------|----------|------------|
| A | 1 st half | 49.49 | 12.84 | 70 | 3 |
| A | 2nd half | 47.45 | 11.58 | 70 | 3 |
| A | FP exit | 106.05 | 28.72 | 70 | 3 |
| B | 1 st half | 48.67 | 13.47 | 68 | 3 |
| B | 2nd half | 40.87 | 10.44 | 68 | 3 |
| B | FP exit | 108.08 | 30.59 | 68 | 3 |
| C | 1 st half | 43.50 | 14.84 | 83 | 3 |
| C | 2nd half | 41.09 | 11.18 | 83 | 3 |
| C | FP exit | 87.52 | 28.01 | 83 | 3 |

The table contains the detailed values from results presented in **Figure 5**, **Figure 10**, and **Figure 11**. Ent, entry; Mid, midline; Ex, exit; FP, floor plate; stdev, standard deviation.

axons fail to enter the floor plate and turn into the longitudinal axis prematurely along the ipsilateral floor-plate border. The positive signal derived from the Contactin2/NrCAM interaction depends on sufficient contact between growth cone and floor-plate cells. Thus, we hypothesized that the failure to detect the positive signal due to lower NrCAM levels on the floor-plate cells could be counteracted by a forced increase in growth cone-floor plate contact. We reasoned that the concomitant downregulation of NrCAM and Endoglycan would rescue the NrCAM phenotype, because the decrease in adhesion due to lower NrCAM, resulting in the failure of commissural axons to enter the floor plate, would be counteracted by an increase in adhesion in the absence of Endoglycan. This is indeed what we observed (*Figure 8*). As found previously (*Stoeckli and Landmesser, 1995*; *Pekarik et al., 2003*), axons were frequently turning prematurely along the ipsilateral floor-plate border in the absence of NrCAM (*Figure 8A*). In accordance with our hypothesis, ipsilateral turns were reduced to control levels when NrCAM and Endoglycan were downregulated concomitantly (*Figure 8B,G*). The rescue of the NrCAM phenotype was only seen for Endoglycan, as concomitant downregulation of NrCAM with Podocalyxin or CD34 had no effect on ipsilateral turns (*Figure 8*).

## Endoglycan levels modulate growth cone movement in the floor plate

To get more insight into the role of Endoglycan in the regulation of contacts between axons and floor-plate cells, we established ex vivo live imaging of commissural axons during midline crossing. Intact spinal cords of HH22 chicken embryos, which were co-injected and unilaterally electroporated with constructs expressing farnesylated td-Tomato (td-Tomato-f) under the control of the dI1 neuron-specific Math1 enhancer together with farnesylated EGFP (EGFP-f) under the control of the β-actin promoter, were cultured and imaged for 24 hr (*Figure 9*). This method allowed us to follow the behavior and trajectories of the very first wave of single dI1 axons entering, crossing, and exiting the floor plate in a conserved environment (arrowheads, *Figure 9A$_1$–A$_3$*). The expression of EGFP-f in all transfected cells and brightfield images helped us to define the floor-plate boundaries (white dashed lines) and midline (yellow dashed lines in *Figure 9B,C*).

Spinal cords of embryos co-injected with the Math1::tdTomato-f plasmid and dsRNA derived from Endoglycan (dsEndo) or a plasmid encoding chicken Endoglycan under the β-actin promoter (Endo OE) were imaged for 24 hr and compared to spinal cords dissected from control-injected embryos (Endo ctrl, *Video 2*, temporally color-coded projections in *Figure 9D$_1$,E$_1$and F$_1$*). In contrast to control-injected spinal cords, the post-crossing segment of dI1 axons was disorganized in dsEndo and Endo OE conditions. In both these conditions, caudal turns were seen (*Video 2*). As our in vivo data suggested a difference in adhesion between floor-plate cells and dI1 axons, we analyzed axonal midline crossing with kymographs in two different regions of interest (ROI; shown in *Figure 9D$_1$, E$_1$, F$_1$*). This allowed us to follow growth cone movement across the floor plate and along the floor-plate border. Interestingly, our analyses indicated that the transfection of dsRNA derived from Endoglycan in dI1 neurons and floor-plate cells led to a decrease in the growth cones' speed in the first half of the floor plate (13 μm/hr) and in an increase in the second half (44 μm/hr; *Figure 9E$_2$*) compared to control-injected spinal cords (*Figure 9D$_2$* and *Video 3*), where speed in the first and second halves did not differ (26 μm/hr). In spinal cords overexpressing *Endoglycan*, growth cone speed was accelerated in the entire floor plate (50 μm/h; *Figure 9F$_2$*; *Video 3*). The analysis of axon growth in a second ROI confirmed the disorganization seen in both mutants in *Video 2*. Although the axonal trajectories in control-injected embryos (*Figure 9D$_3$*) were well organized and mostly parallel, axonal behavior in mutants caused 'smeared' patterns (asterisks *Figure 9E$_3$*) due to stalling and pixels moving obviously in caudal direction indicating caudal axonal turns (arrowheads in *Figure 9F$_3$*). These phenotypes confirmed our analyses of open-book preparations of spinal cords lacking Endoglycan (*Figure 2*). Axonal stalling (arrowheads) and caudal turns (arrows) could also be observed at the floor-plate exit site of spinal cord overexpressing *Endoglycan* (*Video 4*).

The obvious differences in axonal behavior in experimental compared to control spinal cords was corroborated by quantitative analyses of specific aspects. Firstly, we quantified how much time the

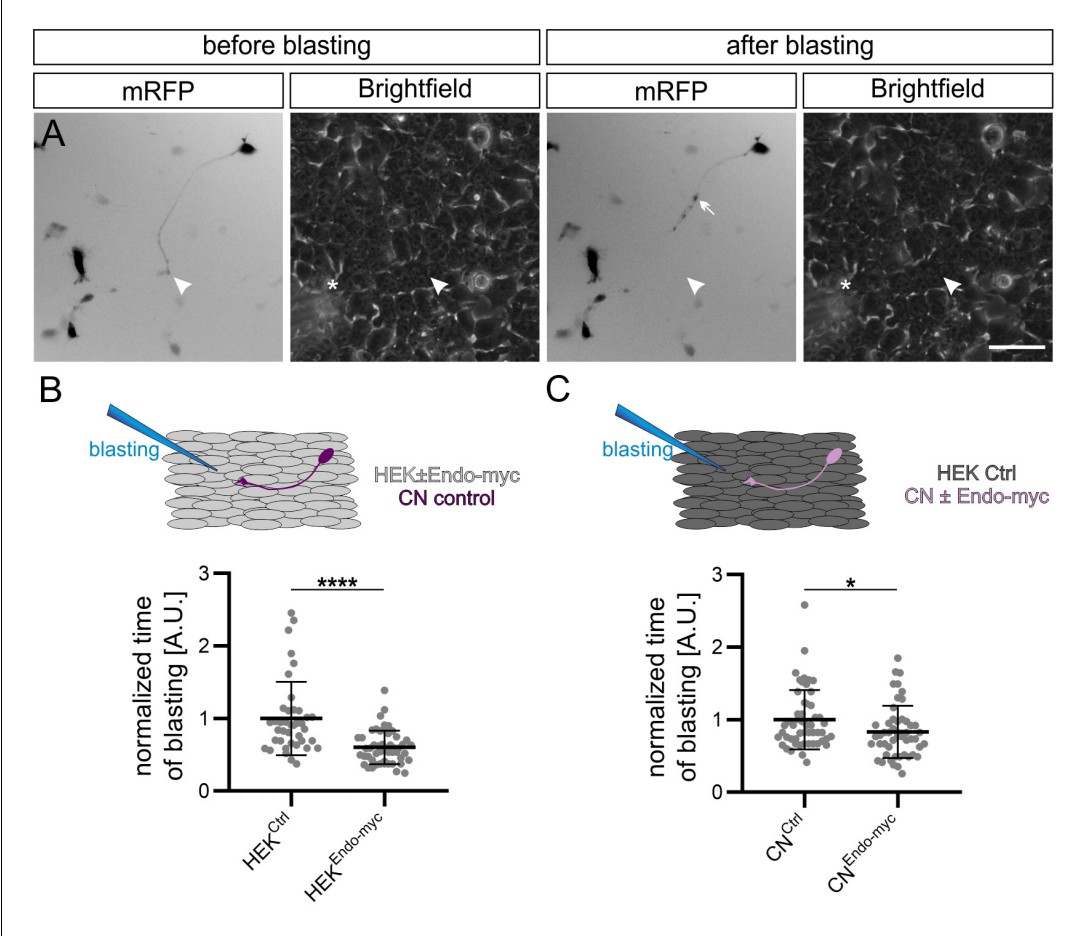

**Figure 6.** Endoglycan reduces the adhesive strength of commissural neuron growth cones. (**A**) Example of snapshots taken before and after growth cone blasting of mRFP-positive commissural neurons cultured on a HEK cell layer (shown in the brightfield channel). White arrowheads and asterisks show the location of the growth cone and the approximate location of the micropipette tip, respectively. (**B**) Control mRFP-transfected commissural neurons were plated either on a layer of control HEK cells or HEK cells expressing Endoglycan. The time it took to detach a growth cone from the HEK cells was taken as a measure for the adhesive strength. The adhesion of growth cones to HEK cells expressing Endoglycan was significantly reduced (0.59 ± 0.26 compared to 1 ± 0.61 on control HEK cells). N(replicates)=3, n(growth cones)=40 (HEK[Ctrl]) and 50 (HEK[Endo-myc]). (**C**) Similar observations were made when commissural neurons were transfected with Endoglycan instead of the HEK cells. Detachment was faster for growth cones of mRFP-transfected neurons co-transfected with Endoglycan and plated on control HEK cells (0.83 ± 0.33) compared to control neurons transfected only with mRFP (1 ± 0.41). N(replicates)=6, n(growth cones)=52 (CN[Ctrl]) and 51 (CN[Endo-myc]). *p<0.05, ****p<0.0001, Two tailed Mann-Whitney test. Error bars represent standard deviation. CN, commissural neuron; Ctrl, control. Scale bar: 50 μm. Source data and statistics are available in **Figure 6—source data 1** spreadsheet.

The online version of this article includes the following source data for figure 6:

**Source data 1.** Raw data of growth cone blasting experiments.

growth cones spent migrating from the floor-plate entry site to the exit site (**Figure 10A₁**). Confirming the observations made in our videos and kymographic analysis, growth cones overexpressing *Endoglycan* crossed the floor plate faster, in only 4.4 ± 1.4 hr (mean ± SD), compared to controls (5.4 ± 1.3 hr) and the dsEndo condition (5.5 ± 1.2; **Figure 10B**). Furthermore, we compared the average time for crossing each half of the floor plate for each condition (**Figure 10A₁, C**). Growth cones migrated equally fast through both halves in controls (2.7 ± 0.9 hr versus 2.6 ± 1.0 hr, **Figure 10C**). After overexpression of *Endoglycan*, there was a slight but significant shortening of the time growth cones spent crossing the first half compared to the second half (2.1 ± 0.7 hr versus 2.3 ± 0.8 hr, **Figure 10C**). In contrast, the unilateral silencing of *Endoglycan* induced a highly significant difference in the migration speed of growth cones in the first (electroporated) versus the second half of the floor plate. It took 3.1 ± 0.9 hr to cross the first half but only 2.5 ± 1.0 hr to cross the second half

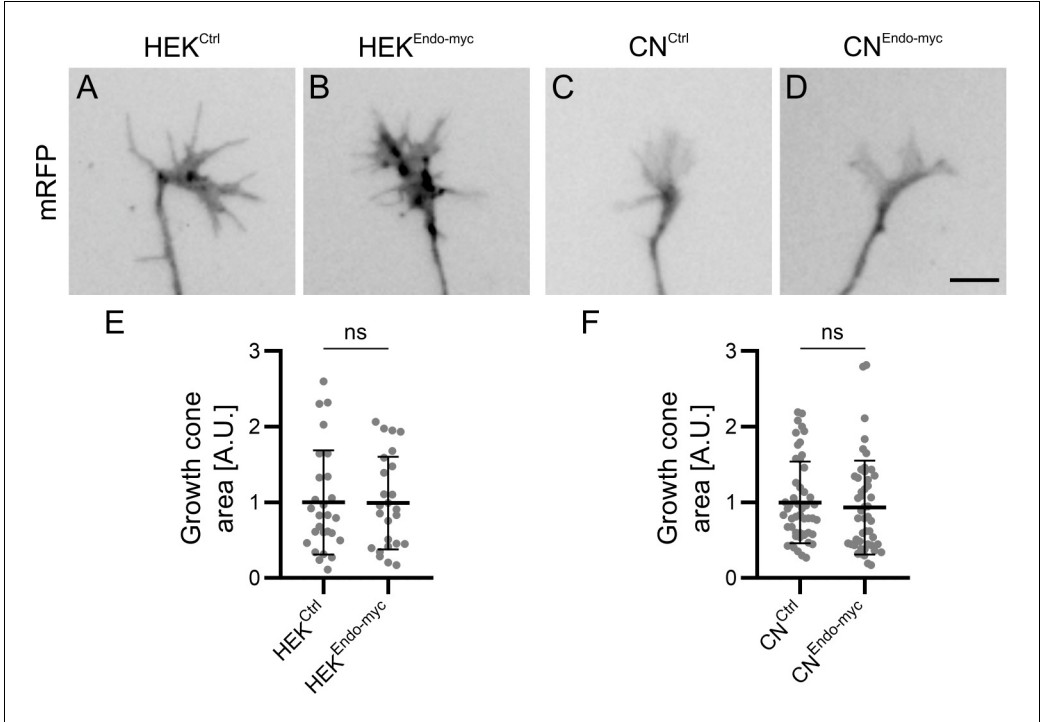

**Figure 7.** Reduced attachment of growth cones in the presence of higher levels of Endoglycan cannot be explained by changes in growth cone area. To exclude that the observed differences in adhesive strength between growth cones and HEK cells were influenced by growth cone size, we measured growth cone areas by tracing the edge of the mRFP-positive growth cones from images taken before blasting with a x20 objective. Note that the numbers of growth cones in the growth cone blasting experiment (*Figure 6*) and the area measurements shown here differ because in two replicates in ~50% of the measured growth cones only videos were taken but no still images. Therefore, these growth cones are not included in the size measurement. The area of each growth cone was measured in imageJ using the tracing tool and the value was normalized to the average growth cone area of the control condition (either HEK$^{Ctrl}$ or CN$^{Ctrl}$) for each replicate. A.U., arbitrary unit. Values were 1 ± 0.69 for HEK$^{Ctrl}$ (**A**) versus 0.99 ± 0.61 for HEK$^{Endo-myc}$ (**B**), and 1 ± 0.54 for CN$^{Ctrl}$ (**C**) versus 0.93 ± 0.62 for CN$^{Endo-myc}$ (**D**; mean ± standard deviation). Statistical analysis: **E**, p=0.9694 (ns, unpaired T-test), N (replicates) = 3; n (growth cones)=27 (HEK$^{Ctrl}$) and 25 (HEK$^{Endo-myc}$); **F**, p=0.3261 (ns, Mann Whitney test), N (replicates) = 6; n (growth cones)=50 (CN$^{Ctrl}$) and 49 (CN$^{Endo-myc}$). Bar: 10 µm. Source data and statistics are available in *Figure 7—source data 1* spreadsheet. The online version of this article includes the following source data for figure 7:

**Source data 1.** Raw data of growth cone area measurements.

(*Figure 10C*). An alternative way of demonstrating the differences in migration speed is shown in *Figure 10D,E*. We calculated the ratios of the time spent in the first (*Figure 10D*) or the second half of the floor plate (*Figure 10E*) compared to the total time used for floor-plate crossing for the different conditions (*Figure 10B*). Indeed, knockdown of Endoglycan induced a significant increase in the ratio spent in the first half of the floor plate (0.56 ± 0.1) compared to both control (0.51 ± 0.1) and overexpression of Endoglycan (0.48 ± 0.1; *Figure 10D*). In the second half of the floor plate, there was a significant decrease in spinal cords electroporated with dsEndo (0.44 ± 0.1) compared to control (0.49 ± 0.1) and *Endoglycan*-overexpressing spinal cords (0.52 ± 0.1; *Figure 10E*).

Secondly, we also analyzed growth cone morphologies by comparing the average area in the first and the second halves, as well as at the exit site of the floor plate (*Figure 10A₂*). The difference in growth speed was reflected in growth cone morphology and size (*Figure 11* and *Video 2*). Growth cones tended to be small and have a simple morphology at fast speed. At choice points, like the floor-plate exit site, growth cone size and complexity increased. The average area of growth cones in the floor plate in control spinal cords was 50 ± 13 µm² in the first, and 47 ± 12 µm² in the second half of the floor plate (*Figure 11A*). The growth cone area significantly increased at the floor-plate exit site, where growth cones need to choose to grow rostrally rather than caudally (*Figure 11A*). At the exit site, growth cone area was 106 ± 29 µm² in control embryos. In agreement with their faster speed in the floor plate overexpressing *Endoglycan*, growth cones were significantly smaller in the first (43.5 ± 15 µm²) and the second half (41 ± 11 µm²) of the floor plate, as well as at the exit site

($88 \pm 28$ μm$^2$) compared to controls (*Figure 10F–H* and *Figure 11C*). Reduction of Endoglycan expression in the axons and in the first half of the floor plate resulted in reduced migration speed, but the average size of the growth cones was not significantly different from controls. (*Figure 10F*). The fact that growth cone were significantly faster in the second, non-transfected half of the floor plate was reflected by a significant reduction in growth cone area ($41 \pm 10$ μm$^2$) compared to control ($49 \pm 13$ μm$^2$) and compared to the first, transfected half ($49 \pm 13$ μm$^2$; *Figure 10G*, *Figure 11B*, *Table 2*).

Finally, live imaging of growth cones crossing the floor plate provided support for our hypothesis that axon-floor plate contact was causing the displacement of floor-plate cells observed after knock-down of *Endoglycan*, the 'corkscrew' phenotype. The tortuous, 'corkscrew'-like phenotype of axons was seen exclusively in spinal cords after knockdown of *Endoglycan* (*Figure 12* and *Videos 5* and *6*). We could observe roundish EGFP-f-positive cells (arrows) that obstructed the smooth trajectory of dI1 axons in the commissure (arrowheads in *Figure 12A$_{1-5}$* and *Video 5*). Although we could not use markers to identify these cells as floor-plate cells, their position indicated that they had to be mislocalized floor-plate cells. Moreover, dI1 axons were found to form loops in the layer where floor-plate cell somata were localized (arrowhead in *Figure 12B$_{1-5}$* and *Video 6*). In the first half of the floor plate electroporated with dsEndo, clusters of roundish EGFP-f-positive cells in the commissure (arrows) were apparently causing axons to deviate from their trajectory by strongly adhering to them (arrowheads in *Figure 12C$_{1-5}$* and *Video 6*). We never found such aberrant behavior in control embryos or in embryos overexpressing *Endoglycan*. Moreover, we only observed these events after many dI1 axons had already crossed the floor plate (after at least 10 hr), supporting the hypothesis that the phenotype was due to excessive growth cone-floor plate adhesion resulting in floor-plate cell displacement. Furthermore, these observations suggest that Endoglycan regulates migratory speed of growth cones by modulating their adhesion to floor-plate cells.

Taken together, our live imaging studies support results from in vitro adhesion experiments indicating that the level of Endoglycan expression modulates adhesive strength between dI1 commissural growth cones and floor-plate cells. In contrast, axon-axon interactions did not seem to be different in the presence and absence of Endoglycan, as we did not find any effect on pre-crossing axons after perturbation of Endoglycan levels (*Figure 13*).

In summary, our results demonstrate a vital role for Endoglycan in commissural axon guidance at the ventral midline. The observed phenotype is consistent with the hypothesis that Endoglycan is an essential regulator of cell-cell contacts by modulating the strength of adhesion between commissural axon growth cones and floor-plate cells. This model is supported by observations in vitro and in vivo. Neuronal attachment was negatively affected by the presence of an excess of Endoglycan in a glycosylation-dependent manner, indicating that Endoglycan decreases adhesive strength during neural circuit assembly.

## Discussion

We identified *Endoglycan*, a member of the CD34 family of sialomucins, in a screen for axon guidance cues involved in commissural axon pathfinding at the midline of the spinal cord. The phenotypes obtained in our in vivo experiments and in live imaging observations in intact spinal cords are consistent with an anti-adhesive role of Endoglycan. This is further supported by the reduced adhesion of growth cones in in vitro assays. Thus, Endoglycan may act like a

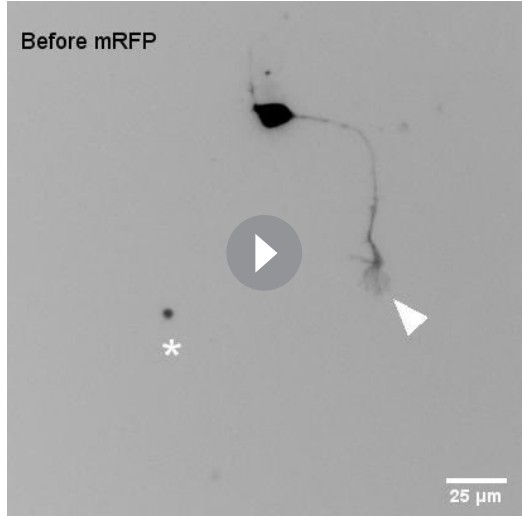

**Video 1.** Example of growth cone blasting assay. Snapshots taken from a video showing the detachment of an mRFP-positive commissural neuron growth cone (shown in black, indicated by white arrowhead) on a layer of HEK cells (bright-field view) before and after detachment. The position of the tip of the glass micropipette is indicated by asterisk.
https://elifesciences.org/articles/64767#video1

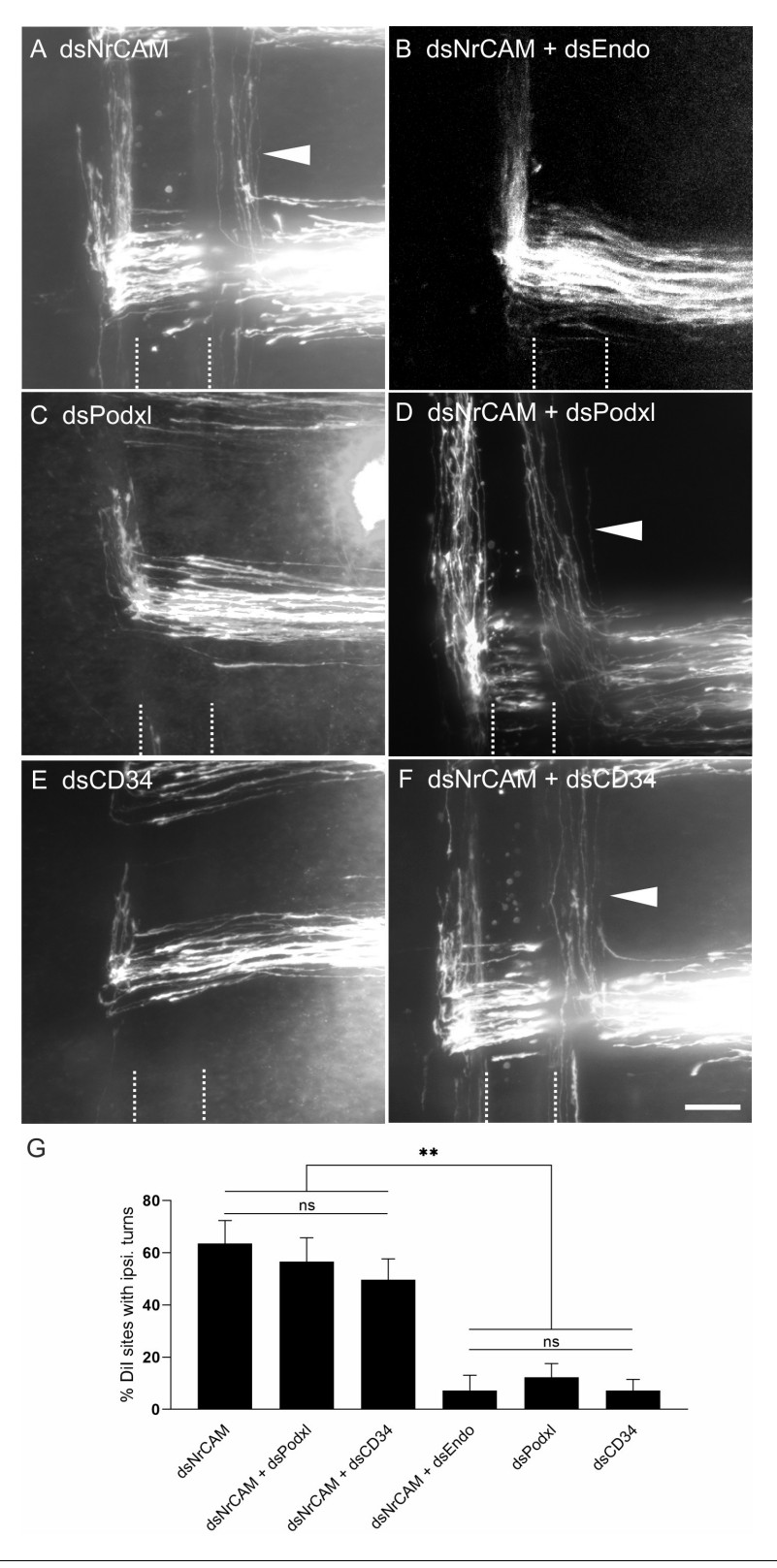

**Figure 8.** Downregulation of Endoglycan, but not its family members, rescues the axon guidance phenotype induced by downregulation of NrCAM. The perturbation of axon/floor-plate contact by downregulation of NrCAM resulted in the failure of commissural axons to enter the floor-plate area and caused their premature turns along the ipsilateral floor-plate border (arrowhead in **A**) at 63.6 ± 8.8% of the injection sites (n = 7 embryos, 90 injection sites; **G**). These results are in line with previous reports (*Stoeckli and Landmesser, 1995*; *Philipp et al., 2012*). When both NrCAM and Endoglycan

*Figure 8 continued on next page*

*Figure 8 continued*

were downregulated, the number of ipsilateral turns was reduced to control levels (B,G; 7.3 ± 5.8%, n = 10 embryos; 78 injection sites), consistent with the idea that a decrease in adhesion due to a lack of NrCAM can be balanced by an increase in adhesion between floor plate and growth cones due to a lack of Endoglycan. Downregulation of either Podocalyxin (C) or CD34 (E) did not impair axon guidance (see also *Figure 2*; 12.3 ± 5.2% (n = 9) and 7.25 ± 4.3% (n = 8), respectively). In contrast to Endoglycan, neither concomitant downregulation of Podocalyxin (D) nor CD34 (F) could rescue the NrCAM-induced ipsilateral turns, as aberrant axon behavior was still observed at 56.7 ± 9.2% (n = 8 embryos, 77 injection sites) and 49.6 ± 8.1% (n = 10 embryos, 100 injection sites), respectively. The floor plate is indicated by dashed lines. For statistical analysis, one-way ANOVA followed by Tukey's multiple comparisons test was used, **p<0.01 or lower, see source data and statistics in *Figure 8—source data 1* spreadsheet; (ns) p≥0.05. Bar: 50 µm. The online version of this article includes the following source data for figure 8:

**Source data 1.** Quantification of ipsilateral turns - raw data.

'lubricant' modulating the motility of growth cones in the floor plate by lowering stickiness. Such a function is supported by the structural features of sialomucins. The function of CD34 family members has not been characterized in detail, but all the results obtained so far are compatible with an anti-adhesive role (*Nielsen and McNagny, 2008*). One exception are reports from lymph node cells, the so-called high endothelial venules (HEVs), where a very specific glycosylation patterns was implicated in the interaction of CD34 and Endoglycan with L-selectin (*Furness and McNagny, 2006*). However, in agreement with most published studies on the role of CD34 and Podocalyxin (for reviews see *Furness and McNagny, 2006*; *Nielsen and McNagny, 2008* and *Nielsen and McNagny, 2009*), our observations suggest that Endoglycan acts as an anti-adhesive rather than as adhesive factor. This model is supported by results from in vivo and in vitro experiments that confirm a negative effect of Endoglycan on cell-cell adhesion (*Figure 14*).

The adhesion-modulating effect of Endoglycan is mediated by the negatively charged mucin domain. Similar to the role suggested for the polysialic acid modification of NCAM (*Rutishauser, 2008*; *Brusés and Rutishauser, 2001*; *Burgess et al., 2008*), Endoglycan could lower cell-cell adhesion by increasing the distance between adjacent cell membranes due to repulsion caused by the bulky, negatively charged posttranslational modifications of its extracellular domains. A similar effect was found for PSA-NCAM in hindlimb innervation (*Tang et al., 1994*; *Landmesser et al., 1990*) and in the visual system, where retinal ganglion cell axons innervating the tectum were found to regulate axon-axon adhesion versus axon-target cell adhesion (*Rutishauser et al., 1988*). The same mechanism was found in motoneurons, where axon-axon versus axon-muscle fiber adhesion was a determining factor for the appropriate innervation pattern. In contrast to PSA-NCAM that continues to play a role in synaptic plasticity in the adult nervous system, the function of Endoglycan appears to be restricted to development. Our findings together with structural features of Endoglycan suggest that the mechanisms by which Endoglycan reduces adhesion of growth cones and floor-plate cells might be similar. Additional studies will be required to verify whether the effect of Endoglycan as modulator of adhesive strength uses the exact same mechanism as PSA-NCAM.

At first sight, the effect of Endoglycan on floor-plate morphology appears to suggest a positive regulation of cell-cell adhesion. Floor-plate cells are precisely aligned in control embryos but are protruding into the commissure in the absence of Endoglycan. Therefore, one might conclude that in the absence of Endoglycan cell-cell adhesion between floor-plate cells is compromised, resulting in the observed structural changes. However, this scenario can be ruled out based on the analysis of younger embryos. At HH21, the floor plate was intact in the absence of Endoglycan, indicating that Endoglycan is not required for adhesion between floor-plate cells (*Figure 3*). The morphology of the floor plate is only compromised once many axons have crossed the midline (*Figure 3G–L*). These findings are supported by our live imaging data of growth cones crossing the floor plate (*Figure 12*, *Videos 5* and *6*). Contacts between commissural axons and floor-plate cells have to be broken when later crossing commissural axons arrive and cross (*Yaginuma et al., 1991*). Commissural axons crossing the floor plate are suggested to do so by close interaction with short filopodial processes of floor-plate cells (*Dumoulin et al., 2021*). Thus, the aberrant morphology of the floor plate at HH25 could be explained by the inability of axons to break contacts with floor-plate cells in the absence of Endoglycan, consistent with our hypothesis that Endoglycan is a negative regulator of adhesion. Further support for this hypothesis was contributed by in vitro findings that the adhesive strength between growth cones of commissural neurons and HEK cells was reduced when Endoglycan was expressed either in the HEK cells or in the neurons (*Figure 6*). Live imaging data demonstrated that

the perturbation of the balance in growth cone-floor plate adhesion led to impaired timing of midline crossing, which in turn might also interfere with the correct sensing and reading of guidance cues by dI1 growth cones, and prevented them from making the correct decision at the floor-plate exit site (*Figures 9* and *10*).

Thus, we concluded that the function of Endoglycan in commissural axon guidance is to lower cell adhesion: the absence of Endoglycan results in too much stickiness. At the midline of the spinal cord, excessive adhesion causes axons to adhere too much to floor-plate cells and prevents their displacement by follower axons. Rather than acting as a guidance cue or guidance receptor, we suggest that Endoglycan affects neural circuit formation by modulating the interaction of many different guidance cues and their surface receptors.

In summary, we propose an 'anti-adhesive' role for Endoglycan in axon guidance that is fine-tuning the balance between adhesion and de-adhesion (*Figure 14*). Precise regulation of cell-cell contacts is required in both processes and is fundamental for developmental processes that depend on a high degree of plasticity and a plethora of specific molecular interactions.

# Materials and methods

## Key resources table

| Reagent type (species) or resource | Designation | Source or reference | Identifiers | Additional information |
|---|---|---|---|---|
| Antibody | Anti-digoxigenin-AP antibody (rabbit polyclonal) | Roche | RRID:AB_514497 | ISH (1:10000) |
| Antibody | Anti-myc (rabbit polyclonal) | Abcam | RRID: AB_307014 | IF (1:500) |
| Antibody | Anti-Hnf3β (mouse monoclonal) | DSHB | RRID: AB_2278498 | IF (supernatant) |
| Antibody | Anti-Axonin-1 (goat polyclonal) | *Stoeckli and Landmesser, 1995* | N/A | IF (1:500) |
| Antibody | Anti-Axonin-1 (rabbit polyclonal) | *Stoeckli and Landmesser, 1995* | N/A | IF (1:1000) |
| Antibody | Anti-neurofilament-M (RMO-270, mouse monoclonal) | ThermoFisher Scientific | RRID: AB_2532998 | IF (1:200) |
| Antibody | Anti-GFP-FITC (goat polyclonal) | Rockland | RRID: AB_218187 | IF (1:400) |
| Antibody | Anti-NrCAM (mouse monoclonal) | *Fitzli et al., 2000* | N/A | IF (1:1000) |
| Antibody | Anti-Islet1 (mouse monoclonal) | DSHB | RRID: AB_528315 | IF (supernatant) |
| Antibody | Anti-Nkx2.2 (mouse monoclonal) | DSHB | RRID: AB_531794 | IF (supernatant) |
| Antibody | Anti-Pax3 (mouse monoclonal) | DSHB | RRID: AB_528426 | IF (supernatant) |
| Antibody | Anti-Pax6 (mouse monoclonal) | DSHB | RRID: AB_528427 | IF (supernatant) |

*Continued on next page*

*Continued*

| Reagent type (species) or resource | Designation | Source or reference | Identifiers | Additional information |
|---|---|---|---|---|
| Antibody | Cleaved Caspase-3 (rabbit polyclonal) | Cell Signaling | RRID: AB_2341188 | IF (1:200) |
| Cell line (*H. sapiens*) | HEK293T | American Type Culture Collection | RRID: CVCL_0063 | |
| Cell line (*H. sapiens*) | HEK293T-PODXL2-myc (Endo-myc) | This paper | N/A | |
| Recombinant DNA reagent | Math1::chEndoglycan (plasmid) | This paper | N/A | |
| Recombinant DNA reagent | Hoxa1::chEndoglycan (plasmid) | This paper | N/A | |
| Recombinant DNA reagent | β-actin::chEndoglycan (plasmid) | This paper | N/A | |
| Recombinant DNA reagent | β-actin::EGFP-F (plasmid) | This paper | N/A | |
| Recombinant DNA reagent | β-actin::mRFP (plasmid) | *Wilson and Stoeckli, 2011* | N/A | |
| Recombinant DNA reagent | Math1::tdTomato-F (plasmid) | *Wilson and Stoeckli, 2011* | N/A | |
| Chemical compound, drug | O-glycosidase and Neuraminidase | NEB | Cat# E0540S | |
| Software, algorithm | Fiji/ImageJ | *Schindelin et al., 2012* | RRID:SCR_002285 | |
| Biological sample (Gallus gallus) | Hubbard JA57 strain | Brüterei Stöckli, Ohmstal | N/A | |

## Identification and cloning of *Endoglycan*

We had used a PCR-based subtractive hybridization screen to search for guidance cues for post-crossing commissural axons (for details, see *Bourikas et al., 2005*). To this end, we isolated floor-plate cells from HH26 and HH20 embryos (*Hamburger and Hamilton, 1951*). Among the differentially expressed genes, we found a sequence from the 3'-UTR of *Endoglycan*. Subsequently, a cDNA fragment from the coding sequence of *Endoglycan* (PODXL2; 1028–1546 bp) was obtained by RT-PCR using total RNA isolated from HH40 cerebellum. For reverse transcription, 1 μg total RNA was mixed with 0.3 μl RNasin (Promega), 1 μl dNTPs (5 mM), 1 μl random nonamers, 1 μl DTT (Promega), in 20 μl Superscript II buffer (Invitrogen). Reverse transcription was carried out for 1 hr at 42°C. Two μl of this mixture were used for PCR with 2.5 μl forward primer (10 μM; 5'-CAGACACGCAGACTCTTTC-3') and 2.5 μl reverse primer (10 μM; 5'-CTAAAGATGTGTGTCTTCCTCA-3') using the Expand Long Template PCR System (Roche). The PCR conditions were 35 cycles at 95°C for 30 s, 57°C for 30 s and 68°C for 3 min. The PCR product was cut with BamHI/BclI and cloned into pBluescript II KS. For cloning of full-length chicken Endoglycan, we used 5'-ATGGTGAGAGGAGCTGCG-3' and 5'-GTGTTTGAGGAAGACACACATCTTTAG-3' as forward and reverse primers, respectively. A plasmid containing the full-length ORF of human *Endoglycan* was obtained from SourceBioScience.

## Preparation of DIG-labeled RNA probes and in situ hybridization

For in vitro transcription, 1 μg of the linearized and purified plasmids encoding Endoglycan (EndoORF: 1028-1546pb, Endo3'UTR: 3150–3743 bp and 5070–5754 bp; numbers are derived from

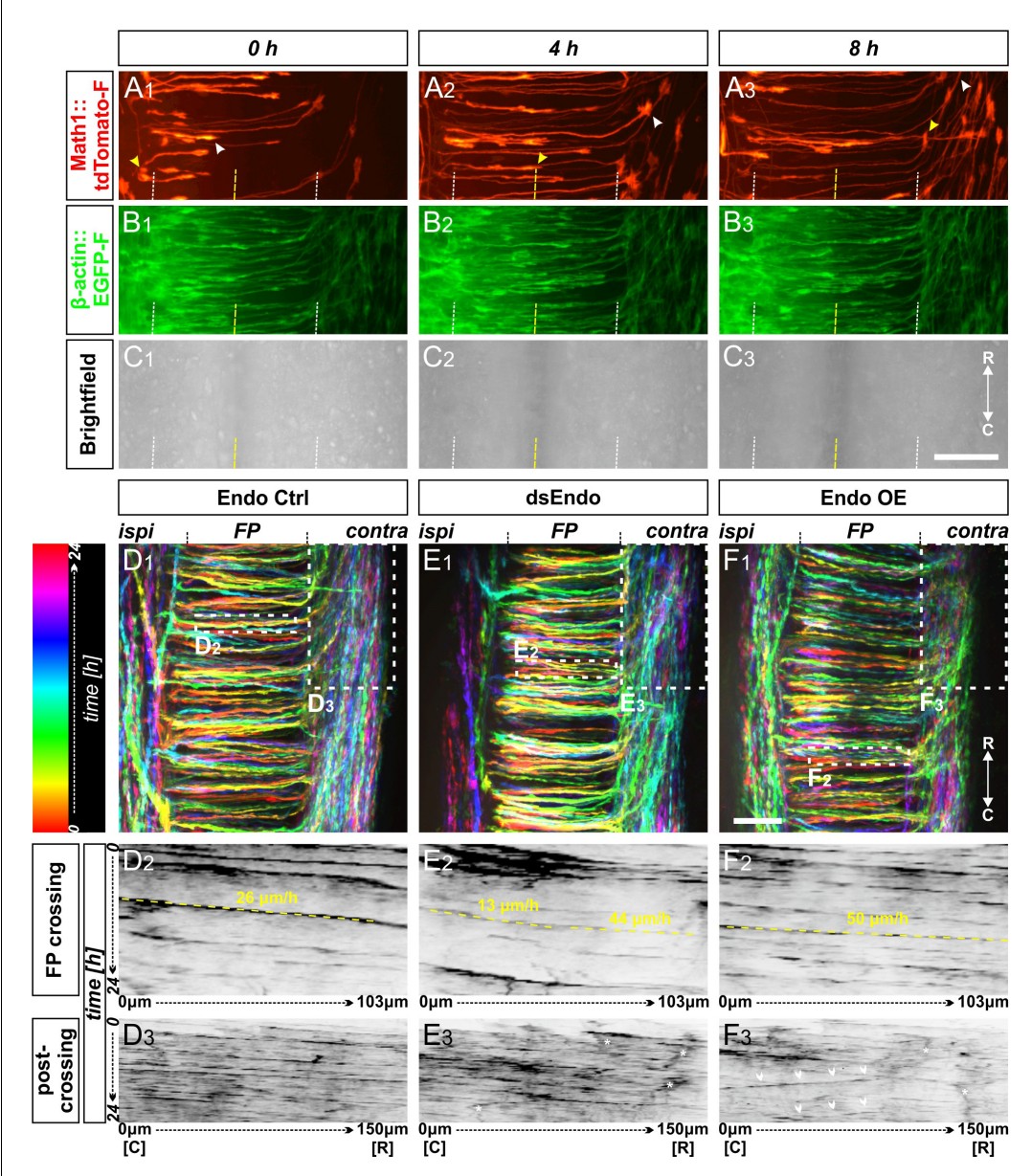

**Figure 9.** Live imaging of cultured intact spinal cords revealed major impacts of different Endoglycan levels on midline crossing. (A-C) Live imaging allowed tracing and quantitative analysis of dI1 axons' trajectories in cultured intact chicken spinal cords. (A$_{1-3}$) The behavior and trajectory of single tdTomato-positive dI1 axons could be tracked over time when they crossed the floor plate and turned rostrally (yellow and white arrowheads). (B-C) EGFP-F expression under the β-actin promoter and brightfield images helped to visualize the floor-plate boundaries (white dashed line) and the midline (yellow dashed line). (D$_1$, E$_1$, F$_1$) Temporally color-coded projections of 24 hr time-lapse movies (*Video 2*). Kymograph analysis of the regions of interest selected in the floor plate of each condition shown in (D$_1$,E$_1$,F$_1$) was used to calculate growth cone speed during floor-plate crossing (D$_2$, E$_2$, F$_2$) and after turning into the longitudinal axis (D$_3$, E$_3$, F$_3$). Yellow dashed lines outline a representative example of the slope (velocity) of a single axon crossing the floor plate in each condition. TdTomato-positive axons in control-injected spinal cords (D$_{1-3}$) crossed the floor plate at a steady speed of 26 μm/h (D$_2$) and turned rostrally in a highly organized manner (D$_3$). In contrast, growth cone speed in the first half of the floor plate that was electroporated with dsRNA derived from Endoglycan (dsEndo) was markedly slowed down to only 13 μm/hr. In the second, non-electroporated half of the floor plate, axons electroporated with dsEndo were faster than control axons (44 μm/hr; E$_2$). Axons overexpressing Endoglycan were faster in both halves of the floor plate (50 μm/hr; F$_2$). Downregulation or overexpression of Endoglycan clearly impacted the rostral turning behavior visualized by less organized patterns (D$_3$-F$_3$). Asterisks mark axons stalling and thus causing a 'smeared' pattern in the kymographs. Arrowheads indicate caudally turning axons. R, rostral; C, caudal; ipsi, ipsilateral; contra, contralateral; FP, floor plate. Scale bars: 50 μm.

the human sequence), Podocalyxin (ChEST190L9), and CD34 (ChEST91D7) were used to prepare DIG-labeled in situ probes as described earlier (*Mauti et al., 2006*). The same fragments were used to prepare dsRNA (*Pekarik et al., 2003*; *Baeriswyl et al., 2008*; *Andermatt and Stoeckli, 2014b*).

## Northern blot

Total RNA was extracted from cerebrum, cerebellum, spinal cord, muscle, heart, lung, and kidney from HH38 embryos using the RNeasy Mini Kit (Qiagen) and loaded on a denaturing formaldehyde gel (4.5 µg of total RNA per lane). The RNA was blotted onto a positively charged nylon membrane (Roche) overnight, using 10x SSC as a transfer medium. The membranes were hybridized with 1.5 µg preheated DIG-labeled RNA probes for *Endoglycan* and *GAPDH* at 68°C overnight. The membrane was then washed twice with 2xSSC/0.1%SDS for 5 min at room temperature and twice with 0.1xSSC/0.5% SDS for 20 min at 68°C. For detection, buffer 2 (2% blocking reagent dissolved in 0.1 M maleic acid, 0.15 M NaCl, pH 7.5) was added for 2–3 hr at room temperature. After incubation with anti-digoxigenin-AP antibody dissolved in buffer 2 (1:10,000; Roche, RRID:AB_514497) for 30 min at room temperature the membrane was washed twice in washing buffer (0.3% Tween 20 dissolved in 0.1 M maleic acid, 0.15 M NaCl, pH 7.5) for 20 min. Subsequently, detection buffer (0.1 M Tris-HCl, 0.1 M NaCl, pH 9.5) was applied for 2 min before adding CDP-star (25 mM, cat# C0712, Roche) for 5 min in the dark. For detection of the chemiluminescence a Kodak BioMAX XAR film was used.

## In ovo RNAi

For functional studies in the spinal cord, we silenced *Endoglycan* with three different long dsRNAs. They were produced from bp 1028–1546 of the ORF, as well as bp 3150–3743 and bp 5070–5754 from the 3'UTR. The fact that we obtained the same phenotype with three different, non-overlapping dsRNAs derived from *Endoglycan* confirms the specificity of the approach and the absence of off-target effects. dsRNA was produced as detailed in *Pekarik et al., 2003* and *Wilson and Stoeckli, 2011*. Because no antibodies recognizing chicken Endoglycan are available, we used in situ hybrid-

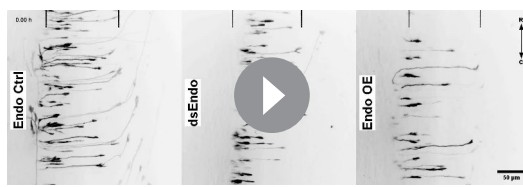

**Video 2.** 24 hr time-lapse recordings of tdTomato-positive dI1 axons (shown in black) in control (Endo control), Endoglycan knockdown (dsEndo), and Endoglycan overexpression (Endo OE) conditions. Dashed lines represent floor plate boundaries. R, rostral; C, caudal.
https://elifesciences.org/articles/64767#video2

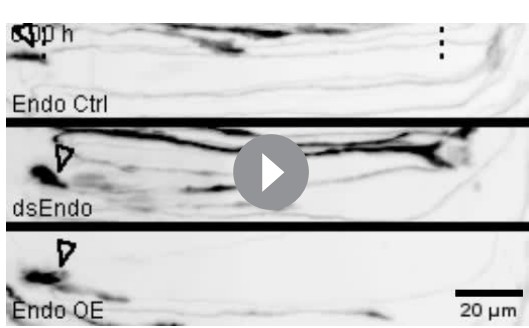

**Video 3.** Representative examples of the floor plate crossing of tdTomato-positive dI1 axons (shown in black) taken from 24 hr time-lapse recordings from control (Endo control), Endoglycan knockdown (dsEndo), and Endoglycan overexpression (Endo OE) conditions. An arrowhead in each condition points at the migrating growth cone. Dashed lines represent floor-plate boundaries.
https://elifesciences.org/articles/64767#video3

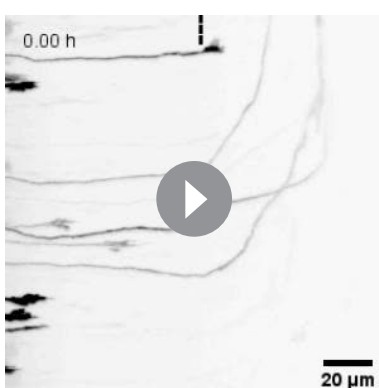

**Video 4.** Example of a 24 hr time-lapse recordings of tdTomato-positive dI1 axons (shown in black) showing guidance defects at the exit site of the floor plate in Endoglycan overexpression condition. Stalling growth cones are shown by arrowheads and caudally turning growth cones by arrows. Rostral is up. Dashed line represents the floor-plate exit site.
https://elifesciences.org/articles/64767#video4

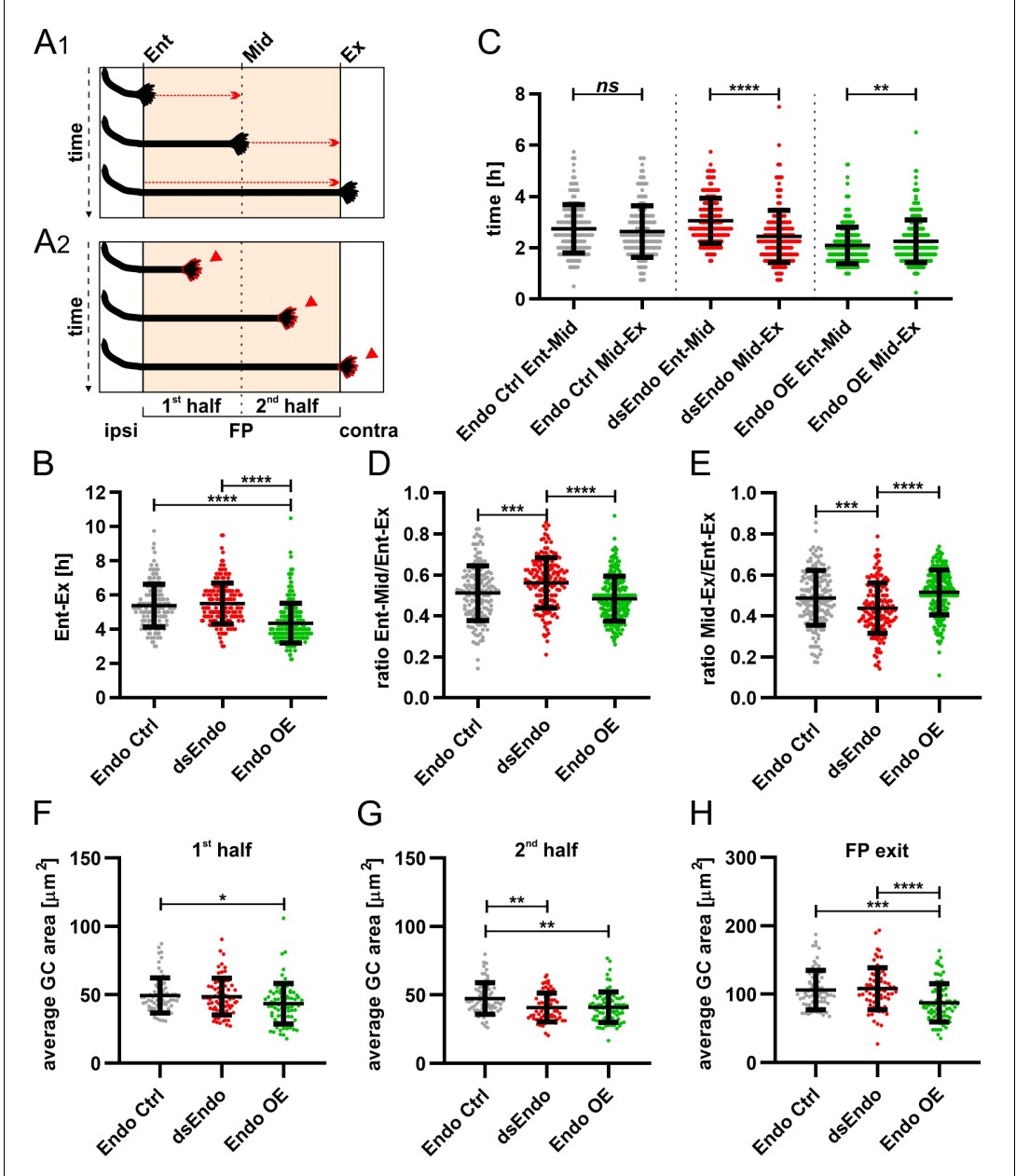

**Figure 10.** Too much or too little of Endoglycan impaired the timing and morphology of single dI1 growth cones migrating in the floor plate. Data at the single axon level extracted from 24 hr time-lapse recordings of tdTomato-positive dI1 axons crossing the floor plate. ($A_1$) The time of floor-plate crossing was measured for the entire floor plate, for the first and for the second half for each condition. ($A_2$) The average growth cone area was measured in the first half, the second half and at the exit site of the floor plate for each condition. (B) Overexpression of Endoglycan significantly decreased the time axons needed to cross the entire floor plate compared to control and dsEndo conditions (Kruskal-Wallis test with Dunn's multiple-comparisons test). (C) The average time of crossing the first half and the second half of the floor plate was compared. Interestingly, there was a highly significant difference in the spinal cords unilaterally electroporated with dsEndo, as axons spent much longer in the first compared to the second half of the floor plate. There was no difference between the two halves of the floor plate in the control condition, but there was a significant decrease in growth cone speed between the first (electroporated) half of the floor plate and the second half, where only axons were overexpressing Endoglycan (Wilcoxon test). (D) and (E) The ratios of the time axons spent in the first half (D) or the second half (E) of the floor plate divided by the time they needed to cross it entirely were compared between conditions. Unilateral knockdown of Endoglycan resulted in a significant increase of the ratio in the first half and a decrease in the second half compared to both control and overexpression conditions (one-way ANOVA with Sidak's multiple-comparisons test). (F-H) The average dI1 growth cone area at each position of the floor plate (as depicted in $A_2$) was compared across all conditions. (F) Overexpression of Endoglycan induced a significant reduction in the average growth cone area compared to the control condition (Kruskal-Wallis test with Dunn's multiple-comparisons test) but not compared to Endoglycan knockdown (p value = 0.08). (G) In the second half of the floor plate, the

*Figure 10 continued on next page*

*Figure 10 continued*

average growth cone area was reduced in both Endoglycan knockdown and overexpression condition compared to control (one-way ANOVA with Sidak's multiple-comparisons test). (H) At the floor-plate exit site, overexpression of Endoglycan induced a significant decrease in the average growth cone area compared to both control and knockdown conditions (one-way ANOVA with Sidak's multiple-comparisons test). Error bars represent standard deviation. $p < 0.0001$ (****), $p < 0.001$ (***), $p < 0.01$ (**), $p < 0.05$ (*), and $p \geq 0.05$ (ns) for all tests. N(embryos)=3 for each condition; n(axons, panel C-E)=161 (Endo Ctrl), 168 (dsEndo), 234 (Endo OE); n(axons, panel F-H)=70 (Endo Ctrl), 78 (dsEndo), 83 (Endo OE). See *Table 2* for detailed results. Ent, entry; Mid, midline; Ex, exit; ipsi, ipsilateral; contra, contralateral; FP, floor plate; GC, growth cone. Source data and statistics are available in *Figure 10—source data 1* spreadsheet.

The online version of this article includes the following source data for figure 10:

**Source data 1.** Raw data of live imaging experiments.

---

ization to assess the successful downregulation of the target mRNA (*Figure 2—figure supplement 3*). Downregulation efficiency was about 40%. Because we transfect only around 50% of the cells in the electroporated area, transfected cells express only very low levels of Endoglycan.

For rescue experiments, the dsRNA was co-injected with 150 (low), 300 (middle), or 750 ng/μl (high) plasmid encoding the ORF of chicken *Endoglycan*. The ORF was either expressed under the control of the Math1 promoter for dI1 neuron-specific expression, or the Hoxa1 promoter for floor-plate specific expression of *Endoglycan*.

## Commissural neuron and motoneuron adhesion assay

Dissociated commissural or motoneurons of HH25/26 chicken embryos were cultured as described previously (*Avilés and Stoeckli, 2016*; *Mauti et al., 2006*) either on HEK293T cells stably expressing human Endoglycan-myc under the control of the CMV promoter or on untransfected HEK293T (ATTC, Cat# CRL-3216, RRID:CVCL_063) cells as control. The plasmid encoding human Endoglycan

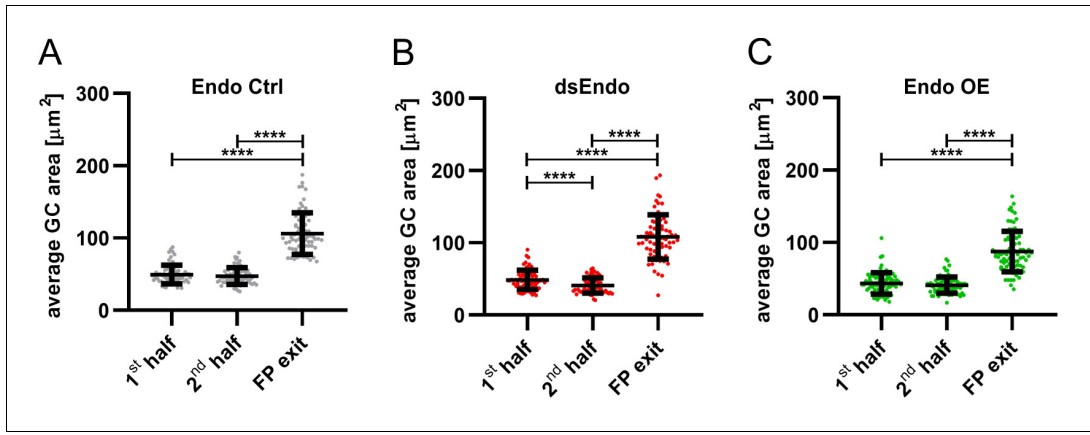

**Figure 11.** Growth cone size is enlarged at the floor-plate exit site. Data at the single axon level extracted from 24 hr time-lapse recordings of dI1 axons crossing the floor plate. The average growth cone area was measured in the first half, the second half and at the exit site of the floor plate for each condition. (A) No difference in the area was detected between growth cones in the first half and the second half of the floor plate in controls. However, growth cones were found to be much enlarged at the exit site compared to when they were in the floor plate (Friedman test with Dunn's multiple-comparisons test). (B) Unilateral down regulation of Endoglycan induced a significant decrease in the average growth cone area in the second part of the floor plate compared to the first part. However, dI1 growth cones still got much larger when exiting the floor plate compared to when they were located in the floor plate (one-way ANOVA with Sidak's multiple-comparisons test). (C) After unilateral overexpression of Endoglycan, no difference in growth cone area was detected between the first and the second half of the floor plate. Like in all other conditions, they were found to be much enlarged at the exit site (Friedman test with Dunn's multiple-comparisons test). Error bars represent standard deviation. $p < 0.0001$ (****) for all tests. N(embryos)=3 for each condition; n(axons)=70 (Endo Ctrl), 78 (ds Endo), 83 (Endo OE). See *Table 2* for detailed results. Source data and statistics are available in *Figure 11—source data 1* spreadsheet.

The online version of this article includes the following source data for figure 11:

**Source data 1.** Raw data of growth cone size measurements.

was obtained from SourceBioScience (Nottingham, UK). Cultures of commissural neurons were fixed with 4% paraformaldehyde for 15 min at 37°C and stained with goat anti-Axonin1(Contactin2) and rabbit anti-myc antibodies (Abcam, RRID:AB_307014). The number of Axonin-1-positive neurons was counted from seven to nine images per replicate taken with a ×40 water objective in random regions of a well where the confluence of HEK cells was at least 60%. The number of neurons per image was normalized to the average number on control HEK cells for each replicate. Cultures of motoneurons were fixed for 1 hr at room temperature in 4% paraformaldehyde and stained with mouse anti-neurofilament (RMO 270; Thermofisher Scientific, RRID:AB_2532998) and rabbit anti-myc antibodies (Abcam, RRID:AB_307014). The number of neurofilament-positive cells was counted in 16 randomly selected frames (0.4 mm$^2$). Similar results were obtained in three independent experiments. One representative example is shown in *Figure 5—figure supplement 1*. Both commissural and motoneurons were tested for adhesive strength after HEK cells expressing Endoglycan were treated with O-glycosidase (8'000 U/ml) or α2–3,6,8 Neuraminidase (5 U/ml; NEB Cat# E0540S, kit with both enzymes) for 2 hr before commissural neurons or motoneurons were added (*Figure 5*; *Figure 5—figure supplement 2*).

## Growth cone blasting assay

Commissural neurons were dissected from the most dorsal region of spinal cords of HH25/26 embryos that were unilaterally electroporated in ovo at HH17-18 with a plasmid encoding mRFP under the β-actin promotor (30 ng/μl) or co-electroporated with a plasmid encoding the open-reading frame of Endoglycan under the β-actin promotor (300 ng/μl). Dissociated neurons were plated (400 neurons per mm$^2$) on a layer of HEK cells (control or expressing Endoglycan, ~60% confluent) plated the day before on poly-L-lysine-coated (20 μg/ml) 35-mm dishes (Sarstedt). Neurons on HEK cells were cultured overnight at 37°C with 5% CO$_2$ in medium consisting of MEM/Glutamax (Gibco), 4 mg/ml Albumax (Gibco), N3 (100 μg/ml transferrin, 10 μg/ml insulin, 20 ng/ml triiodothyronine, 40 nM progesterone, 200 ng/ml corticosterone, 200 μM putrescine, 60 nM sodium selenite; all from Sigma) and 1 mM pyruvate. The growth cone blasting assay was adapted from *Lemmon et al., 1992*. Before blasting growth cones, a final concentration of 20 mM HEPES was added to the cells to maintain the pH of the medium. Cells were maintained at 37°C in a temperature-controlled chamber (Life Imaging Services) and were visualized with an inverted IX81 microscope (Olympus) equipped with a DP80 camera (Olympus) and a ×20 air objective (2CPLFL PM20x/0.40,

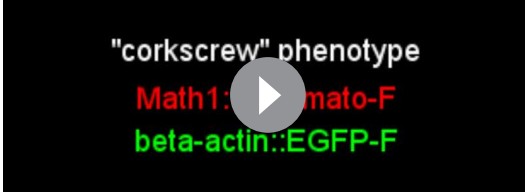

**Video 6.** The trajectory of single dI1 axons in the first half of the floor plate (electroporated half) in the absence of Endoglycan was aberrant and showed a 'corkscrew'-like phenotype. Example of tdTomato-farnesylated-positive dI1 axons taken from 24 hr time-lapse recordings of an 'Endoglycan-knockdown' spinal cord. Arrowheads show how two growth cones are migrating within the first half (electroporated half) of the floor plate. The first one made a loop within the floor-plate cell layer (arrowheads), as shown by the 3D coronal rotation. The second one was attracted toward mislocalized EGFP-positive cells (arrow), made contact with them and then carried out a U-turn toward them before continuing its migration in the direction of the midline. The 3D coronal view confirmed that the second growth cone during its U-turn and the mislocalized cells were located within the commissure (arrowhead).
https://elifesciences.org/articles/64767#video6

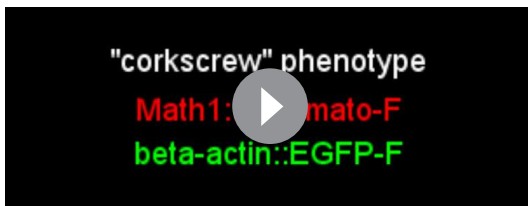

**Video 5.** The trajectory of single dI1 axons in the first half of the floor plate (electroporated half) in the absence of Endoglycan was aberrant and showed a 'corkscrew'-like phenotype. Example of a tdTomato-positive dI1 axon taken from a 24 hr time-lapse recording of an 'Endoglycan-knockdown' spinal cord. Arrowheads show how the growth cone is migrating within the first half (electroporated half) of the floor plate and enters in contact with mislocated EGFP-positive cells (arrows) that induced a 'corkscrew' like morphology of the shaft. The 3D coronal rotation clearly shows that the axon and cells are located within the commissure (arrowhead).
https://elifesciences.org/articles/64767#video5

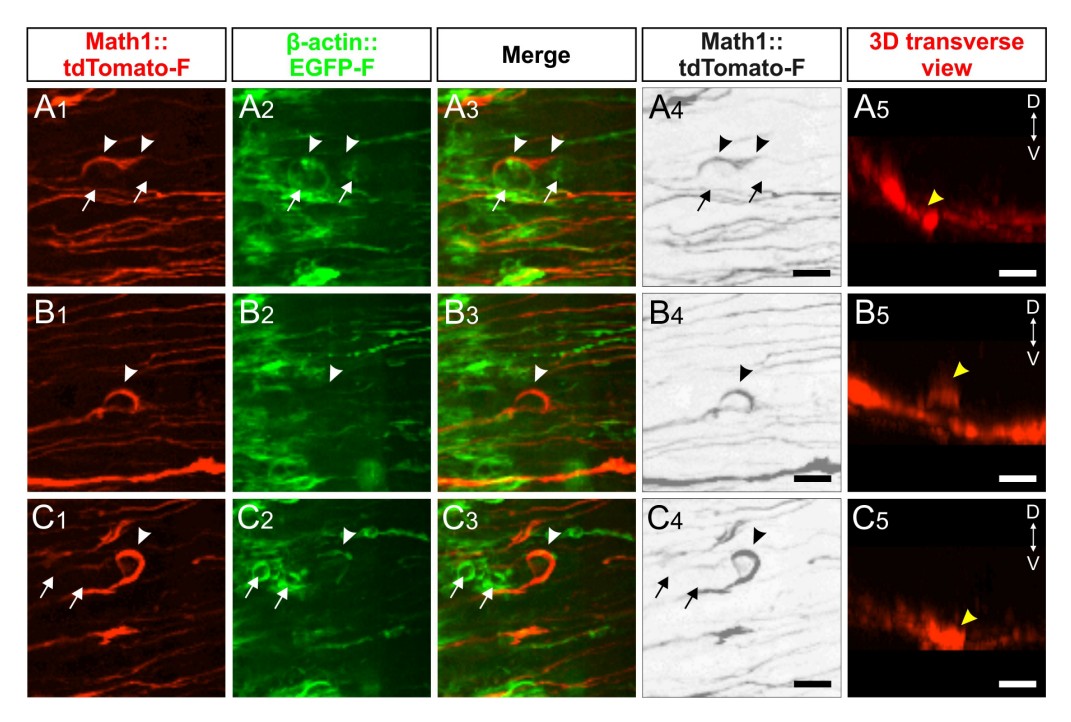

**Figure 12.** Live imaging of dI1 axons after perturbation of Endoglycan expression explains 'corkscew-like' phenotypes by aberrant interactions between axons and floor-plate cells. (A-C) 'Corkscrew'-like phenotypes of dI1 axons expressing farnesylated tdTomato were observed by live imaging in the first half of the floor plate (electroporated half) only after Endoglycan was silenced (see also *Videos 5* and *6*). (A) Dislocated EGFP-positive cells (arrows) caused dI1 axons to deviate from a smooth trajectory inducing a 'corkscrew'-like morphology (arrowheads, $A_{1-4}$). These axons and cells were located in the commissure as shown in a coronal view (yellow arrowhead, $A_5$ and *Video 5*). (B) Some axons were found to form a loop (arrowheads, $B_{1-4}$) invading the layer of the floor plate where somata of floor-plate cells are located as shown in the transverse view (yellow arrowheads, $B_5$ and *Video 6*). (C) Clusters of dislocated roundish EGFP-positive cells (arrows) seemed to retain growth cones in the first half of the floor plate (arrowheads). These axons and cells were located in the commissure as shown in a coronal view (yellow arrowhead, $C_5$ and *Video 6*). D, dorsal; V, ventral. Scale bars: 10 µm.

Olympus). Growth cones were blasted off the HEK cells with a glass micropipette with an opening of 8–13 µm positioned 75 µm away from the leading edge of the growth cone and 50 µm above the growth cone. The height of the micropipette tip above the cells was set by adding 50 µm in the z-axis to the focus plane on the HEK cell surface where the growth cone was localized using the Olympus CellSens Dimension 2.2 software. Then, the micropipette tip was brought into the focus of this z-localization at a distance of 75 µm away from the leading edge of the growth cone that was previously set with the tracing tool of the same software. The glass micropipette was connected to a pump controlled by a liquid chromatography controller (LCC-500, Pharmacia) and a constant flux of 20 µl/min of sterile PBS containing phenol red (Gibco) was directed at the growth cone until it detached. For reproducible results, it was important to use degassed PBS, as the formation of tiny bubbles in the tubing otherwise caused variability in the force exerted on the growth cones. The time growth cones took to detach from the HEK cells was measured manually by a person blind to the experimental condition and for each growth cone a video was taken. A second person (not blind to the experimental condition) was switching between controls and experimental cells to make sure that small instabilities in the pump, the flow of PBS or the temperature of the set-up would not introduce artefacts. We only counted the time a growth cone took to detach, if the HEK cell layer was intact after blasting (see *Video 1*). Note that only results generated with the same glass micropipette were normalized to the control and compared to each other. The size of growth cones within each group was heterogeneous, but there was no difference in average growth cone size between the different conditions (*Figure 7*).

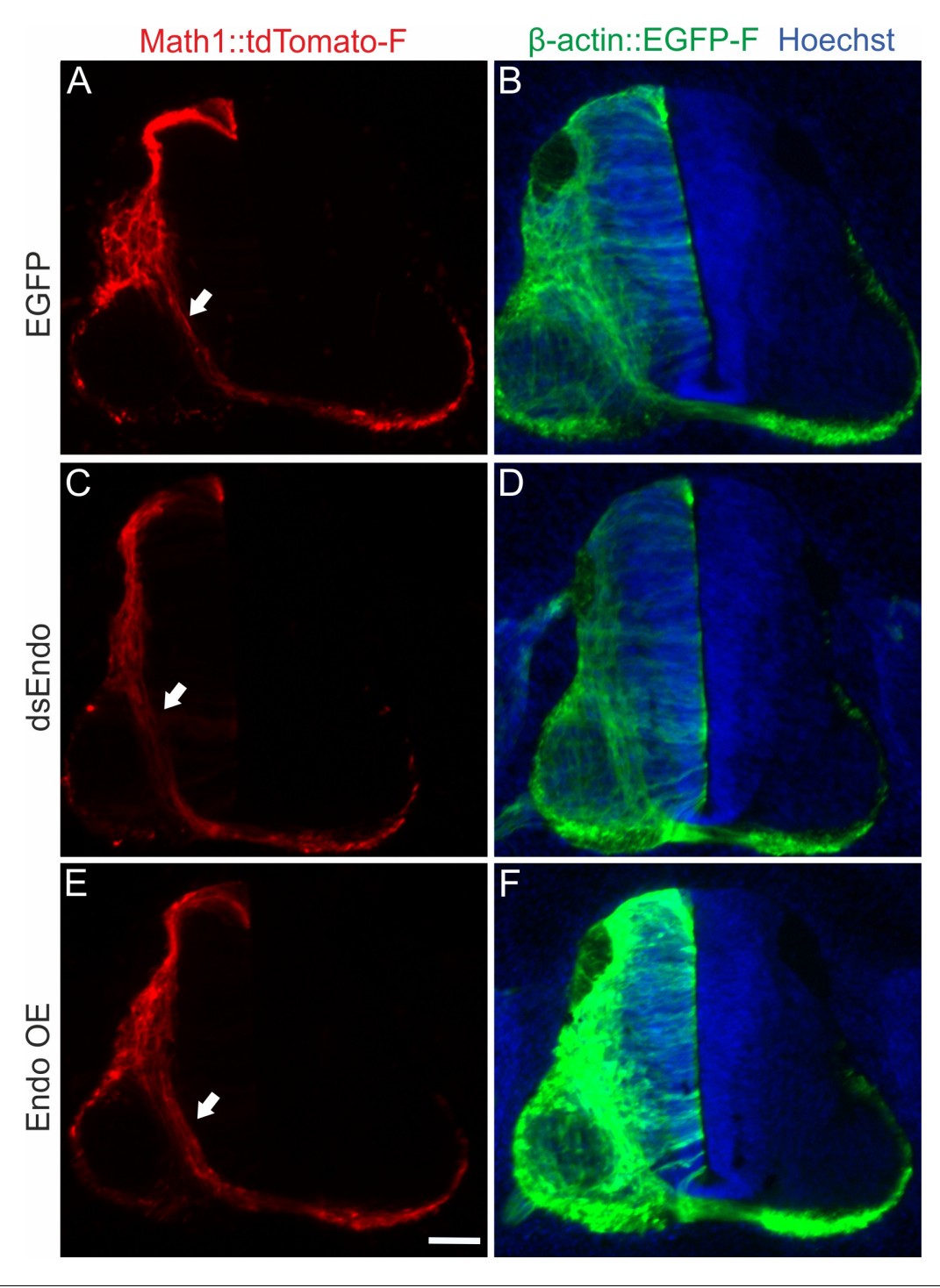

**Figure 13.** Endoglycan does not affect pre-crossing commissural axons. We found no differences in timing or trajectories of pre-crossing commissural axons labeled by the co-electroporation of Math1::tdTomato-F, when we compared control embryos electroporated with an EGFP plasmid (**A,B**) with embryos electroporated with dsEndo (**C,D**) or embryos overexpressing Endoglycan (**E,F**).

## Immunohistochemistry

Cryostat sections were rinsed in PBS at 37°C for 3 min followed by 3 min in cold water. Subsequently, the sections were incubated in 20 mM lysine in 0.1 M sodium phosphate (pH 7.4) for 30 min

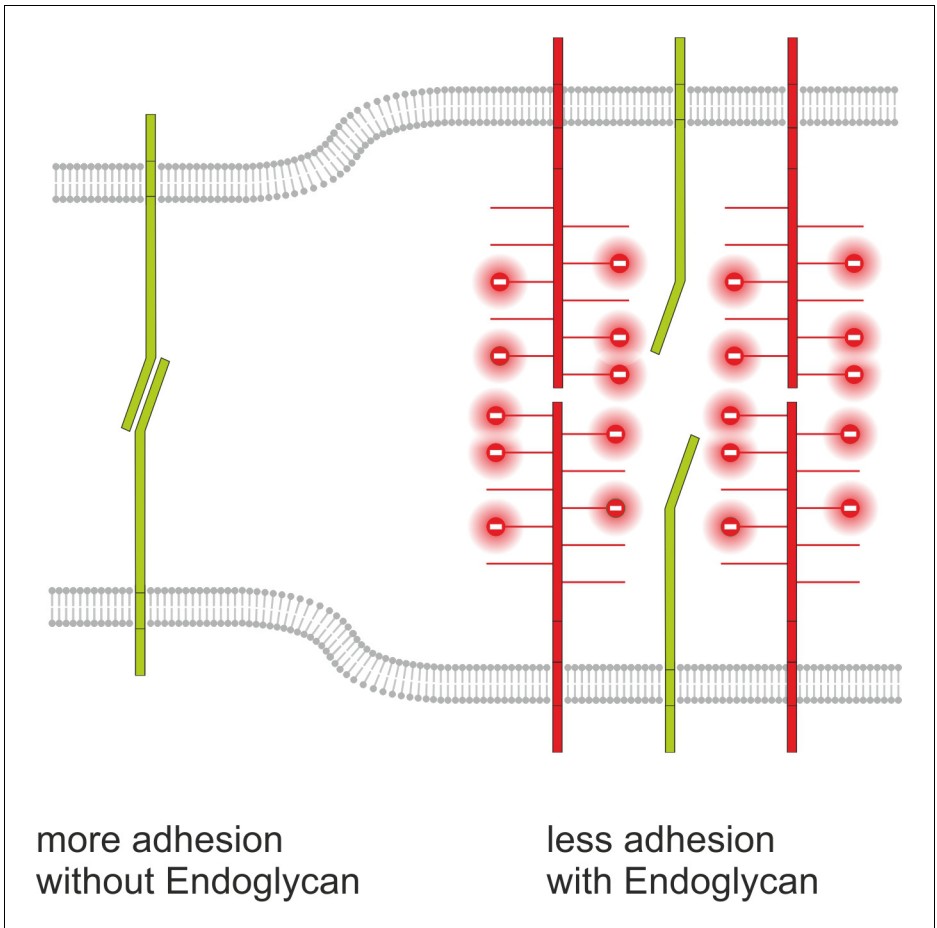

**Figure 14.** Endoglycan modulates cell-cell contact by interference with adhesive strength. Based on our in vivo and in vitro studies, we postulate a model for Endoglycan function in neural circuit formation that suggests an 'anti-adhesive' role by modulation of many specific molecular interactions due to decreasing cell-cell contact. This model is consistent with our rescue experiments demonstrating that the source of Endoglycan did not matter, but the expression level did, as aberrant phenotypes were prevented, when Endoglycan was expressed either in the axon or in the floor plate.

at room temperature before being rinsed in PBS three times for 10 min. The tissue was permeabilized with 0.1% Triton in PBS for 30 min at room temperature and then washed again three times with PBS for 10 min. To prevent unspecific binding of the antibody, the tissue was blocked with 10% fetal calf serum (FCS) in PBS for one hour. Goat anti-GFP (1:400; Rockland, RRID:AB_218187), anti-axonin-1 (rabbit 1:1000 or goat 1:500), anti-NrCAM (goat 1:1000), mouse anti-HNF3β (supernatant; 4C7, DSHB), mouse anti-Islet1 (supernatant; 40.2D6, DSHB, RRID:AB_528315), mouse anti-Nkx2.2 (supernatant; 74.5A5, DSHB, RRID:AB_531794), mouse anti-Pax3 and Pax6 (supernatants, DSHB, RRID:AB_528426 and AB_528427, respectively) were dissolved in 10% FCS/PBS and incubated overnight at 4°C. After three washes in PBS, 10% FCS in PBS was applied again for one hour, followed by the incubation with goat anti-rabbit IgG-Alexa488 (1:250; Molecular Probes, RRID:AB_2576217), donkey anti-rabbit IgG-Cy3 (1:200; Jackson ImmunoResearch, RRID:AB_2307443) or goat anti-mouse IgG-Cy3 (1:250; Jackson ImmunoResearch, RRID:AB_2338680) diluted in 10% FCS in PBS for 90 min at room temperature. The tissue was rinsed five times in PBS for 12 min and then mounted in Celvol (Celanese) or Mowiol. The staining of cryostat sections was analyzed with an upright microscope equipped with fluorescence optics (Olympus BX51). Apoptosis was analyzed as described previously (*Baeriswyl and Stoeckli, 2008*). For analysis of cell death in the floor plate, we used cleaved caspase-3 staining of sections taken from HH25 embryos using a cleaved caspase-3 polyclonal antibody (1:200, Cat# 9661S, Cell Signaling, RRID:AB_2341188).

## Quantification of commissural axon guidance errors

To analyze commissural axon growth and guidance, the embryos were sacrificed between HH25 and 26. The spinal cord was removed, opened at the roof plate ('open-book' preparation) and fixed in 4% paraformaldehyde (PFA) for 40 min to 1 hr at room temperature. To visualize the trajectories of commissural axons, Fast-Dil (5 mg/ml, dissolved in ethanol, Molecular Probes) was injected into the dorsal part of the spinal cord as described previously (*Wilson and Stoeckli, 2012*). Dil injections sites with pathfinding errors were analyzed by a person blind to the experimental condition, using an upright microscope equipped with fluorescence optics (Olympus BX51). All measurements including floor-plate width, thickness of the commissure, and spinal cord width were performed with the analy-SIS Five software from Soft Imaging System. For all measurements, embryos injected with dsRNA derived from *Endoglycan* were compared with embryos injected with the EGFP plasmid only, and untreated controls. For statistical analyses, ANOVA with Tukey's or Sidak's multiple comparisons test was used. Details are given in the Figure legends or in the source data.

## Live imaging

Plasmids encoding farnesylated td-Tomato under the Math1 enhancer, upstream of the β-globin minimal promoter, for dl1 neuron-specific expression (Math1::tdTomato-f), and farnesylated EGFP under the β-actin promoter (β-actin::EGFP-f) were co-injected into the central canal of the chicken neural tube in ovo at HH17/18 and unilaterally electroporated, using a BTX ECM830 square-wave electroporator (five pulses at 25 V with 50 ms duration each), as previously described (*Wilson and Stoeckli, 2012*). For the perturbation of Endoglycan levels, either 300 ng/µl dsRNA derived from the 3'-UTR of *Endoglycan* or a plasmid encoding the open-reading frame of *Endoglycan* under the β-actin promoter were co-injected with the Math1::tdTomato-f plasmid. After electroporation, embryos were covered with sterile PBS and eggs were sealed with tape and incubated at 39°C for 26–30 hr until embryos reached stage HH22.

For live imaging, embryos were sacrificed at HH22. Dissection, mounting and imaging of intact spinal cord were carried out as described in detail in *Dumoulin et al., 2021*. Intact spinal cords were dissected and embedded with the ventral side down in a drop (100 µl) of 0.5% low-melting agarose (FMC; *Pignata et al., 2019*) containing a 6:7 ratio of spinal cord medium (MEM with Glutamax (Gibco) supplemented with 4 mg/ml Albumax (Gibco), 1 mM pyruvate (Sigma), 100 Units/ml Penicillin, and 100 µg/ml Streptomycin in a 35-mm Ibidi µ-Dish with glass bottom (Ibidi, #81158)). Once the agarose polymerized, 200 µl of spinal cord medium were added to the drop and live imaging was started.

Live imaging recordings were carried out with an Olympus IX83 inverted microscope equipped with a spinning disk unit (CSU-X1 10'000 rpm, Yokogawa). Cultured spinal cords were kept at 37°C with 5% $CO_2$ and 95% air in a PeCon cell vivo chamber. Temperature and $CO_2$-levels were controlled by the cell vivo temperature controller and the $CO_2$ controller units (PeCon). Spinal cords were incubated for at least 30 min before imaging was started. We acquired 18–35 planes (1.5 µm spacing) of 2 × 2 binned z-stack images every 15 min for 24 hr with a 20x air objective (UPLSAPO 20x/0.75, Olympus) and an Orca-Flash 4.0 camera (Hamamatsu) with the help of Olympus CellSens Dimension 2.2 software. Z-stacks and maximum projections of Z-stack videos were evaluated and processed using Fiji/ImageJ (*Schindelin et al., 2012*). Temporally color-coded projections were generated using Fiji/ImageJ. Kymograph analysis of axons crossing the floor plate or exiting it was performed as previously described (*Medioni et al., 2015*), using a region of interest (ROI) selection, the re-slice function, and the z-projection of the re-sliced results in Fiji/ImageJ, which allowed following pixel movements within the horizontal axis. The ROI in the floor plate was selected as a 120 × 20 µm² rectangle and the one in the post-crossing segment was a rectangle of 175 × 104 µm². Note that the post-crossing segment ROI was rotated by 90° before running the kymograph analysis. The MtrackJ plugin (*Meijering et al., 2012*) was used to virtually trace single tdTomato-positive dl1 axons crossing the floor plate. Only axons that enter, cross and exit the floor plate during the 24 hr imaging period were traced and quantified. Overlays of traced axons with GFP and brightfield channels were used to assess the time the axons needed to cross the floor plate. Videos of axons with the 'corkscrew' phenotype were generated and assembled from z-stacks that were 2D deconvolved (nearest neighbor) using the Olympus CellSens Dimension 2.2 and Fiji/Image J software, respectively.

## Acknowledgements

We thank Tiziana Flego for excellent technical assistance, Alexandra Moniz and John Darby Cole for help with glycosylation dependence experiment, and members of the lab for discussions and critical reading of the manuscript. This project was supported by the Swiss National Science Foundation and the NCCR Brain Plasticity and Repair (Center of Transgenesis Expertise).

## Additional information

### Funding

| Funder | Grant reference number | Author |
|---|---|---|
| Schweizerischer Nationalfonds zur Förderung der Wissenschaftlichen Forschung | | Esther T Stoeckli |
| Schweizerischer Nationalfonds zur Förderung der Wissenschaftlichen Forschung | Brain Plasticy and Repair | Esther T Stoeckli |

The funders had no role in study design, data collection and interpretation, or the decision to submit the work for publication.

### Author contributions

Thomas Baeriswyl, Formal analysis, Investigation, Methodology, Writing - original draft; Alexandre Dumoulin, Conceptualization, Data curation, Formal analysis, Validation, Investigation, Visualization, Methodology, Writing - original draft, Writing - review and editing; Martina Schaettin, Formal analysis, Investigation, Methodology, Writing - review and editing; Georgia Tsapara, Vera Niederkofler, Formal analysis, Investigation, Writing - original draft; Denise Helbling, Investigation; Evelyn Avilés, Resources; Jeannine A Frei, Resources, Investigation; Nicole H Wilson, Resources, Supervision; Matthias Gesemann, Data curation, Investigation, Visualization; Beat Kunz, Formal analysis, Investigation, Methodology; Esther T Stoeckli, Conceptualization, Data curation, Formal analysis, Supervision, Funding acquisition, Investigation, Writing - original draft, Project administration, Writing - review and editing

### Author ORCIDs

Alexandre Dumoulin (iD) http://orcid.org/0000-0002-2420-6877
Esther T Stoeckli (iD) https://orcid.org/0000-0002-8485-0648

### Decision letter and Author response

Decision letter https://doi.org/10.7554/eLife.64767.sa1
Author response https://doi.org/10.7554/eLife.64767.sa2

## Additional files

### Supplementary files

• Transparent reporting form

## Data availability

All data generated and analyzed during this study are included in the manuscript and supporting files.

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
