## [Decision Letter]

**Acceptance summary:**

Your study reports the role of Endoglycan in the navigation of commissural axons in the spinal cord. Using a range of in vitro and in vivo approaches and time-lapse imaging, you have highlighted the interactions between the axons and their local environment during the floor plate crossing. In particular, in a novel in vitro experiment, neuronal attachment was tested when exposed to forces exerted by a flow of a buffer solution in the presence of Endoglycan. Such experiments strengthen the view that Endoglycan regulates adhesion between commissural axons and their first intermediate target, the floor plate, thereby facilitating axon growth through the ventral spinal cord midline. Your study highlights the importance of balancing adhesive and anti-adhesive forces between growth cones and guidepost cells for proper axon guidance.

**Decision letter after peer review:**

[Editors’ note: the authors submitted for reconsideration following the decision after peer review. What follows is the decision letter after the first round of review.]

Thank you for submitting your work entitled "Endoglycan plays a role in axon guidance and neuronal migration by negatively regulating cell-cell adhesion" for consideration by *eLife*. Your article has been reviewed by a Senior Editor, a Reviewing Editor, and three reviewers. The following individual involved in review of your submission has agreed to reveal their identity: Valerie Castellani (Reviewer #3).

Our decision has been reached after consultation between the reviewers. Based on these discussions and the individual reviews below, we regret to inform you that your work will not be considered further for publication in *eLife* at this time.

The reviewers found your study proposing the sialomucin Endoglycan as a new player in axon guidance in commissural axon growth and Purkinje cell migration of interest, especially in its role in regulating cell adhesion in these processes. Indeed, the mechanisms that regulate and adjust cell-cell adhesive contacts and how these contacts are coupled to guidance decisions are poorly understood, and your study, which was carefully executed, addresses an important and underestimated aspect of axon guidance. However, the reviewers share the opinion that the experimental evidence supporting the mode of action of Endoglycan as an anti-adhesive factor was lacking, in both the in vivo and in vitro settings.

in vivo, you demonstrate through RNA interference that Endoglycan is necessary for proper navigation and migration, but the basic mechanisms underlying the contribution of Endoglycan and how it functions at specific steps of the navigation rather than all along, were not considered to be addressed in depth. While your knockdown experiments cause misrouting and increased tortuosity, and your rescue experiments implicate a role for Endoglycan in guidance, it is unclear how increased adhesion when Endoglycan is disrupted is responsible for these outcomes. The "loosening" of the floor plate cells rather than their enhanced clumping is a puzzle; might the loosening of the floor plate cells cause the misrouting? And a further puzzling aspect is that axon fasciculation near the floor plate when Endoglycan is perturbed appears unchanged.

A more difficult criticism to address, is direct evidence for involvement of Endoglycan in regulating cell-cell adhesion: to demonstrate changes in fasciculation rather than neurite length (which could vary due to many factors), to present both high and low levels of Endoglycan to axons in the non-neuronal cells, and to directly measure cell-cell adhesion strength in the presence of Endoglycan. These experiments are challenging yet would be important and very welcome to the field, should you be able to execute them.

A final note is that the Purkinje cell experiments were little criticized but detracted a bit from the main story.

We hope that these comments, in full below, can aid you in revising your manuscript.*Reviewer #1:*

In this manuscript, Baeriswyl et al., convincingly show that the sialomucin Endoglycan plays an important role in commissural axon growth and Purkinje cell migration, and suggest that the main function of Endoglycan is to unspecifically regulate cell-cell adhesion.

The manuscript thus identifies a new player in axon guidance (confirming data from a previous screen by the group). However, it is currently not exactly clear how Endoglycan affects axon growth. While the authors make some strong claims about adhesion, there is currently no direct evidence for an involvement of Endoglycan in regulating cell-cell adhesion.

If Endoglycan indeed opposes cell adhesion, why don't the floor plate cells stick together better in the knockdowns than in controls (rather than the other way around as it is shown in Figure 2)? The authors address this issue in the Discussion, however, their arguments are not very convincing. The fact that at HH21 the floor plate was intact in the absence of Endoglycan doesn't necessarily mean that cell adhesion is not affected or that Endoglycan is not required for adhesion between floorplate cells. An alternative explanation might be that there are only weak mechanical forces acting on these cells at early stages, which might increase during development and eventually be strong enough to tear apart cells that have reduced adhesion.

Furthermore, if cell-cell adhesion is changed, I would also expect a change in fasciculation of axon bundles on their way to the floor plate. Figure 1 seems to suggest that this might not be the case. To address this issue, fasciculation should be quantified.

I also don't find the in vitro experiments very convincing. Figure 5 shows that neurite length is decreased in Endoglycan overexpressing motoneurons but not when it is overexpressed in COS cells. First, if the main effect of Endoglycan is the unspecific regulation of cell-cell adhesion, I would have expected a stronger effect of the overexpression in COS cells even if the transfection efficiency was only ~50%. Second, it would be good to see a similar set of experiments with decreased expression of Endoglycan. What would the authors expect? And lastly, changes in neurite length can be caused in many different ways, certainly not allowing to conclude “that Endoglycan acts as a negative regulator of cell-cell adhesion”.

Similarly, Figure 5—figure supplement 1 only shows a few images, no quantification of data is shown. For how long were neurons allowed to grow? And even if there are reproducibly less neurons growing on transfected HEK cells, this could be a consequence of many factors (one of them of cause being a change in adhesion).

In summary, the manuscript presents new and exciting data on axon guidance, which might be very relevant for many other systems in which cell-cell adhesion is important. However, while some of the data suggest that Endoglycan might mainly act through the regulation of cell adhesion, this remains an hypothesis until it is directly shown. In order to convincingly demonstrate how Endoglycan regulates axon growth, the authors should directly measure cell-cell adhesion strength and quantify its dependence on the presence of Endoglycan.

Reviewer #2:

In this manuscript by Baeriswyl et al., the authors describe defects in spinal commissural axon guidance and cerebellar Purkinje cell migration that occur after interfering with the function of the sialomucin Endoglycan. Baeriswyl et al., suggest that these phenotypes reflect a function of Endoglycan in reducing cell-cell adhesion.

The authors show that Endoglycan knockdown in the chick spinal cord floor plate or floor plate plus commissural neurons disrupts clustering of floor plate cells and causes increased axon tortuosity during floor plate crossing and abnormal caudal turning of commissural axons after midline crossing. Expression of Endoglycan using commissural neuron- or floor plate-specific promoters can rescue these axon guidance defects, depending on expression levels. The authors further show that knockdown of Endoglycan in the floor plate can suppress the premature commissural axon turning phenotype that results from NrCAM knockdown, and in vitro experiments support the idea that Endoglycan can reduce adhesion of motor neurons to heterologous cells. Lastly, knockdown of Endoglycan in the cerebellum disrupts Purkinje cell positioning and overall cerebellar morphology.

This work presents a series of novel observations, and the idea that Endoglycan modulates cell-cell adhesion to control axon guidance and neuronal migration is of potential interest to the field, even though such an anti-adhesive mechanism for axon guidance has been extensively explored in the context of other molecules, e.g. NCAM. The main conclusions about the mechanism of Endoglycan function in axon guidance and cell migration, however, are not sufficiently supported by the data, and some important control experiments are missing. These problems, together with the limited conceptual novelty of the findings, do not make this manuscript a strong candidate for publication in *eLife*.

1) The experiments involving Endoglycan knockdown in the spinal cord are difficult to interpret for multiple reasons. It is not clear that increased axon-floor plate adhesion is responsible for the observed effects.

a) Knockdown is targeted to the floor plate or "one half of the spinal cord including the floor plate". Data that validate successful knockdown in the targeted cell populations should be included. Also, why is knockdown in commissural neurons alone not attempted?

b) The data showing normal floor plate differentiation (Figure S3) are not sufficient to exclude abnormal tissue patterning as a cause for the axon guidance defects. Shh expression appears reduced (panel C), which could explain abnormal turning of axons after midline crossing. Patterning along the anterior-posterior axis and expression of rostro-caudal guidance cue gradients should be examined quantitatively.

c) Are the axon guidance defects simply a result of reduced floor plate cell clustering? Can guidance errors be observed before changes in floor plate morphology?

d) The authors claim that changes in floor plate morphology coincide with commissural axon crossing of the midline and interpret this as support for the idea that axon-floor plate contact causes the change in floor plate cell clustering. However, axons have clearly already crossed the floor plate at HH21 (Figure 2N,O) before the floor plate phenotype appears. Moreover, successful knockdown of Endoglycan by HH21 would have to be demonstrated to allow interpretation of this experiment. Lastly, even if the timing was consistent, it should be directly tested whether reduced floor plate cell clustering after Endoglycan knockdown depends on commissural axon crossing, e.g. by ablating commissural neurons.

e) It is important to show that a control dsRNA will not "rescue" the NrCAM knockdown defect. Moreover, rescue of the defect by Endoglycan knockdown could result from reduced floor plate repulsion and other mechanisms besides an effect on axon-floor plate adhesion, so the explanatory power of this experiment is limited. Lastly, does loss of NrCAM affect the Endoglycan knockdown phenotypes (turning after crossing, tortuous trajectory during crossing)?

2) Direct support for an anti-adhesive role of Endoglycan in commissural neurons and Purkinje cells is lacking.

a) How would increased floor plate adhesion explain the rostro-caudal axon guidance defects?

b) The in vitro results using spinal motor neurons do not connect to the relevant cell types. Could commissural neuron and Purkinje cell adhesion/growth be studied in similar assays?

c) It would be important to model adhesive commissural axon-floor plate interactions in vitro and study the effect of Endoglycan gain or loss of function in this system. The same applies for Purkinje neurons and the substrate for their migration.

d) The cerebellar phenotype after Endoglycan knockdown could be explained by numerous mechanisms other than increased Purkinje neuron adhesion. Could removal of cell adhesion molecules rescue the migration defect?

Reviewer #3:

The authors took advantage of a substractive hybridization screen that they made several years ago to investigate the functions of endoglycan in the navigation of spinal commissural projections and in the development to the cerebellum. Endoglycan belongs to a family of cyalomucins also comprising CD34 and Podocalyxin, known to regulate cell-cell interactions through anti-adhesive properties.

The authors report for the first time a requirement for endoglycan in the two contexts that they have examined, which is thus an interesting finding. They also propose that endoglycan functions are mediated via anti-adhesive properties.

The mechanisms that regulate and adjust cell- cell adhesive contacts and how these contacts are coupled to guidance decisions are yet poorly known. Therefore, the study addresses an important and underestimated aspect of axon guidance. Nevertheless, I have the feeling that, beyond the demonstration through RNA interference that endoglycan is necessary for proper navigation and migration, the basic mechanisms underlying the contribution of endoglycan and what makes it that it plays a role at some specific steps of the navigation rather than all along, have been superficially addressed. We are indeed let with a very unclear picture on how it contributes to enable proper navigation of the floor plate, and proper purkinje cell migration. The substractive screen was designed to pick up genes important for rostral turning but from the described phenotypes, it looks like what is primarily affected is the FP crossing. Indeed, the axons still appear able to turn, even though the turning is abnormally disconnected to the crossing. Overall and for these reasons, I have a number of issues which I think are necessary to address. They are listed below.

1) Subtractive hybridization screen: the authors should describe the general strategy and method, even though both have been reported in their previous studies. The rationale is needed to correlate the approach with the expected and observed phenotypes of endoglycan knock-down. In particular, the screen was designed for identifying cues instructing rostral turning guidance. How is regulation of cell adhesion during floor plate relevant for the turning? should be expect a rostral-caudal gradient of endoglycan?

2) In Figure 2 at HH21: the axonin labeling seen in the picture is indeed very strong, not as if there were only a few axons that already had crossed, as indicated in the result section of the manuscript.

3) The HH21 untreated control is not shown. This is needed to compare the shape of the FP at this stage or at least would it be the case, the authors should make it clear that the shape is expected to be similar to that at later stage.

The FP appears strongly disorganized, based on the staining of FP cell soma and nuclei. It would strongly add to illustrate how radial fibers of FP cells look like, because commissural axons navigate within this fiber network. A better morphological characterization of the FP structure is also needed. Are cells losing their bipolar morphology? Are the radial fiber still present and attached to the basal side? In the cerebellum, endoglycan depletion alters cell proliferation, could it be the case for FP cells?

The authors consider that since at HH21 the FP is unaffected when only a few axons hare reached the FP, the FP disorganization at HH26 is therefore an indirect consequence of the presence of the axons. This conclusion is weak and lacks experimental support.

4) The disorganization of FP cells could well alter the presentation pattern of local guidance cues or local cell adhesion molecules. Expression profiles of the principal players should be investigated at protein levels, when antibodies are available.

5) How knock-down of endoglycan impacts on levels of endoglycan is not shown. This is also true for dose-dependent rescue: the authors report that axon guidance was rescued by addition of endoglycan in dI1 neurons at a low concentration or in floor-plate cells at a medium concentration. One would like to see whether these different conditions really result in differences of endoglycan levels. It might be possible to get insights into endoglycan levels be done, maybe with western blot of pieces of spinal cords. On the least, knock-down efficiency and rescue could be assessed on cells transfected with tagged endoglycan?

6) The anti-adhesive role of endoglycan during FP crossing is rather deduced from the literature than supported by the data. The authors need to investigate first whether manipulating the adhesive/anti-adhesive balance results in alterations of FP crossing and post-crossing. According to their model, abrogating endoglycan results in an unbalanced weight of NrCAM over contactin. The authors could for example test whether over expressing contactin strictly mimics NrCAM/endoglycan knock-down.

7) Why were the in vitro experiments achieved with motoneurons rather than commissural neurons? Moreover, the performed analysis rather assesses whether endoglycan regulates axon outgrowth, not really cell-cell contacts and adhesion. it is somehow surprising that this outgrowth effect, if it applies also to commissural axons, does not result in alteration of the pre-crossing navigation, for example by delaying the growth towards the FP.

8) "Cell-autonomous" versus "non-cell autonomous" endoglycan contributions are unclear for FP navigation. On the one hand endoglycan is reported enriched in the FP at HH20, a stage when there are not impact of endoglycan KD on FP cell organization. Later on, higher expression is found in the dorsal spinal cord where commissural neurons are, and at this stage they have crossed the FP. Thus, this increase might rather be important for post-crossing navigation.

9) Is endoglycan relevant for axon fasciculation?

10) The impact of endoglycan deletion on cerebellar development is rather impressive. The interpretations are simplest because Purkinje cells are the only source of endoglycan. This makes it possible to reconstitute the sequence of direct and indirect events leading to the different abnormalities the authors found. Nevertheless, very few attempts are done for characterizing the nature of the interactions and adhesion required during Purkinje cell migration. One would like to know more about the adhesion molecules that are needed for this cell-type, or the nature of the migration process that is impaired. Is it that the leading process sticks to the substrate, or is the nuclei translocation prevented?

11) Also, the Discussion should be extended on the question of folding. Is folding defect a direct consequence of the decrease amount of produced granule cells? or the lack of purkinje cells at their final position?

12) The authors mention that “Pathfinding was normal in embryos electroporated with dsRNA derived from Podocalyxin". They also quote in their introduction that this cyalomucin is also expressed in the developing nervous system. Therefore, why this experiment was done is unclear, as well as a conclusion lacks. Was it done to document a specific contribution of endoglycan over the other cyalomucins? or a specific functional property of endoglycan?

[Editors’ note: further revisions were suggested prior to acceptance, as described below.]

Thank you for submitting your work entitled "Endoglycan plays a role in axon guidance and neuronal migration by negatively regulating cell-cell adhesion" for consideration by *eLife*. Your article has been reviewed by a Reviewing Editor and a Senior Editor, a Reviewing Editor, and three reviewers. The following individual involved in review of your submission has agreed to reveal their identity: Valerie Castellani (Reviewer #3).

Our decision has been reached after consultation between the reviewers. Based on these discussions and the individual reviews below, we regret to inform you that your work will not be considered further for publication in *eLife*.

The three reviewers believe that your study offers some new, potentially important insights into the regulation of axon pathfinding. The findings are certainly new and exciting. think that the work nicely shows that Endoglycan contributes to commissural axon navigation. In this revision, the phenotypes have been characterized in more detail, and important control experiments were added.

Nonetheless, the reviewers do not find supporting evidence for your model, that Endoglycan facilitates floor plate navigation by controlling the strength of "adhesion and anti-adhesion" contacts resulting from cell-axon or axon-axon contacts during navigation.

In the reviewer consultation after the reviews below were submitted, the reviewers were in agreement in believing that the in vivo analyses alone do not truly inform on the mechanism of Eendoglycan-mediated cell adhesion. They would like to see in vitro experiments of the type your lab should be able to execute, but using commissural, not motor, neurons, to complement your in vivo analysis. Examining adhesion of commissural neurons to heterologous cells and/or floor plate cells in vitro, combined with manipulations of Endoglycan expression, could directly support (or not) the idea that Endoglycan modulates adhesive interactions between commissural axons and floor plate cells. Without such experiments, the statements about adhesion need to be toned down; adhesion should be critically discussed as one possible mechanism.

Since the requested experiments would take longer than two months, especially in this time of lab ramping-up, your manuscript will be rejected at this time.

Reviewer #1:

In the revised version of their manuscript, Baeriswyl et al., have addressed several of my concerns. Additional control experiments have helped solidify some of the authors' conclusions, and the live imaging experiments provide new information about the effect of Endoglycan knockdown and overexpression on commissural axon behavior during floor plate crossing. However, direct support for the anti-adhesive or "lubricating" role of Endoglycan in the context of commissural axon-floor plate or Purkinje neuron-substrate interactions is still lacking. I am surprised that the authors decided not to include some very doable experiments in their extensive revision work to strengthen this hypothesis. I therefore believe that the authors either need to tone down their conclusions about molecular mechanism or perform the necessary experiments to test their interpretation more directly.

1) The experiments involving motor neurons cultured on Endoglycan-expressing HEK cells should be carried out with commissural neurons instead. These neurons are easy to grow in culture, making this a very feasible experiment. Examining adhesion of commissural neurons to heterologous cells and/or floor plate cells in vitro, combined with manipulations of Endoglycan expression, could directly support (or not) the idea that Endoglycan modulates adhesive interactions between commissural axons and floor plate cells. Without direct evidence of this kind, the language centered around anti-adhesive or lubricating properties of Endoglycan in an axon guidance context does not seem justified.

2) The data on cerebellum development after Endoglycan knockdown are difficult to interpret due to the dramatic effects on overall tissue morphology. Without extensive experiments focused on molecular mechanisms underlying the phenotype, these findings do not integrate well with the rest of the paper and should be removed.

Reviewer #2:

The authors have conducted several new experiments to support their hypothesis that Endoglycan is regulating cell-cell adhesion. While I still think that the manuscript presents exciting new data on axon guidance, the authors still do not provide any direct evidence for an involvement of Endoglycan in regulating cell-cell adhesion.

I appreciate that the authors tried to measure adhesive strength in an in vitro assay as designed by Vance Lemmon and colleagues. However, I am a bit concerned about their statement "we did not succeed in measuring the differences reproducibly". Perhaps there are just no differences?

Unfortunately, also the ablation of commissural neurons suggested by reviewer 2 was not attempted. This experiment would have tested whether the reduced floor plate cell clustering after Endoglycan knockdown indeed depends on direct interactions between axons and floor place cells, which is currently not clear.

The authors present new figures (Figure 6, Figure 7 and Figure 8) and movies "demonstrating that the observed differences in axonal midline crossing are indeed explained by aberrant interactions between growth cones and floor-plate cells". However, the nature of these interactions remains unclear. The 'corkscrew' phenotype shown in Figure 8, Video 4 and Video 5 could also be explained by a change in microtubule dynamics (Krieg et al., 2017). Endoglycan could very well interfere with other signals present during floor plate crossing, providing an alternative explanation for the observed axonal growth patterns.

The authors also claim that "we now only show adhesion of cells (Figure 4)." Figure 4 shows a reduction in cell number, which could not only be a consequence of reduced adhesion but also because of other effects, such as apoptosis.

If I understand correctly, the authors suggest that axons pull on FP cells: "Most importantly, evidence from our in vivo and live cell imaging experiments suggests that axons cause the disruption of the floor plate (see Figure 8 and Video 1 and Video 3, subsection “Endoglycan acts as “lubricant” for growth cone movement in the floor plate”)." I cannot see any such evidence in the presented data. Time lapse movies showing how axons pull on FP cells are currently missing. The ablation of commissural neurons mentioned above would also provide such evidence.

Overall, I'm not sure if I understand the concept of a lubricant in this context. The authors state: "Endoglycan is not affecting adhesion between floor-plate cells, as it is not a cell adhesion molecule, or an “anti-adhesion” molecule to be precise. Rather Endoglycan acts as a “lubricant” that is required between axons and floor-plate cells, or between cells that move with respect to each other, because it allows for the necessary dynamics in the adhesion between growth cones and floor-plate cells mediated by adhesion molecules." What is the difference between anti-adhesion and a lubricant? Both reduce friction, it should be the same? And do the authors suggest that whole axons are moving over the floor plate: "In contrast, at the floor plate axons need to move relative to the floor plate."? As far as I'm aware, axons per se do not move much. Growth is mostly achieved at the distal end of the axon. And if a lubricant would allow sliding there, growth cones could not generate forces required for their motility. Furthermore, if adhesion would be reduced, growth cones should also be smaller. Yet, Figure 11 seems to suggest that growth cone sizes are similar in all conditions.

Thus, while the authors clearly demonstrate, for the first time, the importance of Endoglycan in axon guidance and neuronal migration, I am still not convinced that it acts mostly through regulation of cell-cell adhesion as suggested by the authors.

Reviewer #3:

The study has been significantly extended, in particular with a series of novel experiments performed with live imaging. The resulting data clarify the role of endoglycan and its mode of action. For example, the authors observed that manipulations of endoglycan levels have impact on the velocity of commissural axons and growth cone size. Differences appear tiny but statistically significant. They propose a model whereby endoglycan provided by FP cells and axons acts as a lubricant to facilitate the FP navigation and the axon-FP cell contacts. This is an interesting idea that beyond endoglycan, shed light on yet poorly understood roles of modulators.

In addition, the authors addressed several concerns that were raised by their previous version of the work. They checked whether manipulations of endoglycan alter the expression of molecules known to guide commissural axons, which as far as we can tell from general expression patterns is not the case.

I also think that the part on the cerebellum model is now more integrated to the rest of the work, and I agree that it enable to make a broader message on the functional properties of endoglycan.

I still consider that, for better consistency, the experiments to demonstrate the regulation of adhesion by endoglycan should have been carried out on the population of neurons that is investigated here (commissural neurons) and not on motoneurons. That said, I believe that what the authors report with motoneurons might apply to commissural neurons and the findings are consistent with their model on commissural axon navigation. The authors modified their initial analysis, now concentrating on adhesion rather than outgrowth, which is to me very pertinent. They also carried out some biochemical analysis to document endoglycan mode of action, showing contribution of glycosylation.

Overall, I found that the study is improved by the novel data.

Here are some specific comments:

For all figures related to the time-lapse imaging.: the number of experiments, embryos etc… are not indicated.

It seems to me that there are some redundancy between Figure 6 and Figure 7 in the questions that the experiments address.

[Editors’ note: further revisions were suggested prior to acceptance, as described below.]

Thank you for submitting your article "Endoglycan plays a role in axon guidance and neuronal migration by negatively regulating cell-cell adhesion" for consideration by *eLife*. Your article has been reviewed by Didier Stainier as the Senior Editor, a Reviewing Editor, and three reviewers. The following individual involved in review of your submission has agreed to reveal their identity: Kristian Franze (Reviewer #3).

The reviewers have discussed the reviews with one another and the Reviewing Editor has drafted this decision to help you prepare a revised submission.

Summary:

This study presents a role for the sialomucin protein Endoglycan in axon guidance and neuronal migration. The authors describe defects in spinal commissural axon guidance and cerebellar Purkinje cell migration that occur after interfering with Endoglycan function. With an array of in vivo and in vitro approaches, especially a novel in vitro experiment, in which neurons detached faster from other cells in the presence of Endoglycan when exposed to forces exerted by a flow of buffer solution, the authors strengthen the view that Endoglycan reduces adhesion between commissural axons and their first intermediate target, the floor plate, thereby facilitating axon growth across the spinal cord midline. This work establishes a novel function for Endoglycan in axon guidance and highlights the importance of balancing adhesive and anti-adhesive forces between growth cones and guidepost cells for proper axon navigation, adding to our understanding of how proper neuronal networks are formed during development.

Essential revisions:

Your study highlights Endoglycan in regulating neuronal growth and migration, pointing to a mechanism involving anti-adhesion. The reviewers felt that you have addressed all the major criticisms, and with the addition of your apt in vitro experiments addressing cell adhesion (new Figure 5), that the manuscript is generally improved. They comment that this will be "a very nice paper that significantly contributes to the field".

There are still revisions called for, as listed below, primarily altering the wording used to describe the action of endoglycan. Moreover, the reviewers encourage you to consider removing the data on the cerebellum.

I) Cerebellar data: Two of the three reviewers urge you to consider removing the cerebellar data on Purkinje cells, as this analysis distracts from the now well fleshed-out message relating to axon guidance. The effects of Endoglycan knockdown on Purkinje cell migration are compelling per se, but the interpretation of these defects is complicated by the overall disrupted morphology of the cerebellum and lack of data that directly demonstrate whether and how Endoglycan modulates Purkinje cell adhesion to other relevant cell types. In your rebuttal, you indicate that defects in granule cell proliferation are consistent with impaired Purkinje cell migration. This is certainly true, but numerous alternative interpretations exist, and no attempt is made to probe the authors' model further. Without extensive experiments focused on molecular mechanisms underlying the phenotype, these findings do not integrate well with the rest of the paper. You might consider publishing the cerebellar work as a smaller study. One question is whether cell migration/translocation and axon outgrowth rely on adhesion/anti-adhesion in the cerebellum as they do in the floor plate.

II) Endoglycan at the floor plate:

1) “Lubricant"? vs negatively regulating cell adhesion:

a) While you now provide an in vitro experiment suggesting a role of endoglycans in regulating cell adhesion, there is no experimental evidence for a role of endoglycans as a lubricant in vivo (which would certainly be very difficult to achieve, if at all possible). This should be kept in mind when interpreting the data. You are thus encouraged to tone down the wording referring to the proposed mechanism throughout the manuscript, for example in statements such as "Endoglycan acts as “lubricant” for growth cone movement in the floor plate".

b) The second part of the title is not backed up by the data ("by negatively regulating cell-cell adhesion"); the abstract should explicitly state that adhesion is affected in vitro.

c) Statements such as "Thus, the aberrant morphology of the floor plate at HH25 is explained by the inability of axons to break contacts with floor-plate cells in the absence of Endoglycan" should be phrased more carefully. Replacing "is explained" by "could be explained" would not make the study weaker but rather protect you in the future should it turn out that there is a different mechanism at play. Finally, because of the potential problems mentioned above associated with the blast experiment, you might omit the word "strong" in the sentence "Strong support for this hypothesis was contributed by in vitro findings that the adhesive strength… ".

d) It is admittedly difficult to distinguish a lubricant effect from a system that would control the strength of an adhesion or anti-adhesion effect (for example by controlling the number of binding partners or their availability). If the idea is that endoglycan does not target a specific adhesion mechanism but rather has a general effect, would you consider the role of endoglycan to be similar to the one attributed to PSA of NCAM adhesion molecules? This should be discussed.

2) Blasting assay: The growth cone blasting assay introduced in the revised version of the manuscript is a very difficult experiment, and your efforts should be praised. You have carefully considered some of the confounding factors; however, some open questions remain.

a) Factors such as the distance of the pipette from the substrate (i.e., from the top surface of the HEK cells) and the area and height of the growth cones will also be crucial determinants of the time required to detach growth cones. How was the height of the pipette tip above the cells controlled, and how accurately could it be determined? Was the growth cone area the same between the groups? Where the experiments done blindly?

b) Furthermore, why was PBS used, which usually does not contain any calcium? Calcium has a key role in cell adhesion.

Details of these aspects should be provided (including images of growth cones at higher magnification and, if possible, a quantification of growth cone areas), and the experiment should be critically discussed.

3) A more detailed description of what constitutes an abnormal injection site in Figure 3 would greatly help with interpretation of those data. Quantification of "normal injection sites" in Figure 3 somewhat unsatisfying, especially for the comparison of gain-of-function and loss-of-function effects. After all, more adhesion and less adhesion should not produce the exact same phenotypes. Could the abnormal injection sites be further categorized, e.g. as stalled, 'corkscrew', premature turning, aberrant postcrossing A-P guidance etc.?

4) You have now used a carpet of HEK cells rather than just a cell adhesion molecule coated on plastic, which you argue "probably also helped to avoid artefacts". It is not clear how a homogeneous, controlled coating of a plastic surface with purified proteins should cause more artifacts than cells expressing a multitude of different proteins at their surface which may vary in space and time. While it is very difficult to see any details in Figure 4, HEK cells in the two groups (Figure 4A) seem to have different morphologies. If this is true, changes in HEK cells could lead to changes in cell-cell contacts to neurons that are independent of Endoglycans. This should be discussed.

5) In the Results section, the interpretation of defects as a problem with adhesion is introduced very early. It would be better to draw that conclusion later, after the experiments in Figure 4 and Figure 5. In general, some of the conclusions in the Results section that combine outcomes from multiple experiments might be better suited for inclusion in the Discussion.

---

## [Author Response]

[Editors’ note: the authors resubmitted a revised version of the paper for consideration. What follows is the authors’ response to the first round of review.]

The reviewers found your study proposing the sialomucin Endoglycan as a new player in axon guidance in commissural axon growth and Purkinje cell migration of interest, especially in its role in regulating cell adhesion in these processes. Indeed, the mechanisms that regulate and adjust cell-cell adhesive contacts and how these contacts are coupled to guidance decisions are poorly understood, and your study, which was carefully executed, addresses an important and underestimated aspect of axon guidance. However, the reviewers share the opinion that the experimental evidence supporting the mode of action of Endoglycan as an anti-adhesive factor was lacking, in both the in vivo and in vitro settings.in vivo, you demonstrate through RNA interference that Endoglycan is necessary for proper navigation and migration, but the basic mechanisms underlying the contribution of Endoglycan and how it functions at specific steps of the navigation rather than all along, were not considered to be addressed in depth. While your knockdown experiments cause misrouting and increased tortuosity, and your rescue experiments implicate a role for Endoglycan in guidance, it is unclear how increased adhesion when Endoglycan is disrupted is responsible for these outcomes. The "loosening" of the floor plate cells rather than their enhanced clumping is a puzzle; might the loosening of the floor plate cells cause the misrouting? And a further puzzling aspect is that axon fasciculation near the floor plate when Endoglycan is perturbed appears unchanged.

We thank the editor and reviewers for the efforts and comments made during the evaluation of our manuscript. We are happy that our work was considered to be carefully executed and that is addresses an important and underestimated aspect of axon guidance. Because the consensus was that the mode of Endoglycan action was not sufficiently explained in our in vivo and in vitro studies, we added many new experiments supporting our conclusion that Endoglycan affects neural circuit formation by modulating cell-cell adhesion during cell migration and axon guidance. In particular, and as detailed below in our one-to-one response to the reviewers’ comments, we have added ex vivo live cell imaging in intact spinal cords to provide additional results supporting our model. Furthermore, we demonstrate that glycosylation is required for Endoglycan function, in line with our hypothesis that Endoglycan acts by lowering cell-cell contacts through its high glycosylation content in the mucin domain. We added control experiments that exclude an indirect effect of Endoglycan on axon guidance at the spinal cord midline through changes in the expression of known axon guidance cues.

Initially, we were also surprised by our findings that the floor plate integrity suffers in the absence of Endoglycan. As indicated, one might think that downregulation of Endoglycan would strengthen the adhesion between floor-plate cells rather than “loosening” them. However, it looks like Endoglycan is not affecting adhesion between floor-plate cells, as it is not a cell adhesion molecule, or an “anti-adhesion” molecule to be precise. Rather Endoglycan acts as a “lubricant” that is required between axons and floorplate cells, or between cells that move with respect to each other, because it allows for the necessary dynamics in the adhesion between growth cones and floor-plate cells mediated by adhesion molecules. This was our conclusion based on all our results presented in the original version of our manuscript. We have made considerable efforts to add further support for this conclusion in our revised manuscript by demonstrating that axon – floor-plate cell interactions change with the amount of Endoglycan (Figure 6, Figure 7, Figure 8 of the revised manuscript). The reviewers also found it puzzling that axon fasciculation near the floor plate was not perturbed. In agreement with our view of Endoglycan function, this makes sense. Endoglycan would not be required between axons that move in the same direction, as they do not need to move against each other. Rather they move in sync, in loose contact but without fasciculation in the strict sense. In contrast, at the floor plate axons need to move relative to the floor plate. Later crossing axons are dislocating previously crossing axons by breaking their contact with the floor-plate cells. Thus, here, a “lubricant” is required to allow axons to move across the midline by balancing adhesion in a dynamic way. We realize that the mechanism of Endoglycan action was maybe not explained clearly enough. We thus tried to put more emphasis on the idea of Endoglycan as a lubricant. With this in mind, it makes sense that the floor-plate integrity is not affected in the absence of Endoglycan unless axons are exerting forces. Please see below for a point-to-point response to issues raised by reviewers.

A more difficult criticism to address, is direct evidence for involvement of Endoglycan in regulating cell-cell adhesion: to demonstrate changes in fasciculation rather than neurite length (which could vary due to many factors), to present both high and low levels of Endoglycan to axons in the non-neuronal cells, and to directly measure cell-cell adhesion strength in the presence of Endoglycan. These experiments are challenging yet would be important and very welcome to the field, should you be able to execute them.

Indeed, it has taken us much more than two months to carry out all the additional experiments. We have tried measurements of adhesion in vitro but this was not very convincing. Therefore, we have changed the strategy and added new live imaging studies and additional in vivo experiments. The live imaging experiments support the fact that Endoglycan affects axon – floor-plate adhesion rather than axon-axon contacts before or after midline crossing. Taken together our previous and our new results strongly support our conclusions that Endoglycan acts as a “lubricant” allowing axons to navigate their intermediate target, the floor plate. See below for details.

A final note is that the Purkinje cell experiments were little criticized but detracted a bit from the main story.

We respectfully disagree. These experiments clearly take away the focus from axon-axon interactions, which is never what we suggested, but unfortunately was not sufficiently clear. In fact, we think that these experiments add to the conclusion that Endoglycan acts as a “lubricant” by regulating cell-cell adhesion in a more general way, not restricted to midline crossing. We have rephrased the text to better explain this point.

Reviewer #1:In this manuscript, Baeriswyl et al., convincingly show that the sialomucin Endoglycan plays an important role in commissural axon growth and Purkinje cell migration, and suggest that the main function of Endoglycan is to unspecifically regulate cell-cell adhesion.The manuscript thus identifies a new player in axon guidance (confirming data from a previous screen by the group). However, it is currently not exactly clear how Endoglycan affects axon growth. While the authors make some strong claims about adhesion, there is currently no direct evidence for an involvement of Endoglycan in regulating cell-cell adhesion.

Endoglycan does not affect axon growth per se. For instance, we did not see any difference in the arrival of commissural axons at the floor plate (see also Figure 13). Axons were not found to arrive at the floor-plate entry site later in embryos treated with dsEndo or earlier in embryos overexpressing Endoglycan. The difference was however significant during floor-plate crossing, as shown by our live imaging data added as Figure 6, Figure 7, Table 2, Figure 11, Video 1, Video 2, Video 3, Video 4, Video 5, Results.

If Endoglycan indeed opposes cell adhesion, why don't the floor plate cells stick together better in the knockdowns than in controls (rather than the other way around as it is shown in Figure 2)? The authors address this issue in the Discussion, however, their arguments are not very convincing. The fact that at HH21 the floor plate was intact in the absence of Endoglycan doesn't necessarily mean that cell adhesion is not affected or that Endoglycan is not required for adhesion between floorplate cells. An alternative explanation might be that there are only weak mechanical forces acting on these cells at early stages, which might increase during development and eventually be strong enough to tear apart cells that have reduced adhesion.Furthermore, if cell-cell adhesion is changed, I would also expect a change in fasciculation of axon bundles on their way to the floor plate. Figure 1 seems to suggest that this might not be the case. To address this issue, fasciculation should be quantified.

Initially, we were also surprised to find the changes in floor-plate integrity. However, at second glance, our data clearly support a role of Endoglycan as a “lubricant”. It is not the opposite of an adhesion molecule but a dynamic regulator of adhesion that allows the movement of growth cones across the midline. Axons that cross the floor plate need to dislocate the axons that have crossed before. This process generates stress/force between the axons and the floor-plate cells. In the absence of the lubricant, the mechanical forces are too big and the floor plate is disrupted. Therefore, we find single cells dislocated into the axonal commissure, Purkinje cells stuck on their way to the periphery, but never a general disassembly of the tissue. This is clearly illustrated by the integrity of the spinal cord tissue after electroporation of dsRNA or after overexpression of Endoglycan (new Figure 3—figure supplement 1, Figure 3—figure supplement 3 and Figure 13, Results).

Reviewer 1 suggests an alternative explanation: only weak mechanical forces would act at HH21 and therefore might not be sufficient to explain, why the floor plate does not lose its integrity at that stage, but would do at HH25, when forces get stronger. This is actually absolutely in line with our explanation: Axons (the producers of mechanical forces) are compromising floor-plate integrity because they stick too strongly to the floor-plate cells in the absence of the lubricant Endoglycan (See Video 4 and Video 5, subsection “Endoglycan acts as ‘lubricant’ for growth cone movement in the floor plate”). As pointed out, we cannot completely rule out that the adhesion between floor plate cells is also affected. However, the “gaps” in the floor plate can only be explained by a higher adhesion between axons and floor-plate cells, if cells would adhere less strongly to each other and therefore the floor plate would “fall apart” only later, why would it do so in the absence of Endoglycan that enhances the adhesiveness?

Similarly, our findings that the Endoglycan loss-of-function phenotype can also be rescued when Endoglycan is provided by the axons further supports our “lubricant” hypothesis. Strongest support comes from our live imaging data, where we clearly find that the interaction between axons and floorplate cells is compromised but only after early axons have crossed the floor plate, in line with our observations and interpretation of the findings shown in Figure 2.

I also don't find the in vitro experiments very convincing. Figure 5 shows that neurite length is decreased in Endoglycan overexpressing motoneurons but not when it is overexpressed in COS cells. First, if the main effect of Endoglycan is the unspecific regulation of cell-cell adhesion, I would have expected a stronger effect of the overexpression in COS cells even if the transfection efficiency was only ~50%. Second, it would be good to see a similar set of experiments with decreased expression of Endoglycan. What would the authors expect? And lastly, changes in neurite length can be caused in many different ways, certainly not allowing to conclude “that Endoglycan acts as a negative regulator of cell-cell adhesion”.

We have removed Figure 5 and replaced it by new Figure 4 and Figure 5—figure supplement 2 (subsection “Endoglycan is a negative regulator of cell adhesion”). We realize that these results were confusing in the way they were presented. Figure 4 shows that fewer neurons attach to HEK cells stably expressing Endoglycan, whereas Figure 5—figure supplement 2shows that the “anti-adhesive” effect of Endoglycan is dependent on its glycosylation in the mucin domain, as removal of poly-sialic acid or O-linked glycosylation abolishes the anti-adhesive effect.

Similarly, Figure 5—figure supplement 1 only shows a few images, no quantification of data is shown. For how long were neurons allowed to grow? And even if there are reproducibly less neurons growing on transfected HEK cells, this could be a consequence of many factors (one of them of cause being a change in adhesion).

See above. We have removed (old) Figure 4 looking at neurite length. We have quantified the decrease in adhesiveness by counting cells per area in new Figure 4 (Results; see also Figure 5—figure supplement 2) for a role in glycosylation in the adhesion-modulating effect.

In summary, the manuscript presents new and exciting data on axon guidance, which might be very relevant for many other systems in which cell-cell adhesion is important. However, while some of the data suggest that Endoglycan might mainly act through the regulation of cell adhesion, this remains an hypothesis until it is directly shown. In order to convincingly demonstrate how Endoglycan regulates axon growth, the authors should directly measure cell-cell adhesion strength and quantify its dependence on the presence of Endoglycan.

We have included live imaging analyses as additional evidence and strong support for our conclusion that Endoglycan is a regulator of cell-cell adhesion by acting as a lubricant (Figure 6, Figure 6, Figure 8; Video 1, Video 2, Video 3, Video 4, Video 5, Results). Please note that it is NOT our intention to say that Endoglycan regulates axon growth. This does not appear to be the case, as axons get to the floor plate at the expected time (observations made by the analyses of open-book preparations, see Figure 2, and observed in live imaging preparations). There is no change in the pre-crossing axon tract (Figure 13). Axons are only affected when they are interacting with their intermediate target, this is during midline crossing, where indeed axons are slower in the absence of Endoglycan (thus stick more) and faster upon overexpression of Endoglycan (when they stick less). We do not think that they are affected in their intrinsic growth potential. This is also reflected by the fact that axons are faster in the second half of the floor plate under the conditions shown in Figure 6E_2_. If axons would be affected in intrinsic growth potential, they would be expected to maintain their speed.

We accept the criticism that we do not provide a direct measurement for the adhesive strength with and without Endoglycan. We tried to measure adhesive strength in an in vitro assay as designed by Vance Lemmon and colleagues (Lemmon et al., (1992)). However, we did not succeed in measuring the differences reproducibly. We therefore opted to carry out the live imaging studies, which support a change in the interaction/adhesion between floor-plate cells and growth cones (see Figure 8 and Video4, Video 5).

Reviewer #2:In this manuscript by Baeriswyl et al., the authors describe defects in spinal commissural axon guidance and cerebellar Purkinje cell migration that occur after interfering with the function of the sialomucin Endoglycan. Baeriswyl et al., suggest that these phenotypes reflect a function of Endoglycan in reducing cell-cell adhesion.The authors show that Endoglycan knockdown in the chick spinal cord floor plate or floor plate plus commissural neurons disrupts clustering of floor plate cells and causes increased axon tortuosity during floor plate crossing and abnormal caudal turning of commissural axons after midline crossing. Expression of Endoglycan using commissural neuron- or floor plate-specific promoters can rescue these axon guidance defects, depending on expression levels. The authors further show that knockdown of Endoglycan in the floor plate can suppress the premature commissural axon turning phenotype that results from NrCAM knockdown, and in vitro experiments support the idea that Endoglycan can reduce adhesion of motor neurons to heterologous cells. Lastly, knockdown of Endoglycan in the cerebellum disrupts Purkinje cell positioning and overall cerebellar morphology.This work presents a series of novel observations, and the idea that Endoglycan modulates cell-cell adhesion to control axon guidance and neuronal migration is of potential interest to the field, even though such an anti-adhesive mechanism for axon guidance has been extensively explored in the context of other molecules, e.g. NCAM. The main conclusions about the mechanism of Endoglycan function in axon guidance and cell migration, however, are not sufficiently supported by the data, and some important control experiments are missing. These problems, together with the limited conceptual novelty of the findings, do not make this manuscript a strong candidate for publication in eLife.1) The experiments involving Endoglycan knockdown in the spinal cord are difficult to interpret for multiple reasons. It is not clear that increased axon-floor plate adhesion is responsible for the observed effects.

We have added live imaging studies (Figure 6, Figure 7, Figure 8) demonstrating that the observed differences in axonal midline crossing are indeed explained by aberrant interactions between growth cones and floor-plate cells (see Figure 8 and Video 4, Video 5; subsection “Endoglycan acts as ‘lubricant’ for growth cone movement in the floor plate”).

a) Knockdown is targeted to the floor plate or "one half of the spinal cord including the floor plate". Data that validate successful knockdown in the targeted cell populations should be included. Also, why is knockdown in commissural neurons alone not attempted?

We did knockdown Endoglycan only dorsally (see graphic in Figure 1D and Figure 1J, Results). Moreover, the requirement for Endoglycan in dI1 neurons can also be concluded from our rescue experiments, where the expression of Endoglycan only in dI1 neurons, using the Math1::Endoglycan construct, was sufficient to rescue the phenotype observed after downregulation of Endoglycan in one side of the spinal cord including the floor plate (Results).

b) The data showing normal floor plate differentiation (Figure S3) are not sufficient to exclude abnormal tissue patterning as a cause for the axon guidance defects. Shh expression appears reduced (panel C), which could explain abnormal turning of axons after midline crossing. Patterning along the anterior-posterior axis and expression of rostro-caudal guidance cue gradients should be examined quantitatively.

We have added more detailed analyses of spinal cord patterning after downregulation of Endoglycan in one half of the spinal cord including the floor plate (new Figure 3—figure supplement 1, Results). Furthermore, we extended the analysis of Shh expression to include also Wnt5a along the anteroposterior axis and found no changes in any of the experimental groups compared to controls (Figure 3—figure supplement 4, Results). Along the same lines, we also did not find any differences in the expression of the known guidance molecules Axonin-1/Contactin2 and NrCAM (Figure 3—figure supplement 3, Results).

c) Are the axon guidance defects simply a result of reduced floor plate cell clustering? Can guidance errors be observed before changes in floor plate morphology?

The gaps in the floor plate are unlikely to explain the axon guidance phenotypes alone, as the floor plate is still largely intact as a structure. Most importantly, evidence from our in vivo and live cell imaging experiments suggests that axons cause the disruption of the floor plate (see Figure 8 and Video 4, Video 5; Results). See also response to reviewer 1.

d) The authors claim that changes in floor plate morphology coincide with commissural axon crossing of the midline and interpret this as support for the idea that axon-floor plate contact causes the change in floor plate cell clustering. However, axons have clearly already crossed the floor plate at HH21 (Figure 2N,O) before the floor plate phenotype appears. Moreover, successful knockdown of Endoglycan by HH21 would have to be demonstrated to allow interpretation of this experiment. Lastly, even if the timing was consistent, it should be directly tested whether reduced floor plate cell clustering after Endoglycan knockdown depends on commissural axon crossing, e.g. by ablating commissural neurons.

As the reviewer correctly states, some axons have already crossed the midline by HH21. These are mostly more ventral populations of commissural neurons. The changes in floor plate integrity require axons to dislocate previously crossing axon from contact with floor-plate cells. This is why we chose to show HH21, a stage, where we did not yet see the gaps in the floor plate but where some early axons have already crossed (Figure 2). This is to show that the floor plate does not disintegrate due to downregulation of Endoglycan but is affected only at later stages when axon-floor-plate cell contacts have to be dissolved by later following axons. This conclusion is also confirmed by our live-imaging results where we saw aberrant axon-floor-plate cell interactions only after 10 hours, indicating that it is the excessive adhesion of axons with the floor-plate cells that results in the dislocation of floor-plate cells (Figure 8, Video 4 and Video 5).

e) It is important to show that a control dsRNA will not "rescue" the NrCAM knockdown defect. Moreover, rescue of the defect by Endoglycan knockdown could result from reduced floor plate repulsion and other mechanisms besides an effect on axon-floor plate adhesion, so the explanatory power of this experiment is limited. Lastly, does loss of NrCAM affect the Endoglycan knockdown phenotypes (turning after crossing, tortuous trajectory during crossing)?

As suggested by the reviewer, we have extended our analysis of the concomitant downregulation of NrCAM and Endoglycan with dsCD34 and dsPodxl (Figure 5, Results). Both dsRNAs were not able to counteract the loss of NrCAM function, as axons were still found to turn aberrantly along the ipsilateral floor-plate border. Both genes are expressed in the spinal cord during the time window of commissural axon navigation across the floor plate (Figure 2—figure supplement 2), although mRNA for Podxl1 was only found in precursors of dI1 neurons, not mature neurons. However, the protein may of course persist in neurons.

And indeed, the “corkscrew-phenotype” was no longer observed after concomitant downregulation of NrCAM and Endoglycan.

2) Direct support for an anti-adhesive role of Endoglycan in commissural neurons and Purkinje cells is lacking.a) How would increased floor plate adhesion explain the rostro-caudal axon guidance defects?

This is still an open question. We can rule out a change in morphogen gradients, as both the Shh gradient and Wnt5a expression were unchanged in embryos electroporated with dsEndoglycan (Figure 3—figure supplement 4).

We can only speculate that the changes in axon-floor-plate cell interactions will prevent growth cones from reading these cues correctly (Discussion).

b) The in vitro results using spinal motor neurons do not connect to the relevant cell types. Could commissural neuron and Purkinje cell adhesion/growth be studied in similar assays?

We do not have a method to culture Purkinje cells and because this in vitro assay is used to support our results from in vivo experiments, we opted for live cell imaging rather than an in vitro assay with commissural neurons. These live cell imaging experiments clearly demonstrate that aberrant interactions between growth cones and floor plate cells explain the observed in vivo phenotypes (see Figure 8 and Video 4 and Video 5).

c) It would be important to model adhesive commissural axon-floor plate interactions in vitro and study the effect of Endoglycan gain or loss of function in this system. The same applies for Purkinje neurons and the substrate for their migration.

An in vitro model of floor plate-growth cone interactions would be very difficult to model, as on both sides a plethora of molecules is involved in a dynamically changing pattern. Therefore, as mentioned above, we opted for an ex vivo live imaging approach. This allows for visualization of individual growth cones and their interaction with the floor plate, thus providing the advantages of an in vitro system, but without the disadvantages of the in vitro system, as the in vivo complexity is maintained. The results of these experiments are completely in line with our model and confirm that Endoglycan fine-tunes the interactions between growth cones and floor-plate cells. (See Figure 6, Figure 7, and Figure 8, along with the Video 1, Video 2, Video 3, Video 4, Video 5, Results).

d) The cerebellar phenotype after Endoglycan knockdown could be explained by numerous mechanisms other than increased Purkinje neuron adhesion. Could removal of cell adhesion molecules rescue the migration defect?

In contrast to the situation at the floor plate, the molecules involved in Purkinje cell migration have not been identified. Therefore, it is not possible to do the experiment suggested by the reviewer.

Reviewer #3:The authors took advantage of a substractive hybridization screen that they made several years ago to investigate the functions of endoglycan in the navigation of spinal commissural projections and in the development to the cerebellum. Endoglycan belongs to a family of cyalomucins also comprising CD34 and Podocalyxin, known to regulate cell-cell interactions through anti-adhesive properties.The authors report for the first time a requirement for endoglycan in the two contexts that they have examined, which is thus an interesting finding. They also propose that endoglycan functions are mediated via anti-adhesive properties.The mechanisms that regulate and adjust cell- cell adhesive contacts and how these contacts are coupled to guidance decisions are yet poorly known. Therefore, the study addresses an important and underestimated aspect of axon guidance. Nevertheless, I have the feeling that, beyond the demonstration through RNA interference that endoglycan is necessary for proper navigation and migration, the basic mechanisms underlying the contribution of endoglycan and what makes it that it plays a role at some specific steps of the navigation rather than all along, have been superficially addressed. We are indeed let with a very unclear picture on how it contributes to enable proper navigation of the floor plate, and proper purkinje cell migration. The substractive screen was designed to pick up genes important for rostral turning but from the described phenotypes, it looks like what is primarily affected is the FP crossing. Indeed, the axons still appear able to turn, even though the turning is abnormally disconnected to the crossing. Overall and for these reasons, I have a number of issues which I think are necessary to address. They are listed below.1) Subtractive hybridization screen: the authors should describe the general strategy and method, even though both have been reported in their previous studies. The rationale is needed to correlate the approach with the expected and observed phenotypes of endoglycan knock-down. In particular, the screen was designed for identifying cues instructing rostral turning guidance. How is regulation of cell adhesion during floor plate relevant for the turning? should be expect a rostral-caudal gradient of endoglycan?

The screen was designed to identify genes with differential expression in the floor plate at stage HH26, when axons have crossed, compared to HH19/20, before dI1 axons have entered the floor plate. From the identified genes, we randomly selected some for functional testing. Endoglycan was one of them. In fact, it is not even clear why it showed up in our screen, as it is expressed in the floor plate at both stages. Therefore, we feel that it is not indicated to put too much emphasis on this screen. The reason why we looked at this gene in more detail really was its functional contribution to proper floor plate navigation. We did add more information about the screen to the Materials and methods part.

2) In Figure 2 at HH21: the axonin labeling seen in the picture is indeed very strong, not as if there were only a few axons that already had crossed, as indicated in the result section of the manuscript.

The reason to choose HH21 was to use the oldest stage, where we did not see gaps in the floor plate, because only more ventral populations of commissural axons have already crossed the midline. No or only few dI1 axons are in the floor plate at this stage. This strongly supports our conclusion that the floor plate is not directly affected by the loss of Endoglycan but that it is the dynamic contact between axons and the floor plate that causes the changes in floor-plate integrity. Later crossing axons are competing with previously crossing axons for contact with the floor-plate cells. These findings are further supported by the newly added live imaging experiments showing changes in growth cone-floor plate interaction after downregulation of Endoglycan or overexpression of Endoglycan that are consistent with the idea that adhesion between growth cones and floor-plate cells is fine-tuned by the amount of Endoglycan. (Please see Figure 6, Figure 7, and Figure 8, along with the Video 1, Video 2, Video 3, Video 4, Video 5, Results).

3) The HH21 untreated control is not shown. This is needed to compare the shape of the FP at this stage or at least would it be the case, the authors should make it clear that the shape is expected to be similar to that at later stage.

We have added untreated controls to new Figure 2, panels M-O, quantification in V and W.

The FP appears strongly disorganized, based on the staining of FP cell soma and nuclei. It would strongly add to illustrate how radial fibers of FP cells look like, because commissural axons navigate within this fiber network. A better morphological characterization of the FP structure is also needed. Are cells losing their bipolar morphology? Are the radial fiber still present and attached to the basal side? In the cerebellum, endoglycan depletion alters cell proliferation, could it be the case for FP cells?The authors consider that since at HH21 the FP is unaffected when only a few axons hare reached the FP, the FP disorganization at HH26 is therefore an indirect consequence of the presence of the axons. This conclusion is weak and lacks experimental support.

We have looked at apoptosis of floor-plate cells, but did not see any Cleaved-caspase3-positive cells in any of the conditions. Furthermore, based on the analysis of patterning (Figure 3—figure supplement 1) or the expression of Shh and Wnt5a in the floor plate, we do not have any indication that the floor plate may have any major problems in structure or function. This is also based on our analysis of live axons crossing the floor plate. There are single mislocalized cells in the absence of Endoglycan, as also shown in Figure 2, but no major disruption of floor plate morphology or function (Video 1, Video 2, Video 3, Video 4, Video 5, Figure 8).

4) The disorganization of FP cells could well alter the presentation pattern of local guidance cues or local cell adhesion molecules. Expression profiles of the principal players should be investigated at protein levels, when antibodies are available.

We have carried out these control experiments and compared the expression of Contactin-2/Axonin-1, NrCAM and the expression of Shh and Wnt5a between controls and the experimental groups with less or more Endoglycan (electroporation with dsEndo or overexpression of Endoglycan, respectively). We did not find any differences in expression of any of these well-known guidance cues, suggesting that Endoglycan does not act on axonal midline crossing by interfering with the expression of other guidance cues but rather by modulating the interaction between growth cone and floor-plate molecules. Results are shown in Figure 3—figure supplement 3 and Figure 3—figure supplement 4, respectively; Results.

5) How knock-down of endoglycan impacts on levels of endoglycan is not shown. This is also true for dose-dependent rescue: the authors report that axon guidance was rescued by addition of endoglycan in dI1 neurons at a low concentration or in floor-plate cells at a medium concentration. One would like to see whether these different conditions really result in differences of endoglycan levels. It might be possible to get insights into endoglycan levels be done, maybe with western blot of pieces of spinal cords. On the least, knock-down efficiency and rescue could be assessed on cells transfected with tagged endoglycan?

We have added the quantification of Endoglycan knockdown as Figure 2—figure supplement 3. Because we do not have antibodies that recognize Endoglycan, quantification of Endoglycan with the precision level required to measure these differences is not possible. We would like to point out that the electroporation parameters used in these experiments successfully transfect about 50% of the cells. Therefore, knockdown of Endoglycan will result in only 50% of Endoglycan decrease as the theoretical maximum. However, at the cellular level, our results indicate that the loss of Endoglycan expression is close to or complete.

We did not attempt to see differences between the different concentrations used in our rescue experiments (shown as Figure 3). The two enhancers have different efficiencies in driving expression. Again, we would need specific antibodies to be able to see differences.

6) The anti-adhesive role of endoglycan during FP crossing is rather deduced from the literature than supported by the data. The authors need to investigate first whether manipulating the adhesive/anti-adhesive balance results in alterations of FP crossing and post-crossing. According to their model, abrogating endoglycan results in an unbalanced weight of NrCAM over contactin. The authors could for example test whether over expressing contactin strictly mimics NrCAM/endoglycan knock-down.

We have added new evidence for the change in adhesive properties in our Videos and in Figure 8, where the aberrant contact between axons and floor-plate cells is shown in detail (Results). Furthermore, we show that the anti-adhesive function is due to the glycosylation of Endoglycan (Figure 5—figure supplement 2; Results). We extended the analysis of the counterbalance between Contactin2/NrCAM interactions (the adhesion-promoting interaction) and Endoglycan level (the adhesion-impeding interaction) in new Figure 5 (Results). We do not quite understand the statement of the reviewer: …”unbalanced weight of NrCAM over contactin”. The experiments here and those done in a previous study (Philipp et al., 2012) are in line with the notion that Contactin2 and NrCAM interact to make axons enter the floor plate. This interaction is weakened when NrCAM is downregulated, resulting in aberrant ipsilateral turns. However, the concomitant strengthening of axonfloor-plate interactions by downregulation of Endoglycan rescues the phenotype specifically. Other family members of Endoglycan, Podocalyxin1 or CD34, have no effect.

7) Why were the in vitro experiments achieved with motoneurons rather than commissural neurons? Moreover, the performed analysis rather assesses whether endoglycan regulates axon outgrowth, not really cell-cell contacts and adhesion. it is somehow surprising that this outgrowth effect, if it applies also to commissural axons, does not result in alteration of the pre-crossing navigation, for example by delaying the growth towards the FP.

We have changed this experiment, as we do not think that Endoglycan has any effect on axon growth. Therefore, we now only show adhesion of cells (Figure 4). However, we have expanded the analysis and show that the anti-adhesive effect is due to the glycosylation of Endoglycan (Figure 5—figure supplement 2; Results). We did not find any difference in axonal arrival at the floor plate, nor did we see any differences in the pre-crossing trajectory between the experimental groups or in comparison to the controls (Figure 13; Results).

8) "Cell-autonomous" versus "non-cell autonomous" endoglycan contributions are unclear for FP navigation. On the one hand endoglycan is reported enriched in the FP at HH20, a stage when there are not impact of endoglycan KD on FP cell organization. Later on, higher expression is found in the dorsal spinal cord where commissural neurons are, and at this stage they have crossed the FP. Thus, this increase might rather be important for post-crossing navigation.

We have carried out studies to address the location of Endoglycan function in our system. Based on these (Figure 3), it does not matter where Endoglycan comes from, as the rescue works when we replace Endoglycan either in dI1 neurons (with the Math1 enhancer-driven construct) or in the floor plate (with the Hoxa1-driven construct). These findings are in line and support our conclusion that Endoglycan acts as a lubricant: The amount matters but not where it comes from. (Results).

9) Is endoglycan relevant for axon fasciculation?

No, we did not see any changes in axons extending towards or across the midline (Figure 13).

10) The impact of endoglycan deletion on cerebellar development is rather impressive. The interpretations are simplest because Purkinje cells are the only source of endoglycan. This makes it possible to reconstitute the sequence of direct and indirect events leading to the different abnormalities the authors found. Nevertheless, very few attempts are done for characterizing the nature of the interactions and adhesion required during Purkinje cell migration. One would like to know more about the adhesion molecules that are needed for this cell-type, or the nature of the migration process that is impaired. Is it that the leading process sticks to the substrate, or is the nuclei translocation prevented?

We agree that it would be great to know more about the adhesive interactions driving Purkinje cell migration. However, this is clearly beyond the scope of this study.

11) Also, the Discussion should be extended on the question of folding. Is folding defect a direct consequence of the decrease amount of produced granule cells? or the lack of purkinje cells at their final position?

It has been shown that Purkinje cells are the source of Shh and that Shh signaling promotes granule cell proliferation. The increasing number of granule cells is responsible for the folding of the cerebellar anlage. These studies are very concisely summarized in a review by De Luca et al., (2016). We have added this to the text (Discussion).

12) The authors mention that “Pathfinding was normal in embryos electroporated with dsRNA derived from Podocalyxin". They also quote in their introduction that this cyalomucin is also expressed in the developing nervous system. Therefore, why this experiment was done is unclear, as well as a conclusion lacks. Was it done to document a specific contribution of endoglycan over the other cyalomucins? or a specific functional property of endoglycan?

The analysis of the other two sialomucin family members supports the specificity of our approach. We used them as controls in our initial loss-of-function studies (Figure 1). Furthermore, we extended the in vivo experiments demonstrating the balance of adhesion between axons and floor plate by adding the effect of dsPodocalyxin and dsCD34 (Figure 5).

[Editors’ note: what follows is the authors’ response to the second round of review.]

The three reviewers believe that your study offers some new, potentially important insights into the regulation of axon pathfinding. The findings are certainly new and exciting. think that the work nicely shows that Endoglycan contributes to commissural axon navigation. In this revision, the phenotypes have been characterized in more detail, and important control experiments were added.Nonetheless, the reviewers do not find supporting evidence for your model, that Endoglycan facilitates floor plate navigation by controlling the strength of "adhesion and anti-adhesion" contacts resulting from cell-axon or axon-axon contacts during navigation.

We are glad to hear that the reviewers consider our study demonstrating a regulatory role of Endoglycan in axon guidance new and exciting. Of course, we are also disappointed that the reviewers are still not convinced of the mechanisms of Endoglycan activity that we propose. We kept trying to find a solution for the direct measurement of adhesive strength in vitro. Despite the fact that this seems to be a simple experiment, it is not. However, in the meantime, we did succeed in finding a way that demonstrates directly that Endoglycan changes the adhesive strength of growth cones of commissural neurons and carpet cells. For this purpose, we adapted the “growth cone blasting” experiment used by Vance Lemmon and colleagues in the 90s (see below).

In the reviewer consultation after the reviews below were submitted, the reviewers were in agreement in believing that the in vivo analyses alone do not truly inform on the mechanism of Eendoglycan-mediated cell adhesion. They would like to see in vitro experiments of the type your lab should be able to execute, but using commissural, not motor, neurons, to complement your in vivo analysis. Examining adhesion of commissural neurons to heterologous cells and/or floor plate cells in vitro, combined with manipulations of Endoglycan expression, could directly support (or not) the idea that Endoglycan modulates adhesive interactions between commissural axons and floor plate cells. Without such experiments, the statements about adhesion need to be toned down; adhesion should be critically discussed as one possible mechanism.

As suggested by the reviewers, we have carried out in vitro experiments that demonstrate directly that the adhesion between growth cones of commissural neurons and HEK cells expressing Endoglycan is strongly reduced compared to control HEK cells (new Figure 5). As also shown in our in vivo experiments, and the in vitro experiments presented in our previous versions of the manuscript, Endoglycan lowers adhesion no matter whether it is provided by the neurons or by the floor plate or HEK cell carpet. This is what you would expect from a “lubricant”. In the current, revised version, of our manuscript, we have added adhesion assays with commissural neurons and also demonstrated that the effect of Endoglycan depends on its post-translational modification (new Figure 4). Thus, as suggested by the reviewers, we have repeated our findings from the original manuscript, where we did these experiments with motoneurons, with commissural neurons. One of the reasons to use motoneurons for the in vitro adhesion assays was to demonstrate that the effect of Endoglycan is not restricted to commissural neurons but would generally be valid in areas where it is expressed. For this reason, we kept the results obtained with motoneurons in the supplementary material (Supplementary Figures 7 and 8).

Reviewer #1:In the revised version of their manuscript, Baeriswyl et al., have addressed several of my concerns. Additional control experiments have helped solidify some of the authors' conclusions, and the live imaging experiments provide new information about the effect of Endoglycan knockdown and overexpression on commissural axon behavior during floor plate crossing. However, direct support for the anti-adhesive or "lubricating" role of Endoglycan in the context of commissural axon-floor plate or Purkinje neuron-substrate interactions is still lacking. I am surprised that the authors decided not to include some very doable experiments in their extensive revision work to strengthen this hypothesis. I therefore believe that the authors either need to tone down their conclusions about molecular mechanism or perform the necessary experiments to test their interpretation more directly.

We hope that we could relieve any doubts about the validity of our claims about the “lubricant” or adhesion modulating function of Endoglycan with our new in vitro experiments involving commissural neurons (Figure 4 and Figure 5).

1) The experiments involving motor neurons cultured on Endoglycan-expressing HEK cells should be carried out with commissural neurons instead. These neurons are easy to grow in culture, making this a very feasible experiment. Examining adhesion of commissural neurons to heterologous cells and/or floor plate cells in vitro, combined with manipulations of Endoglycan expression, could directly support (or not) the idea that Endoglycan modulates adhesive interactions between commissural axons and floor plate cells. Without direct evidence of this kind, the language centered around anti-adhesive or lubricating properties of Endoglycan in an axon guidance context does not seem justified.

As mentioned above, these experiments have been done now (Figure 4). We have gone one step further and adapted a “growth cone blasting” experiment (Lemmon et al., 1992) that was originally used for cell adhesion molecules coated on tissue culture plastic to our needs.

In brief, we have cultured commissural neurons on a carpet of HEK cells (either control cells or cells expressing Endoglycan) and then we measured the time it took to detach growth cones from the HEK cell carpet depending on the presence of Endoglycan. To this end, we have placed a micropipette from which we directed a constant flow of PBS at the growth cone from a defined angle. We only compared times between control and Endoglycan conditions that were obtained with the same pipette to make sure that the forces affecting the growth cones were constant and reproducible. Using this assay, we could demonstrate that growth cones were detached (or blasted off) much faster, when they were growing on HEK cells expressing Endoglycan compared to control HEK cells.

As we expected, we found the same result when we used HEK cells without Endoglycan and overexpressed Endoglycan in the neurons instead. This again demonstrated that the source of Endoglycan is not important but its presence, in line with what you would expect from a lubricant.

2) The data on cerebellum development after Endoglycan knockdown are difficult to interpret due to the dramatic effects on overall tissue morphology. Without extensive experiments focused on molecular mechanisms underlying the phenotype, these findings do not integrate well with the rest of the paper and should be removed.

Yes, we agree that the changes in cerebellar morphology are dramatic. However, this is what has been described as a consequence of reduced granule cell proliferation. Because granule cell proliferation is dependent on Shh released by Purkinje cells, the observed phenotype is absolutely in line with these descriptions and in line with the developmental trajectory of the cerebellum. The review by De Luca et al., (2016) summarizes the literature and the studies describing the effect of Purkinje cell-derived Shh on granule cell proliferation and cerebellar morphology.

Reviewer #2:The authors have conducted several new experiments to support their hypothesis that Endoglycan is regulating cell-cell adhesion. While I still think that the manuscript presents exciting new data on axon guidance, the authors still do not provide any direct evidence for an involvement of Endoglycan in regulating cell-cell adhesion.I appreciate that the authors tried to measure adhesive strength in an in vitro assay as designed by Vance Lemmon and colleagues. However, I am a bit concerned about their statement "we did not succeed in measuring the differences reproducibly". Perhaps there are just no differences?

No, our difficulties were not due to the fact that there were no differences. But the experiment as we tried it initially was extremely artificial and subject to variations. We kept trying and reduced the pulsatile nature of the medium stream from the micropipette. Using a cell carpet rather than just a cell adhesion molecule coated on plastic probably also helped to avoid artefacts. With the version of the “growth cone blasting” experiments as presented in the revised manuscript, we are confident to have directly shown that the adhesive forces between growth cones and HEK cells are reduced by the presence of Endoglycan.

Unfortunately, also the ablation of commissural neurons suggested by reviewer 2 was not attempted. This experiment would have tested whether the reduced floor plate cell clustering after Endoglycan knockdown indeed depends on direct interactions between axons and floor place cells, which is currently not clear.

We have opted for the above-mentioned version of demonstrating the anti-adhesive role of Endoglycan. Ablating commissural neurons in very young spinal cord preparations would have caused tremendous morphological changes and artifacts that would prevent the clear interpretation of the results.

The authors present new figures (Figure 6, Figure 7 and Figure 8) and movies "demonstrating that the observed differences in axonal midline crossing are indeed explained by aberrant interactions between growth cones and floor-plate cells". However, the nature of these interactions remains unclear. The 'corkscrew' phenotype shown in Figure 8, Video 4 and Video 5 could also be explained by a change in microtubule dynamics (Krieg et al., 2017). Endoglycan could very well interfere with other signals present during floor plate crossing, providing an alternative explanation for the observed axonal growth patterns.

Because our experiments clearly show that Endoglycan has its effect independent of its source, we exclude an effect on microtubule dynamics. It would also be unlikely that a change in microtubule dynamics would manifest itself only in one half of the floor plate or not at all in pre-crossing axon growth. For these reasons, we belief that our explanation that is supported by live imaging results and now corroborated by the new in vitro adhesion assays better explains the activity of Endoglycan.

The authors also claim that "we now only show adhesion of cells (Figure 4)." Figure 4 shows a reduction in cell number, which could not only be a consequence of reduced adhesion but also because of other effects, such as apoptosis.

We have no evidence for any effect on apoptosis, as in none of our experiments Endoglycan had an effect on cell death, neither enhancing nor preventing cell death.

If I understand correctly, the authors suggest that axons pull on FP cells: "Most importantly, evidence from our in vivo and live cell imaging experiments suggests that axons cause the disruption of the floor plate (see Figure 8 and Video 1 and Video 3, Results)." I cannot see any such evidence in the presented data. Time lapse movies showing how axons pull on FP cells are currently missing. The ablation of commissural neurons mentioned above would also provide such evidence.

Maybe our wording was not clear enough. What we mean is that growth cones and axons crossing the floor plate stick to floor-plate cells mediated by cell adhesion molecules, but obviously, this interaction is transient. Axons crossing the axons after the first ones have done so need to displace the previously crossing axons from the floor-plate cells. This requires a detachment between axons and floor-plate cells. However, if this adhesion is too strong (as seen in the absence of Endoglycan) then the “weakest bond” that is broken is not the axon-floorplate cell contact but it may be the contact between floor-plate cells that results in the gaps in floor plate morphology (Figure 2). Another consequence of this increased adhesion between growth cones and floor-plate cells is the “curling back” of axons during floor plate crossing (now Figure 9 and Video 5 and Video 6). The enhanced bond between growth cone and floor-plate cell distorts the straight trajectory of axons resulting in the 'corkscrew' phenotype.

Overall, I'm not sure if I understand the concept of a lubricant in this context. The authors state: "Endoglycan is not affecting adhesion between floor-plate cells, as it is not a cell adhesion molecule, or an “anti-adhesion” molecule to be precise. Rather Endoglycan acts as a “lubricant” that is required between axons and floor-plate cells, or between cells that move with respect to each other, because it allows for the necessary dynamics in the adhesion between growth cones and floor-plate cells mediated by adhesion molecules." What is the difference between anti-adhesion and a lubricant? Both reduce friction, it should be the same? And do the authors suggest that whole axons are moving over the floor plate: "In contrast, at the floor plate axons need to move relative to the floor plate."? As far as I'm aware, axons per se do not move much. Growth is mostly achieved at the distal end of the axon. And if a lubricant would allow sliding there, growth cones could not generate forces required for their motility. Furthermore, if adhesion would be reduced, growth cones should also be smaller. Yet, Figure 11 seems to suggest that growth cone sizes are similar in all conditions.

The reviewer refers to the rebuttal letter, where I tried to explain how we envisage the role of Endoglycan. But it looks like the reviewer misunderstood our concept of how Endoglycan acts. What we wanted to point out is that Endoglycan in not an adhesion molecule but also not a repulsive molecule. Rather, our in vivo and in vitro results support a model that Endoglycan is modulating the interaction of a variety of cell adhesion molecules without specifically binding to them. The activity is best compared to a lubricant that allows movement of two parts against each other due to a decrease in friction. In biological terms, this would be a decrease in cell-cell adhesion by preventing tight interaction of cell adhesion molecules. Endoglycan with its extensive O-glycosylation and sialylation is perfectly suited to interfere with molecular interactions between growth cones and floor plate cells or between migrating Purkinje cells and other precursors in the developing cerebellum. We showed that the post-translational modifications, O-glycosylation and sialylation, are both required for the function of Endoglycan.

Thus, while the authors clearly demonstrate, for the first time, the importance of Endoglycan in axon guidance and neuronal migration, I am still not convinced that it acts mostly through regulation of cell-cell adhesion as suggested by the authors.

We hope that the additional in vitro experiments demonstrating clearly a decrease in cell-growth cone adhesion due to the presence of Endoglycan, no matter whether it is provided by the cell carpet or the growth cone, convinces the reviewer of our model.

Reviewer #3:The study has been significantly extended, in particular with a series of novel experiments performed with live imaging. The resulting data clarify the role of endoglycan and its mode of action. For example, the authors observed that manipulations of endoglycan levels have impact on the velocity of commissural axons and growth cone size. Differences appear tiny but statistically significant. They propose a model whereby endoglycan provided by FP cells and axons acts as a lubricant to facilitate the FP navigation and the axon-FP cell contacts. This is an interesting idea that beyond endoglycan, shed light on yet poorly understood roles of modulators.In addition, the authors addressed several concerns that were raised by their previous version of the work. They checked whether manipulations of endoglycan alter the expression of molecules known to guide commissural axons, which as far as we can tell from general expression patterns is not the case.I also think that the part on the cerebellum model is now more integrated to the rest of the work, and I agree that it enable to make a broader message on the functional properties of endoglycan.

We thank the reviewer for this positive assessment of our revised version of the study.

I still consider that, for better consistency, the experiments to demonstrate the regulation of adhesion by endoglycan should have been carried out on the population of neurons that is investigated here (commissural neurons) and not on motoneurons. That said, I believe that what the authors report with motoneurons might apply to commissural neurons and the findings are consistent with their model on commissural axon navigation. The authors modified their initial analysis, now concentrating on adhesion rather than outgrowth, which is to me very pertinent. They also carried out some biochemical analysis to document endoglycan mode of action, showing contribution of glycosylation.

As suggested, we have repeated the adhesion assay originally done with motoneurons with commissural neurons (now Figure 4). In addition, we have added a series of “growth cone blasting” experiments where we either cultures commissural neurons on HEK cells expressing Endoglycan or overexpressed Endoglycan in commissural neurons. These experiments demonstrate that Endoglycan reduces growth cone adhesion no matter where it comes from. Growth cones were detached faster when they were targeted by a stream of PBS delivered from a micropipette (new Figure 5).

Overall, I found that the study is improved by the novel data.Here are some specific comments:For all figures related to the time-lapse imaging.: the number of experiments, embryos etc… are not indicated.

The number of replicates and/or embryos are given in Table 2 and we have added the values to the figure legend (Figure 8).

It seems to me that there are some redundancy between Figure 6 and Figure 7 in the questions that the experiments address.

Yes, Figure 6 and Figure 7 (now Figure 7 and Figure 8) are in fact the same series of experiments. Figure 8 is the quantification of the observations explained in Figure 7. We chose to divide the imaging examples and the quantification into two figures due to size.

[Editors' note: further revisions were suggested prior to acceptance, as described below.]

Essential revisions:Your study highlights Endoglycan in regulating neuronal growth and migration, pointing to a mechanism involving anti-adhesion. The reviewers felt that you have addressed all the major criticisms, and with the addition of your apt in vitro experiments addressing cell adhesion (new Figure 5), that the manuscript is generally improved. They comment that this will be "a very nice paper that significantly contributes to the field".There are still revisions called for, as listed below, primarily altering the wording used to describe the action of endoglycan. Moreover, the reviewers encourage you to consider removing the data on the cerebellum.I) Cerebellar data: Two of the three reviewers urge you to consider removing the cerebellar data on Purkinje cells, as this analysis distracts from the now well fleshed-out message relating to axon guidance. The effects of Endoglycan knockdown on Purkinje cell migration are compelling per se, but the interpretation of these defects is complicated by the overall disrupted morphology of the cerebellum and lack of data that directly demonstrate whether and how Endoglycan modulates Purkinje cell adhesion to other relevant cell types. In your rebuttal, you indicate that defects in granule cell proliferation are consistent with impaired Purkinje cell migration. This is certainly true, but numerous alternative interpretations exist, and no attempt is made to probe the authors' model further. Without extensive experiments focused on molecular mechanisms underlying the phenotype, these findings do not integrate well with the rest of the paper. You might consider publishing the cerebellar work as a smaller study. One question is whether cell migration/translocation and axon outgrowth rely on adhesion/anti-adhesion in the cerebellum as they do in the floor plate.

We have decided to remove the part on the cerebellum from our paper. Our idea was to demonstrate that Endoglycan plays a general role in adhesion regulation in the CNS. But this clearly made the paper very long and added a second focus. Therefore, we removed Figure 10, Figure 11, Figure 12 of the original manuscript and the paragraphs from the Result section reporting our observations in the cerebellum. We rephrased parts of the Discussion accordingly.

II) Endoglycan at the floor plate:1) “Lubricant"? vs negatively regulating cell adhesion:a) While you now provide an in vitro experiment suggesting a role of endoglycans in regulating cell adhesion, there is no experimental evidence for a role of endoglycans as a lubricant in vivo (which would certainly be very difficult to achieve, if at all possible). This should be kept in mind when interpreting the data. You are thus encouraged to tone down the wording referring to the proposed mechanism throughout the manuscript, for example in statements such as "Endoglycan acts as “lubricant” for growth cone movement in the floor plate".

We have followed the suggestion and removed the work lubricant from the result section. We only use the word “lubricant” once in the Discussion to provide a metaphor for the anti-adhesive role of Endoglycan that facilitates motility of growth cones in the floor plate. We also toned down the wording throughout the text to be more cautious with our conclusions.

The reviewers caution that our evidence that Endoglycan lowers adhesive strength comes from in vitro experiments and that it would be very difficult or impossible to get similar data in vivo. This is certainly true. However, we would like to point out that supporting evidence that lower adhesive forces contribute to the observed phenotypes also comes from our ex vivo observations with live imaging. The Video 5 and Video 6 (like all other videos) show growth cones and axons in intact spinal cords. The videos and the frames shown in Figure 12 present axons stuck to floor-plate cells for some time as a possible explanation for the tortuous path taken by axons. We are aware that this ex vivo preparation is still not in vivo, but the axons are growing in their intact environment. We cannot measure adhesion in this situation but the observation of axons crossing floor plates in control spinal cords compared to spinal cords lacking Endoglycan suggests that cell-cell contacts are clearly different. Nonetheless, we have toned down our conclusions about the proposed mechanism of Endoglycan function, as suggested.

b) The second part of the title is not backed up by the data ("by negatively regulating cell-cell adhesion"); the abstract should explicitly state that adhesion is affected in vitro.

We have changed the title and rephrased the Abstract according to the suggestions by the reviewers.

c) Statements such as "Thus, the aberrant morphology of the floor plate at HH25 is explained by the inability of axons to break contacts with floor-plate cells in the absence of Endoglycan" should be phrased more carefully. Replacing "is explained" by "could be explained" would not make the study weaker but rather protect you in the future should it turn out that there is a different mechanism at play. Finally, because of the potential problems mentioned above associated with the blast experiment, you might omit the word "strong" in the sentence "Strong support for this hypothesis was contributed by in vitro findings that the adhesive strength… ".

We have rephrased the respective passages as suggested (Materials and methods).

d) It is admittedly difficult to distinguish a lubricant effect from a system that would control the strength of an adhesion or anti-adhesion effect (for example by controlling the number of binding partners or their availability). If the idea is that endoglycan does not target a specific adhesion mechanism but rather has a general effect, would you consider the role of endoglycan to be similar to the one attributed to PSA of NCAM adhesion molecules? This should be discussed.

We have rephrased our paragraph comparing the roles of PSA-NCAM and Endoglycan to strengthen the similarity between the mechanisms (Materials and methods).

2) Blasting assay: The growth cone blasting assay introduced in the revised version of the manuscript is a very difficult experiment, and your efforts should be praised. You have carefully considered some of the confounding factors; however, some open questions remain.a) Factors such as the distance of the pipette from the substrate (i.e., from the top surface of the HEK cells) and the area and height of the growth cones will also be crucial determinants of the time required to detach growth cones. How was the height of the pipette tip above the cells controlled, and how accurately could it be determined? Was the growth cone area the same between the groups? Where the experiments done blindly?

We have added a detailed description of the technical details that were crucial for the growth cone blasting experiments. The experiments were done by two people (A.D. and B.K.). The position of the pipette tip was carefully controlled by the motorized microscope stage taking the focal plane of the growth cone as basis. The horizontal distance was measured between the tip of the pipette and the edge of the targeted growth cone before the pump was started. The time it took to detach the growth cone from the HEK cell carpet was taken by a person blind to the experimental condition. To avoid influences on the time measurements from drifts in temperature or flow rate due to changes in the oxygen content in the PBS, one person was not blind to the experimental condition and switched between experimental conditions on purpose. We added a detailed description to the Materials and method section. We also added measurements of growth cone areas between the groups as new Figure 7, including source data. The analysis clearly demonstrated that despite some variability in growth cone size within each group. There was no significant difference between the different groups.

b) Furthermore, why was PBS used, which usually does not contain any calcium? Calcium has a key role in cell adhesion.Details of these aspects should be provided (including images of growth cones at higher magnification and, if possible, a quantification of growth cone areas), and the experiment should be critically discussed.

To get reproducible results, it was extremely important to degas the solution used to blast the growth cones. Culture medium could not have been degassed without changes in pH. Because the same solution was used for both control and experimental conditions, and because the absolute value was not of importance, we belief that the local decrease in Ca^2+^ does not matter. Importantly, the volumes added to the culture dish were very small compared to the volume of the medium. We have added a quantification of growth cone areas (Results and Figure 7, including source data showing growth cones at higher magnification along with measurements of growth cone areas).

3) A more detailed description of what constitutes an abnormal injection site in Figure 3 would greatly help with interpretation of those data. Quantification of "normal injection sites" in Figure 3 somewhat unsatisfying, especially for the comparison of gain-of-function and loss-of-function effects. After all, more adhesion and less adhesion should not produce the exact same phenotypes. Could the abnormal injection sites be further categorized, e.g. as stalled, 'corkscrew', premature turning, aberrant postcrossing A-P guidance etc.?

We have added an explanation about the phenotypes to the quantification in Table 1. Because the “failure to turn into the longitudinal axis” and the “stalling in the floor plate” are not independent of each other (strong stalling in the floor plate of close to all fibers will prevent the analysis of the “no turn” phenotype) we can only quantitatively compare the DiI injection sites with normal trajectories. Nonetheless, we have exchanged Table 1. The new version includes the average percentage of DiI injection sites with the “FP stalling” and the “no turn” phenotypes.

4) You have now used a carpet of HEK cells rather than just a cell adhesion molecule coated on plastic, which you argue "probably also helped to avoid artefacts". It is not clear how a homogeneous, controlled coating of a plastic surface with purified proteins should cause more artifacts than cells expressing a multitude of different proteins at their surface which may vary in space and time. While it is very difficult to see any details in Figure 4, HEK cells in the two groups (Figure 4A) seem to have different morphologies. If this is true, changes in HEK cells could lead to changes in cell-cell contacts to neurons that are independent of Endoglycans. This should be discussed.

We respectfully disagree. Axons are never growing on hard surfaces in vivo. They are adhering to cells in a 3D context. To have just one molecule offered together with tissue culture plastic is a very unphysiological environment. The HEK cells are inherently diverse in morphology and morphology might be different when one compares very confluent versus less confluent areas. However, there was no difference in morphology between the different conditions. The reviewers probably are maybe referring to the few round green cells in Figure 5A? These might be either dying cells or cells in the process of division. They have accumulated more background fluorescence. To show that there is a cell carpet in Figure 5A, the image was taken with suboptimal conditions.

5) In the Results section, the interpretation of defects as a problem with adhesion is introduced very early. It would be better to draw that conclusion later, after the experiments in Figure 4 and Figure 5. In general, some of the conclusions in the Results section that combine outcomes from multiple experiments might be better suited for inclusion in the Discussion.

We have rephrased this paragraph to make it sound like a model to be tested with further experiments rather than a conclusion.